# Stress diminishes outcome but enhances response representations during instrumental learning

**Jacqueline Katharina Meier[1], Bernhard P Staresina[2], Lars Schwabe[1]***

[1]Department of Cognitive Psychology, Universität Hamburg, Hamburg, Germany; [2]Department of Experimental Psychology, and Oxford Centre for Human Brain Activity, Wellcome Centre for Integrative Neuroimaging, Department of Psychiatry, University of Oxford, Oxford, United Kingdom

**Abstract** Stress may shift behavioural control from a goal-directed system that encodes action-outcome relationships to a habitual system that learns stimulus-response associations. Although this shift to habits is highly relevant for stress-related psychopathologies, limitations of existing behavioural paradigms hinder research from answering the fundamental question of whether the stress-induced bias to habits is due to reduced outcome processing or enhanced response processing at the time of stimulus presentation, or both. Here, we used EEG-based multivariate pattern analysis to decode neural outcome representations crucial for goal-directed control, as well as response representations during instrumental learning. We show that stress reduced outcome representations but enhanced response representations. Both were directly associated with a behavioural index of habitual responding. Furthermore, changes in outcome and response representations were uncorrelated, suggesting that these may reflect distinct processes. Our findings indicate that habitual behaviour under stress may be the result of both enhanced stimulus-response processing and diminished outcome processing.

*For correspondence:
lars.schwabe@uni-hamburg.de

Competing interest: The authors declare that no competing interests exist.

## Editor's evaluation

The authors combined a cleverly designed behavioral task with EEG-based multivariate pattern analysis and acute stress induction to assess the neural representations mediating an influence of stress on the balance between goal-directed and habitual responding. They found that stress induction reduced neural outcome representations, and that this representational change correlated with the degree of habitual performance. The topic and approach should be of interest to a wide audience, ranging from clinicians to economists and neuroscientists.

## Introduction

Adaptive behaviour in complex environments requires an intricate balance of deliberate action and efficient responding. For instance, in the supermarket faced with numerous products, it may certainly be helpful to weigh the pros and cons of a specific product before making a choice. Yet, thinking for hours about which toothpaste to buy may interfere with the goal of being home before dinner. The balance of thorough deliberation and efficiency is supported by at least two systems of behavioural control that operate in parallel: (i) a goal-directed system that learns when to perform the action required to achieve a desired outcome (in the form of stimulus [S] – response [R] – outcome [O] associations) and (ii) a habitual system that acquires S-R associations without any links to the response related outcome (**Adams, 1982**; **Adams and Dickinson, 1981**; **Balleine and Dickinson, 1998**). These

two systems are known to rely on distinct neural circuits. While the goal-directed system primarily relies on the medial prefrontal and orbitofrontal cortex as well as the dorsomedial striatum (anterior caudate in humans; *Balleine and O'Doherty, 2010*), the habitual system depends on the dorsolateral striatum (posterior lateral putamen in humans; *Balleine and Dickinson, 1998*; *Balleine and O'Doherty, 2010*; *Corbit and Balleine, 2003*; *Ostlund and Balleine, 2005*; *Tricomi et al., 2009*; *Valentin et al., 2007*; *Yin et al., 2006*). Moreover, it is commonly assumed that the goal-directed system guides early learning, whereas the habitual system takes over once a behaviour has been frequently repeated. Therefore, while buying toothpaste for the first time should be a goal-directed action, this choice should become more habitual if we have bought the specific toothpaste several times before. Adaptive behaviour requires the capacity to flexibly switch back from habitual to goal-directed control in response to environmental changes (e.g. when the previously bought toothpaste is out of stock). Lacking this flexibility in behavioural control may be detrimental to mental health. In particular, overreliance on habitual responding has been linked to several mental disorders, including drug addiction, obsessive-compulsive disorder, schizophrenia, eating disorders, and depression (*Everitt and Robbins, 2005*; *Voon et al., 2015*; *Griffiths et al., 2014*; *Robbins et al., 2012*; *Voon et al., 2015*).

Accumulating evidence indicates that stressful events may tip the balance from goal-directed to habitual control. Specifically, stress (and major stress mediators) has been shown to induce a shift from goal-directed action to habitual responding (*Braun and Hauber, 2013*; *Dias-Ferreira et al., 2009*; *Gourley et al., 2012*; *Schwabe et al., 2012*; *Schwabe and Wolf, 2009*; *Schwabe and Wolf, 2010*; *Schwabe and Wolf, 2013*; *Smeets et al., 2019*; *Soares et al., 2012*). Beyond its crucial relevance for our understanding of behavioural control in general, this stress-induced shift towards the habitual system may be a driving force in mental disorders that are characterized by dysfunctional stress responses on the one hand and aberrant habitual control on the other (*Adams et al., 2018*; *Goeders, 2004*; *Schwabe et al., 2011a*). Although the stress-induced bias to habitual responding may have critical clinical implications, the exact mechanisms through which stress modulates the balance of goal-directed and habitual control are not fully understood. In particular, previous research has been unable to address the fundamental question of whether the stress-induced bias towards habitual behaviour is due to a downregulation of the goal-directed system or the enhancement of the habit system, or both.

Canonical assays for the assessment of goal-directed and habitual control do not allow a distinction between these alternatives. These paradigms are based on the key distinctive feature of goal-directed and habitual control, that is, only the goal-directed system is sensitive to changes in the motivational value of the outcome in absence of new experience with it (*Adams, 1982*; *Dickinson and Balleine, 1994*). Accordingly, classical paradigms have tested the behavioural sensitivity to either a devaluation of an outcome or to the degradation of the action-outcome contingency (*Adams, 1982*; *Corbit and Balleine, 2003*; *Tanaka et al., 2008*; *Valentin et al., 2007*; *Yin et al., 2004*). Although these elegant paradigms provide valuable insight into the mechanisms involved in behavioural control, they are unable to determine whether increased responding to devalued or degraded actions is due to reduced outcome or enhanced S-R processing, or both.

Here, we aimed to overcome these shortcomings of classical paradigms that are used to determine these modes of behavioural control and to examine whether stress leads to an upregulation of response processing at the time of stimulus presentation or a downregulation of outcome-related processing, or both. To these ends, we leveraged EEG in combination with multivariate pattern analysis (MVPA)-based decoding of neural representations. In the present study, we first exposed participants to a stress or control manipulation and then asked them to complete an instrumental learning task during which they could learn S-R(-O) associations. Crucially, we used image categories as response options (R) and outcome categories (O) that have a distinct neural signature and recorded EEG throughout the task. EEG-based classifiers (support vector machines, SVMs) were trained to distinguish between the R and O stimulus categories on a separate delayed-matching-to-sample (DMS) task. We then applied these classifiers to the instrumental learning task to decode the outcome representations relevant to the goal-directed system when participants saw the stimulus S and when they made the response R. Furthermore, we decoded response R representations when participants saw the stimulus S, pointing to existing S-R representations (which may, however, be relevant for both habitual S-R learning and goal-directed S-R-O learning). Using a similar decoding approach on functional magnetic

resonance imaging (fMRI) MRI data, a previous study showed that brain regions implicated in goal-directed control contained information about outcomes and responses, whereas regions associated with habitual responding contained only information about responses (but not outcomes) at the time of stimulus presentation (*McNamee et al., 2015*).

Because previous rodent studies suggested that stress or stress hormone effects on the balance of goal-directed and habitual forms of learning are training-dependent (*Dias-Ferreira et al., 2009*; *Packard, 1999*; *Siller-Pérez et al., 2017*), we also assessed training-dependent dynamics in the stress effect on outcome and response processing. Although we did not aim to test overtraining-induced changes in the balance of outcome and response processing, for which the number of trials may have been too limited in the present study, we included transient outcome devaluations at the beginning, middle, and end of the instrumental learning task to assess whether stress effects on instrumental behaviour are training-dependent. These outcome devaluations were included in order to assess if and how the predicted changes in neural representations are linked to behavioural manifestations of stress-induced changes in behavioural control.

## Results

The goal of this study was to elucidate the mechanisms underlying the impact of stress on the control of instrumental behaviour. Specifically, we aimed to leverage an EEG-based decoding approach to determine stress-induced changes in outcome and response representations that constitute key features that distinguish goal-directed action and habitual responding (*Adams, 1982*; *Adams and Dickinson, 1981*; *Balleine and Dickinson, 1998*; *Balleine and O'Doherty, 2010*). To this end, participants first underwent the Trier Social Stress Test (TSST; *Kirschbaum et al., 1993*), a mock job interview that represents a gold standard in experimental stress research (*Allen et al., 2014*), or a non-stressful control manipulation. Afterwards, participants completed a reinforcement learning task (*Luque et al., 2017*) that allowed us to probe the goal-directed vs. habitual control of behaviour. In this task, participants could learn S-R-O associations (*Figure 1*). Specifically, they acted as 'space traders' who traded two cards, represented by two distinct fractals (green vs. pink), with two alien tribes that were represented by distinct symbols (red vs. blue) in return for cards from two possible categories (objects vs. scenes). On each trial, participants first saw one of the two fractals (S). They were then shown representatives of the two alien tribes next to each other and asked to decide which alien to offer the fractal to (R). Finally, participants received feedback about whether the alien accepted the offer and traded one of the desired cards and how many respective points were earned (O). Importantly, one alien tribe accepted only one type of fractal and traded only one card category. Furthermore, one card category was worth more than the other (high-valued outcome, $O^{high}$, and low-valued outcome, $O^{low}$). Participants had to learn these associations using trial-by-trial feedback. Moreover, there was a response cost associated with each trade that was accepted by the alien.

This task can be solved by 'goal-directed' action-outcome (S-R-O) learning and by 'habitual' S-R learning. In order to reveal the mode of control at the behavioural level, we presented task blocks in which one of the outcomes was devalued (i.e. not worth any points but associated with a response cost, thus resulting in a negative outcome). If behaviour is goal-directed, it should be sensitive to the outcome-devaluation, and participants should avoid the devalued action. Conversely, if behaviour is habitual, it should be less sensitive to the outcome devaluation, and participants should be more prone to perform the frequently repeated but now devalued response. To assess whether the balance of goal-directed and habitual behaviour and its modulation by stress is training-dependent, we presented devaluation blocks early during training, after moderate training, and again after extended training at the end of the task.

Critically, we recorded EEG during task performance and used stimulus categories as R and O stimuli that are known to have distinct neural signatures (*Bae and Luck, 2018*; *Cairney et al., 2018*; *Taghizadeh-Sarabi et al., 2015*; *Treder, 2020*). We trained EEG-based multivariate classifiers in an unrelated DMS task to discriminate between stimulus categories that were used as S (blue vs. red symbols that differed also in shape and line orientation) and O (objects vs. scenes) (*Figure 1*). The DMS task was completed both before and after the reinforcement learning task, and the classifier was trained on the pooled trials of the two DMS sessions (thus ruling out any time-dependent biases of the classifiers). The trained classifiers were then applied to the reinforcement learning task to determine neural representations of R and O.

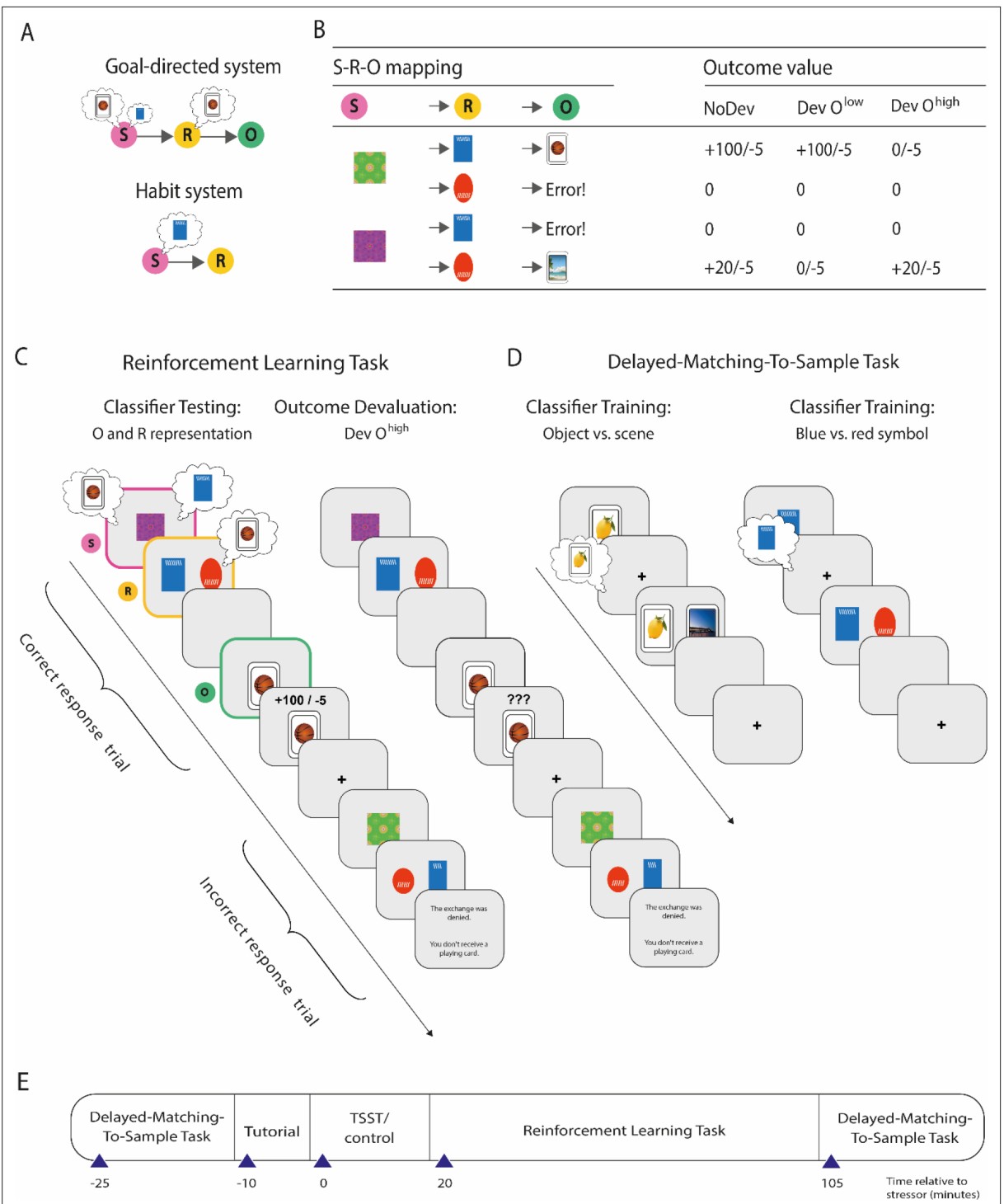

**Figure 1.** Overview of the paradigm used to decode outcome and response representations.

(**A**) Illustration of the goal-directed and habit system. While the goal-directed system encodes associations between stimulus (S), response (R), and outcome (O), the habit system acquires S-R associations independent of the outcome engendered by the response. In accordance with this, the goal-directed system relies on outcome representations, whereas the habitual system does not. In contrast, response representations during stimulus presentation may be relevant for both habitual S-R and goal-directed S-R-O processing. (**B**) S-R-O mappings in the reinforcement learning task and outcomes in trials in which either none of the possible outcomes were devalued (NoDev), the outcome with lower value was devalued (Dev O^low), or the outcome with the higher value was devalued (Dev O^high). (**C**) Schematic representation of the reinforcement learning task in which participants were trained on S-R-O sequences in a trial-by-trial manner. Using an EEG-based support vector machine (SVM), neural representations of the outcome stimuli (object vs. scene) were decoded during stimulus presentation and during response choice. Moreover, neural representations of the response options

*Figure 1 continued on next page*

*Figure 1 continued*

(blue vs. red alien) were decoded during stimulus presentation. During devaluation blocks, participants saw on the last screen of each trial '???' instead of the outcome value. (**D**) The SVM was trained in an unrelated delayed-matching-to-sample task (maintenance phase) that required participants to keep stimuli in mind that belonged to categories used as outcomes or response options during the reinforcement learning task. (**E**) Timeline of the experiment.

Response representations were decoded during the S presentation, whereas O representations were decoded during both the S presentation and participants' choice (R). Outcome representations at the time of S presentation and R are indicative of goal-directed control. In contrast, response representations at the time of stimulus-representations may be relevant for both goal-directed S-R-O learning and habitual S-R learning.

In order to adequately assess whether increased responding to devalued actions is due to reduced goal-directed or enhanced habitual control, or both, we employed a variety of measures and statistical approaches. At the behavioural level, we applied a devaluation paradigm to examine whether stressed participants respond differently to transient outcome devaluations compared to control participants and to assess whether the predicted neural changes are linked to behavioural outcomes. Behavioural analyses focussed mainly on participants' choice in valued and devalued trials. Based on previous animal studies (*Dias-Ferreira et al., 2009*), we additionally investigated whether the effects of stress on the mode of control would be training-dependent and therefore implemented outcome devaluation at the beginning, middle, and end of the instrumental task. At the physiological level, we analysed pulse, systolic and diastolic blood pressure as well as salivary cortisol concentrations at different time points during the experiment to assess for the effectiveness of the stress manipulation. At the neural level, we mainly focused on the decoding of outcome and response representations, which provide insights into the mechanisms through which stress affects sensitivity to outcome devaluation - the primary objective of the present study.

To further address the important question of whether response and outcome representations reflect signatures of distinct control systems, we additionally analysed the correlations between both neural representations. For this purpose, we used Bayesian correlational analyses. Depending on the magnitude of the Bayes factor (reflecting the likelihood ratio of the data under the alternative hypothesis and the data under the null hypothesis), the Bayesian approach can provide evidence in favour of the alternative hypothesis or evidence in favour of the null (*Hoijtink, 2012*; *Kass and Raftery, 1995*; *Nuzzo, 2017*). Thus, we utilized Bayesian analyses to provide clear evidence for or against the null hypothesis. To further assess the association between the behavioural data (i.e. classification accuracy) and the strength of the neural representation (i.e. classification accuracies), we computed Spearman correlations. In order to also analyse previously proposed 'attentional habits' (*Luque et al., 2017*), we analysed event-related potentials (ERPs).

## Successful stress manipulation

Significant subjective and physiological changes in response to the TSST confirmed that stress was successfully induced. Compared to participants in the control group, participants exposed to the TSST experienced the treatment as significantly more stressful, difficult, and unpleasant than those in the control condition (all $t_{56}$ >5.82, all p<0.001, all $d$>1.530, and all 95% CI=1.456–2.112; *Table 1*).

**Table 1.** Subjective responses to the Trier Social Stress Test (TSST) or control manipulation.

| | Control | | Stress | |
|---|---|---|---|---|
| | M | SEM | M | SEM |
| Subjective assessments | | | | |
| Stressfulness | 16.79 | 3.45 | 62.33* | 4.44 |
| Unpleasantness | 15.36 | 3.51 | 57.00* | 5.90 |
| Difficulty | 14.29 | 3.35 | 51.33* | 5.29 |

Subjective assessments were rated on a scale from 0 ('not at all') to 100 ('very much'). *p<0.001, Bonferroni-corrected, significant group difference.

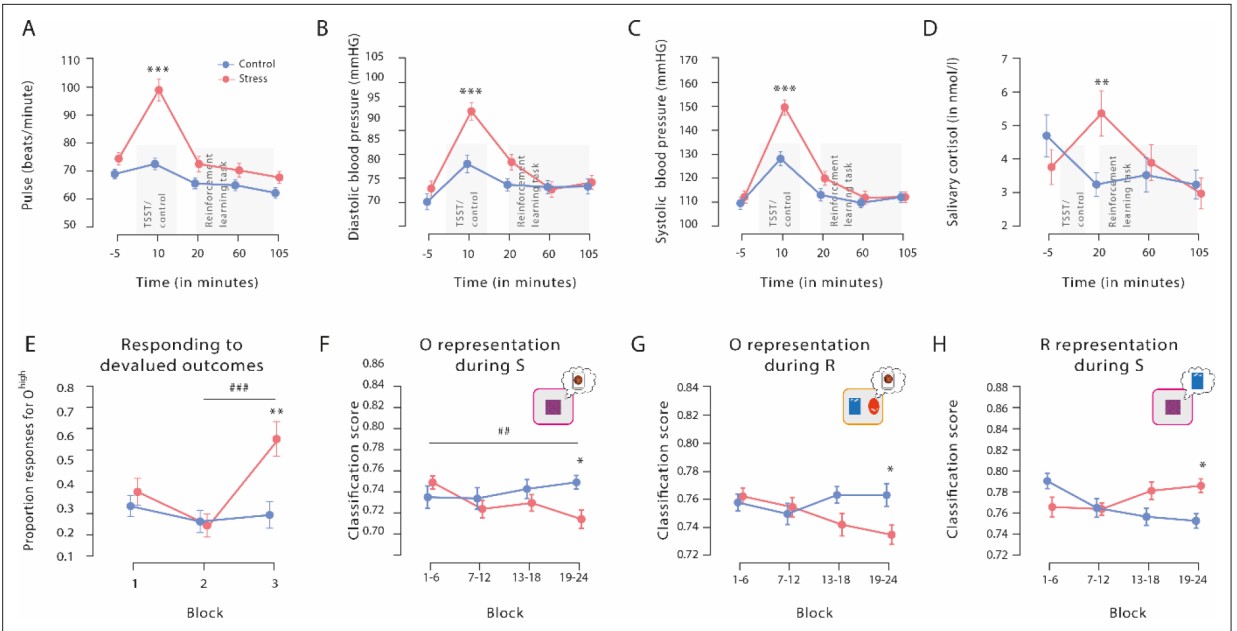

**Figure 2.** Physiological responses to the Trier Social Stress Test (TSST), proportion of responses for devalued outcomes, and outcome and response representations throughout the reinforcement learning task. The exposure to the TSST, but not to the control manipulation, resulted in a significant increase in pulse (**A**), diastolic blood pressure (**B**), systolic blood pressure (**C**), and salivary cortisol (**D**). The grey bars denote the timing and duration of the treatment (TSST vs. control condition) and the respective reinforcement learning task. (**E**) Proportion of responses for devalued outcomes across the reinforcement learning task during Dev $O^{high}$ blocks. As training proceeded, stressed participants increasingly selected actions that led to a devalued outcome. In addition, stressed participants responded significantly more often to the devalued action than non-stressed controls in the third devaluation block at the end of the task. Individual data points are shown in *Figure 3*. The data for Dev $O^{low}$ and NoDev blocks is presented in *Figure 3—figure supplement 1* and *Figure 3—figure supplement 2*, respectively. (**F** and **G**) Outcome representation during stimulus presentation and response choice. As training proceeded, the outcome representations decreased in the stress group, while there were no changes in the control group (blocks 1–4 vs. blocks 19–24). At the end of the learning task, outcome representations were significantly lower in stressed participants than in controls. (**H**) Response representations during stimulus presentation. Stressed participants showed significantly stronger response representations after extended training compared to the control group. Data represents means and error bars represent the SE of the mean. \*\*\* p<0.001, \*\* p<0.01, and \* p<0.05, Bonferroni-corrected (group differences, corrected for all time points and blocks, respectively). ### p<0.001, ## p<0.01, Bonferroni-corrected (block differences, corrected for the number of blocks).

The online version of this article includes the following figure supplement(s) for figure 2:

**Figure supplement 1.** Event-related potentials during Dev $O^{high}$ blocks.

**Figure supplement 2.** Event-related potentials during Dev $O^{low}$ blocks.

**Figure supplement 3.** Event-related potentials during NoDev blocks.

**Figure supplement 4.** Brain areas contributing the most to outcome and response decoding.

At the physiological level, exposure to the TSST elicited significant increases in pulse and systolic and diastolic blood pressure (time point of measurement × group interaction, all $F_{[4, 224]}$>13.55, all p<0.001, all $\eta_p^2$>0.195, and all 95% CI=0.100–0.425; *Figure 2*). As shown in *Figure 2A–C*, although both groups had comparable blood pressure and pulse before and after the TSST (all $t_{56}$ <2.28, all $p_{corr}$>0.081, all d<0.600, and all 95% CI=0.069–0.702), participants in the stress group had significantly higher blood pressure and pulse during the experimental manipulation than those in the control group (all $t_{56}$ >4.21, all $p_{corr}$<0.001, all d>1.107, and all 95% CI=0.549–2.098). Finally, salivary cortisol concentrations increased in response to the TSST but not after the control manipulation (time point of measurement × group interaction: $F_{[3, 168]}$=6.69, p<0.001, $\eta_p^2$=0.107, and 95% CI=0.026–0.188). As shown in *Figure 2D*, participants in the stress and control groups had comparable cortisol concentrations at baseline ($t_{56}$=1.16, $p_{corr}$=1, d=0.304, and 95% CI=−0.216–0.821). However, about 20 min after the treatment, when the reinforcement learning task started, cortisol levels were significantly higher in the stress group than in the control group ($t_{56}$=2.74, $p_{corr}$ = 0.032, d=0.720, and 95% CI=0.185–1.249). As expected, cortisol levels returned to the level of the control group by the end of the task

(60 min: $t_{56}$=0.50, $p_{corr}$ = 1, $d$=0.130, and 95% CI=−0.386–0.645; 105 min: $t_{56}$=0.42, $p_{corr}$=1, $d$=0.111, and 95% CI=−0.405–0.625).

## Stress renders behaviour less sensitive to outcome devaluation

Participants' choice accuracy increased significantly throughout the task ($F$[2, 114] = 10.08, p<0.001, $\eta_p^2$=0.150, and 95% CI=0.042–0.261) and reached an average performance of 99% correct responses in blocks without devaluation (NoDev), indicating that participants learned the task very well. Both groups reached a performance plateau relatively quickly and at about the same time (*Figure 3—figure supplement 2*). Performance in NoDev blocks did further not differ between the control and stress groups, and the time course of learning was comparable in the two groups (time × group interaction: $F$[2, 112] = 2.44, p=0.092, $\eta_p^2$=0.042, and 95% CI=0–0.123; main effect group: $F$[1, 56] = 0.30, p=0.585, $\eta_p^2$=0.005, and 95% CI=0–0.096), suggesting that stress did not affect instrumental learning as such. Furthermore, during NoDev blocks, participants had a higher response accuracy for $S^{high}$ than $S^{low}$ trials ($t_{57}$=3.29, p=0.002, $d$=0.432, and 95% CI=0.161–0.699), suggesting that learning was modulated by the value of the outcome. In the control group, instrumental behaviour did not differ across the different devaluation blocks ($F$[2, 54] = 1.466, p=0.240, $\eta_p^2$=0.052, and 95% CI=0.013–0.049), indicating that the repeated devaluation phases as such did not result in increased sensitivity to the devaluation procedure.

During the reinforcement learning blocks, in which one of the outcomes was devalued (Dev), participants chose the action that was associated with a valued outcome significantly more often than the action that was associated with a devalued outcome (outcome devaluation × stimulus value interaction: $F$[2, 110] = 163.31, p<0.001, $\eta_p^2$=0.785, and 95% CI=0.664–0.789; valued vs. devalued during Dev $O^{high}$: $t_{57}$=12.17, $p_{corr}$<0.001, $d$=1.589, and 95% CI=1.205–1.985; valued vs. devalued during Dev $O^{low}$: $t_{56}$=13.49, $p_{corr}$<0.001, $d$=1.786, and 95% CI=1.363–2.203), providing further evidence of successful instrumental learning. The more pronounced valued vs. devalued difference for $O^{low}$ (i.e. outcome devaluation × stimulus value interaction) persisted even when we analysed only the control group (outcome devaluation × stimulus value interaction: $F$[2, 52] = 70.601, p<0.001, $\eta_p^2$=0.731, and 95% CI=0.391–0.720; valued vs. devalued during Dev $O^{high}$: $t_{27}$=8.482, $p_{corr}$<0.001, $d$=1.603, and 95% CI=1.032–2.16; valued vs. devalued during Dev $O^{low}$: $t_{26}$=8.654, $p_{corr}$ <0.001, $d$=1.665, and 95% CI=1.071–2.246). Importantly, however, there was also a significant outcome devaluation × stimulus value × block × group interaction ($F$[4, 220]=4.86, p<0.001, $\eta_p^2$=0.081, and 95% CI=0.016–0.143). Follow-up ANOVAs revealed that stressed participants increasingly selected actions that led to a devalued outcome during Dev $O^{high}$ blocks at the end of the task (stimulus value × time × group interaction: $F$[2, 112] = 8.89, p<0.001, $\eta_p^2$=0.137, and 95% CI=0.033–0.247; time × group interaction for devalued stimuli: $F$[2, 112] = 9.09, p<0.001, $\eta_p^2$=0.140, and 95% CI=0.035–0.250; time × group interaction for valued stimuli: $F$[2, 112] = 2.19, p=0.116, $\eta_p^2$=0.038, and 95% CI=0–0.116; block 2 vs. block 3 for devalued stimuli in the stress group: $t_{29}$=4.28, $p_{corr}$<0.001, $d$=0.781, and 95% CI=0.366–1.186; block 2 vs. block 3 for devalued stimuli in the control group: $t_{27}$=1.10, $p_{corr}$=1, $d$=0.207, and 95% CI=−0.169–0.580), whereas there was no such effect during Dev $O^{low}$ blocks ($F$[2, 110] = 2.39, p=0.097, $\eta_p^2$=0.042, and 95% CI=0–0.124), or during NoDev blocks ($F$[2, 112] = 0.41, p=0.667, $\eta_p^2$=0.007, and 95% CI=0–0.052).

As shown in *Figure 2E* and *Figure 3*, stressed participants responded significantly more often to the devalued action than the non-stressed controls did in the third devaluation block at the end of the task ($t_{56}$=2.61, $p_{corr}$=0.036, $d$=0.685, 95% CI=0.152–1.213, stress vs. control during the first and second Dev $O^{high}$ blocks: all $t_{56}$ <0.57, all $p_{corr}$ = 1, all $d$<0.149, all 95% CI=−0.472–0.664). Furthermore, while there was no evidence of an interaction of devaluation block number (1 vs. 2) and experimental treatment ($F$[1, 56] = 1.575, p=0.215, $\eta_p^2$=0.027, and 95% CI=−0.013–0.052) when analysing the changes from the first to the second devaluation block, we obtained a significant interaction between block (2 vs. 3) and treatment when analysing the changes from block 2 to block 3 ($F$[1, 56] = 13.589, p<0.001, $\eta_p^2$=0.195, and 95% CI=0.105–0.319). Moreover, follow-up tests revealed that in block 2, groups did not differ in responses for the devalued outcome ($t_{56}$=0.165, p=0.870, Cohen's $d$=0.043,−0.472 to 0.558).

In summary, this data suggests that stress rendered behaviour less sensitive to the outcome devaluation in a training-dependent manner, which was particularly the case for the frequently repeated response to $O^{high}$.

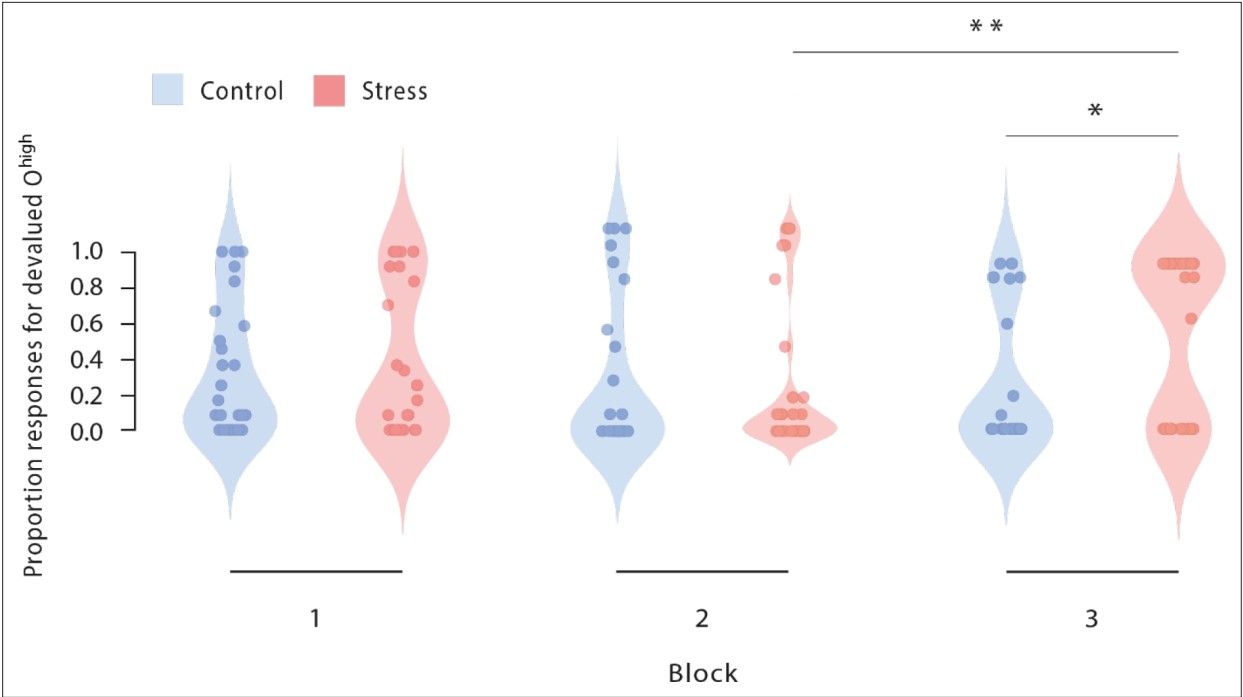

**Figure 3.** Proportion of responses for devalued outcomes across the reinforcement learning task during Dev O$^{high}$ blocks. As training proceeded, stressed participants increasingly selected those actions that led to a devalued outcome (block 2 vs. block 3). In addition, stressed participants responded significantly more often to the devalued action than non-stressed controls in the third devaluation block at the end of the task (stress vs. control). Dots represent mean performance of individual participants. The data for Dev O$^{low}$ and NoDev blocks is presented in *Figure 3—figure supplement 1* and *Figure 3—figure supplement 2*. * p<0.001, Bonferroni-corrected (stress vs. control). # p<0.001, Bonferroni-corrected (vs. the respective other block).

The online version of this article includes the following figure supplement(s) for figure 3:

**Figure supplement 1.** Proportion of responses for devalued outcome across the reinforcement learning task during Dev O$^{low}$ blocks.

**Figure supplement 2.** Proportion of correct responses during NoDev blocks after low, moderate, and high training intensity.

Given that it is well-known that stress may disrupt memory retrieval (*Gagnon and Wagner, 2016*; *de Quervain et al., 1998*; *Roozendaal, 2002*), it might be argued that the increased responding to the devalued action in stressed participants was due to their difficulty in remembering which outcome was valued and which was devalued. We could, however, rule out this alternative. Each block of our task involved, in addition to reinforcement learning trials, so called consumption trials in which participants could freely choose between the two outcome categories, without any response costs. These trials served to assess whether participants were aware of the current value of the outcomes. Here, participants clearly preferred the high-valued cards over low-valued cards during NoDev blocks ($F[1, 56] = 5382.91$, p<0.001, $\eta_p^2=0.990$, and 95% CI=0.984–0.992) as well as the valued card over its devalued counterpart during Dev blocks (Dev O$^{high}$ and Dev O$^{low}$: both $F[1, 56] > 214.72$, both p<0.001, both $\eta_p^2<0.793$, and both 95% CI=0.687–0.848; outcome devaluation × stimulus value interaction: $F[2, 112] = 876.42$, p<0.001, $\eta_p^2=0.940$, and 95% CI=0.919–0.952), irrespective of stress (outcome devaluation × stimulus value × group interaction: $F[2, 112] = 0.05$, p=0.956, $\eta_p^2=0.001$, and 95% CI=0–0.120). This finding demonstrates that participants in both groups were aware of the value of the card stimuli in a specific block but that responses of stressed participants were less guided by this knowledge about the value of the outcome engendered by the response.

Mean reaction times were significantly faster for high-valued stimuli than for low-valued stimuli during NoDev blocks (NoDev: $t_{57}=2.83$, $p_{corr} = 0.019$, $d=0.372$, and 95% CI=0.104–0.637), whereas participants responded faster to low-valued stimuli than to high-valued stimuli during Dev blocks (Dev O$^{low}$: $t_{56}=5.73$, $p_{corr}<0.001$, $d=0.759$, and 95% CI=0.462–1.052; Dev O$^{high}$: $t_{57}=4.41$, $p_{corr}<0.001$, and $d=0.579$, 95% CI=0.298–0.855; outcome devaluation × stimulus value interaction: $F[2, 110] = 28.14$, p<0.001, $\eta_p^2=0.338$, and 95% CI=0.194–0.451). Stress did not influence participants' reaction times

(outcome devaluation × stimulus value × time × group: $F[4, 220] = 0.85$, p=0.494, $\eta_p^2=0.015$, and 95% CI=0–0.043; main effect group: $F[1, 55] = 1.00$, p=0.322, $\eta_p^2=0.018$, 95% CI=0–0.134).

Although our analyses of the neural data focussed mainly on the decoding of outcome and response representations, there is recent evidence suggesting that habitual and goal-directed processes might also be reflected in ERPs (*Luque et al., 2017*). The extent to which a reward related ERP is sensitive to an outcome devaluation is assumed to indicate the degree of habitual or goal-directed processing. Thus, we additionally analysed stress effects on related ERPs depending on outcome devaluation. Our data shows that the occipital stimulus-locked P1 component was insensitive to outcome devaluation (outcome devaluation × stimulus value interaction: $F[2, 102]=0.63$, p=0.536, $\eta_p^2=0.012$, and 95% CI=0–0.069; *Figure 2—figure supplements 1–3*), which might suggest the formation of an 'attentional habit' (*Luque et al., 2017*). However, the P1 was not modulated by reward value ($F[1, 51] = 0.25$, p=0.619, $\eta_p^2=0.005$, and 95% CI=0.002–0.009), which makes the interpretation of the insensitivity to the outcome devaluation difficult. The P1 component was also not modulated by stress (stimulus value × group: $F[1, 51] = 0.13$, p=0.723, $\eta_p^2=0.002$, and 95% CI=0–0.086; outcome devaluation × stimulus value × group: $F[2, 102] = 0.11$, p=0.900, $\eta_p^2=0.002$, and 95% CI=0–0.028). Moreover, we identified a late component that showed a non-significant trend towards sensitivity to the outcome devaluation during Dev O^high blocks in control participants (devalued vs. valued: $t_{24}=1.91$, p=0.068, $d=0.382$, and 95% CI=−0.028–0.785) but not in stressed participants (devalued vs. valued: $t_{27}=1.57$, p=0.127, $d=0.297$, and 95% CI=−0.084–0.673; outcome devaluation × stimulus value × group interaction: $F[2, 102] = 5.20$, $p_{corr}=0.042$, $\eta_p^2=0.093$, and 95% CI=0.008–0.199; stimulus value × group interaction: $F[1, 51] = 6.05$, p=0.017, $\eta_p^2=0.106$, and 95% CI=0.003–0.273; no such effect in NoDev and Dev O^low blocks: stimulus value × group interaction: both $F[1, 51] < 1.44$, both p>0.236, both $\eta_p^2<0.027$, and both 95% CI=0–0.159). This pattern of results suggests that stress interferes with a late ERP component that has been linked to goal-directed processing as it was particularly sensitive to the value of an outcome. However, similar to the behavioural response to a devalued action, the stress-induced decrease in the 'outcome-sensitive' ERP component leaves the question open as to whether this stress effect is due to changes in outcome or response processing.

## Stress reduces outcome representations at the end of training

Thus far, our behavioural data showed that stress rendered behaviour less sensitive to a change in outcome value, which can be interpreted as decreased goal-directed or increased habitual behaviour.

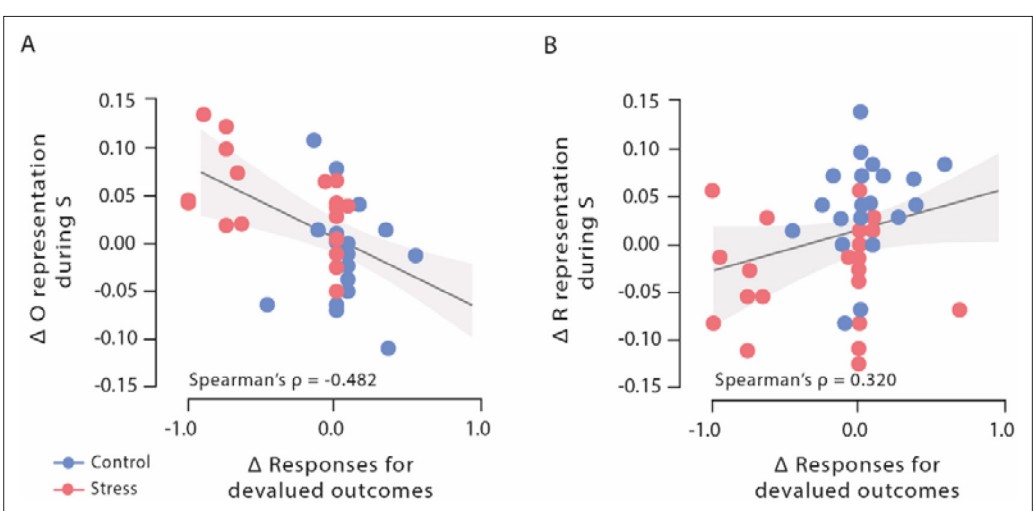

**Figure 4.** Correlations of outcome and response representation during stimulus presentation with responses for devalued outcomes during Dev O^high blocks. (**A**) Decrease of outcome representation during stimulus presentation was significantly correlated with the reduced behavioural sensitivity to the outcome devaluation during Dev O^high blocks. (**B**) Increase in response representation was significantly correlated with an increase in response for devalued outcomes during Dev O^high blocks. Higher difference scores indicate higher decreases in outcome and response representation over time. Regression lines are added for visualization purpose, and the light-coloured background areas indicate its 95% CI.

In addition, stress reduced electrophysiological late latency potentials that appeared to be sensitive to the value of an outcome (**Luque et al., 2017**).

In a next step, we leveraged an EEG-based decoding approach to address the primary objective of this study, that is, to probe the effect of stress on neural outcome and response representations. We trained an MVPA classifier (SVM) based on an independent dataset (DMS task, for details see Materials and methods) to discriminate between categories that were used during the reinforcement learning task as an outcome (object card vs. scene card). This classifier was then used to assess trial-by-trial changes in outcome representations throughout the reinforcement learning task. Goal-directed control should be reflected in high accuracy of outcome category classification during the presentation of the stimulus (S) as well as during the response choice (R).

We first analysed outcome representations during the presentation of the fractal stimulus S before participants had to make a choice. This analysis revealed a significant block × group interaction ($F$[3, 117] = 2.77, p=0.045, $\eta_p^2$=0.066, and 95% CI=0–0.149). As training proceeded, participants in the stress group showed a reduced outcome representation at the time of S presentation (first six vs. last six blocks: $t_{22}$=3.59, $p_{corr}$=0.004, $d$=0.748, and 95% CI=0.277–1.206), whereas the outcome representation remained rather constant in participants of the control group (first six vs. last six blocks: $t_{17}$=1.08, $p_{corr}$ = 0.590, $d$=0.255, and 95% CI=−0.219–0.721; **Figure 2F**). In the last six blocks of the task, the classification accuracy for the outcome was significantly lower in the stress group compared to the control group ($t_{39}$=3.13, $p_{corr}$=0.012, $d$=0.986, and 95% CI=0.326–1.635; lower training intensity [first 18 blocks of the task]: all $t_{39}$<1.13, all $p_{corr}$=1, all $d$<0.355, and all 95% CI=−0.914–0.975; note that the overall pattern remains when trials are blocked differently, see **Supplementary file 1A**). Strikingly, the reduced outcome representation was significantly correlated with the reduced behavioural sensitivity to the outcome devaluation during Dev O$^{high}$ blocks (Spearman's $\rho$ =−0.482, 95% CI=−0.688−0.205, and p=0.001; **Figure 4**).

Analysing the outcome representations at the time of choice between the two aliens revealed a very similar pattern; at the end of the task, participants in the stress group showed a decreased outcome representation at the time point of the choice ($t_{22}$=2.94, $p_{corr}$=0.016, $d$=0.613, and 95% CI=0.161–1.054), whereas there was no such effect in the control group ($t_{17}$=0.59, $p_{corr}$=1, $d$=0.138, and 95% CI=−0.328–0.600). In addition, stressed participants had reduced outcome representations relative to controls, reflected in a significantly reduced classification accuracy, during the response choice at the end of the reinforcement learning task (block × group interaction: $F$[3, 117] = 2.99, p=0.034, $\eta_p^2$=0.071, and 95% CI=0–0.156; stress vs. control, high training intensity: $t_{39}$=2.75, $p_{corr}$=0.036, $d$=0.865, and 95% CI=0.214–1.506; lower training intensities: all $t_{39}$<2.30, all $p_{corr}$>0.108, all $d$<0.725, and all 95% CI=−0.500–1.358; **Figure 2G**). Together, these results show that at the end of training, acute stress reduced the representation of action outcomes that are considered to be a hallmark of goal-directed control.

## Stress boosts response representations at the end of training

While it is assumed that the outcome representation that is crucial for goal-directed S-R-O learning is reduced with increasing habitual behaviour control, response (R) representations at the time of stimulus (S) presentation may be involved in both goal-directed S-R-O and habitual S-R processing. Therefore, we trained another classifier to discriminate between categories that were used as response options during the reinforcement learning task (red vs. blue alien). This classifier was used to examine changes in response representation during the stimulus presentation (S) throughout the reinforcement learning task. For these response representations, a block × group ANOVA revealed a significant interaction effect ($F$[3, 147] = 5.82, p<0.001, $\eta_p^2$=0.106, and 95% CI=0.021–0.192). As shown in **Figure 2H**, participants in the stress group showed a stronger response representation that was reflected in higher classification accuracy for the response categories with increasing training intensity (first half vs. last half: $t_{25}$=2.51, $p_{corr}$=0.038, $d$=0.491, and 95% CI=0.079–0.894), whereas there was even a decrease in the control group (first half vs. last half: $t_{24}$=3.50, $p_{corr}$=0.004, $d$=0.701, and 95% CI=0.256–1.134). In the last six blocks of the reinforcement learning task, stressed participants had significantly higher response representations than participants in the control group ($t_{49}$=2.75, $p_{corr}$=0.032, $d$=0.770, and 95% CI=0.197–1.336; lower training intensities: all $t_{49}$<1.92, all $p_{corr}$>0.244, all $d$<0.537, and all 95% CI=0.025–1.094). Interestingly, this increase in response representations was significantly correlated with an increase in responses for devalued outcomes during Dev O$^{high}$ blocks

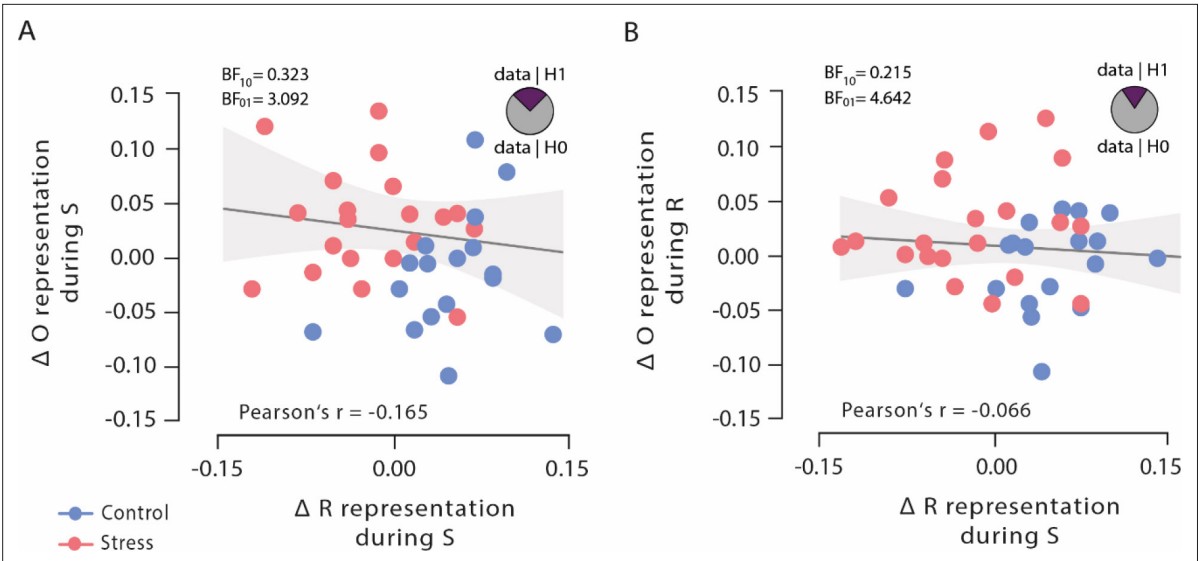

**Figure 5.** Bayesian correlations of outcome representation during stimulus presentation and response selection with response representation during stimulus presentation. (**A**) Outcome representation during stimulus presentation was not correlated with response representation during response selection. As visualized in the pie chart, the corresponding Bayes factor suggests that the observed data are 3.092 times more likely under the null hypothesis (H0) than under the alternative hypothesis (H1). (**B**) Outcome representation during response selection was not correlated with response representation during stimulus presentation. As visualized in the pie chart, the corresponding Bayes factor suggests that the observed data are 4.642 times more likely under the H0 than under the H1. Higher difference scores indicate larger decreases in outcome and response representation, respectively, over time. Regression lines are added for visualization purpose, and the light-coloured background areas indicate its 95% CI.

(Spearman's $\rho$ =0.320, 95% CI=0.049–0.547, and p=0.022; *Figure 4*). Thus, our MVPA results indicate that stress leads to an increased response representation at the time of stimulus presentation.

Importantly, when we grouped the classification data not in four blocks consisting of 144 trials in total (averaged over six successive blocks containing 24 reinforcement learning trials each) but in 2, 3, 6, or 12 blocks, the pattern of results for the neural outcome and response representations was largely comparable (*Supplementary file 1A*).

## Outcome and response representations are uncorrelated

To test whether the observed opposing changes in outcome and response representations after stress reflected independent or linked neural representations, we analysed Bayesian correlation between the classification accuracies in order to explicitly test the evidence in favour of the null and alternative hypothesis, respectively. These analyses revealed moderate evidence for the null hypothesis that outcome representations, both at stimulus presentation and response selection, were uncorrelated with response representation at choice time (both Pearson's $|r|$<0.165, both 95% CI=0.154–0.442, and both $BF_{01}$ >3.092; *Figure 5*), suggesting that outcome representations and response representations may be independent of each other.

## Control variables and performance in the DMS task

At the beginning of the experiment, stress and control groups did not differ in subjective mood (all $t_{56}$<1.20, all p>0.235, all $d$=0.316, and all 95% CI=−0.326–0.833), subjective chronic stress (all $t_{54}$<1.07, all p>0.290, all $d$<0.285, and all 95% CI=−0.677–0.811), depressive mood ($t_{56}$=1.07, p=0.289, $d$=0.281, and 95% CI=−0.238–0.797), state, or trait anxiety (both $t_{56}$<0.44, all p>0.663, both $d$<0.115, and both 95% CI=0.401–0.630; *Supplementary file 1C*). Behavioural performance in the DMS task, used to train the classifier, was, as expected, very high (average performance: 97.5% correct, SD = 0.054) and comparable between groups ($t_{56}$=0.23, p=0.818, $d$=0.061, and 95% CI=−0.455–0.576). The average classification accuracy of the classifier was 72% (SD = 0.068) for the response categories (blue rectangular vs. red oval symbol) and 66% (SD = 0.046) for the outcome categories (object vs. scene image) and did not differ between the stress and control groups (both $t_{51}$<0.89, both p>0.376, both $d$<0.246, and both 95% CI=−0.669–0.786).

Furthermore, we recorded eye-tracking data to control for potential group differences in saccades or eye blinks. These control analyses showed that there were no significant group differences in saccades or eye blinks across the task or trial type (outcome devaluation × stimulus value × time × group: $F$[4, 196] = 0.78, p=0.54, $\eta_p^2$=0.02, and 95% CI=0.002–0.008; outcome devaluation × stimulus value × group: $F$[2, 98] = 1.03, p=0.36, $\eta_p^2$=0.02, and 95% CI=0.005–0.020; see *Supplementary file 1D*).

## Discussion

Previous research showed that stress favours habitual responding over goal-directed action (*Braun and Hauber, 2013*; *Dias-Ferreira et al., 2009*; *Gourley et al., 2012*; *Schwabe et al., 2009*; *Schwabe et al., 2010*; *Schwabe et al., 2011b*; *Schwabe et al., 2012*; *Seehagen et al., 2015*; *Smeets et al., 2019*; *Smeets and Quaedflieg, 2019*; *Soares et al., 2012*; *Wirz et al., 2018*). Although this stress-induced bias towards habitual behaviour has important implications for stress-related mental disorders (*Adams et al., 2018*; *Goeders, 2004*; *Schwabe et al., 2011a*), a fundamental question has remained elusive thus far: is the shift towards habitual behaviour under stress due to diminished goal-directed control or enhanced habitual control, or both? Canonical behavioural assays of the mode of behavioural control cannot distinguish between these alternatives. Here, we used EEG-based decoding of outcome and response representations - the key components of goal-directed and habitual processing - to provide evidence that acute stress results in a decrease of outcome-related processing that is critical for goal-directed control, and paralleled by an increase in response processing.

Our behavioural and ERP data corroborates previous reports of a stress-induced shift from goal-directed to habitual control (*Braun and Hauber, 2013*; *Schwabe et al., 2011b*; *Smeets and Quaedflieg, 2019*). Specifically, stressed participants showed increased responding to a devalued action after extended training, suggesting a reduced behavioural sensitivity to the outcome devaluation which indicates less goal-directed behaviour and more habitual responding (*Adams and Dickinson, 1981*). Recent evidence showed that goal-directed and habitual processing may also be reflected in ERPs that are either sensitive or insensitive to changes in the value of an action outcome (*Luque et al., 2017*). We observed here that stress appears to reduce the sensitivity of late potentials to the outcome devaluation, whereas stress left the occipital P1 component that was insensitive to the outcome devaluation unaffected and may thus be considered habitual (*Luque et al., 2017*). However, similar to the behavioural insensitivity to an outcome devaluation after stress, these ERPs cannot separate reduced goal-directed from increased habitual responding. To disentangle outcome-related processing that is critical for goal-directed learning and S-R processing, which may be relevant both for goal-directed and habitual learning, we used an MVPA-based decoding approach. Critically, our Bayesian analyses indicated that changes in outcome and response representations were uncorrelated, which may be taken as evidence that they do not reflect changes in a single system, but rather (at least partly) dissociable signatures of goal-directed and habitual processing.

We show that stress led to a reduction in outcome representations both at the time of stimulus presentation and at the time of action selection, as well as a parallel increase of response representations during stimulus presentation indicative of enhanced S-R associations (*Figure 2F–H*). Both the stress-induced reduction in outcome representations and the increase in response representations were directly correlated with the behavioural (in)sensitivity to the outcome devaluation (*Figure 4*). However, while outcome representations were negatively correlated with participants' responding to the devalued action, there was a positive correlation for response representations which might lend further support to the view that these representations reflect distinct processes. Taken together, our results indicate that acute stress leads to enhanced S-R processing and impaired outcome-related processing. The latter is in line with evidence showing reduced orbitofrontal activity in the face of elevated stress hormones (*Schwabe et al., 2012*), which is accompanied by increased habitual processing. Based on previous pharmacological studies (*Gourley et al., 2012*; *Schwabe et al., 2010*; *Schwabe et al., 2011b*; *Schwabe et al., 2012*), we assume that these opposing effects of stress on outcome-related processing and S-R processing are based on the concerted action of glucocorticoids and noradrenaline on the neural substrates of goal-directed and habitual control.

Importantly, at the time when stress effects on devaluation sensitivity and outcome representation were observed, stress mediators were no longer elevated. However, it is important to note that

stress effects do not necessarily terminate when acute levels of stress mediators returned to baseline. Specifically, several major stress mediators are known to have 'after-effects' that outlast acute elevations of these measures (*Joëls and Baram, 2009*). For example, glucocorticoids are assumed to act as a 'two-stage rocket', with rapid, non-genomic actions and delayed, genomic actions (*Joëls et al., 2012*). The latter genomic actions typically set in after acute glucocorticoid elevations vanished (*de Kloet et al., 2008*). Thus, the fact that autonomic measures and cortisol levels returned to baseline during the learning task does not imply that the stress system activation had been over at this point. Moreover, acutely elevated stress mediators may have affected early learning processes in a way that became apparent only as training progressed.

Importantly, both our behavioural and our neural decoding data showed that stress affected the balance of outcome-related and S-R processes in a training-dependent manner. Previous human studies could not distinguish between such early and late effects of stress because the mode of behavioural control was only assessed at the end of training (*Schwabe and Wolf, 2009*; *Smeets et al., 2019*). The present results compliment rodent data which shows that (chronic) stress effects on the control of instrumental learning are training-dependent (*Dias-Ferreira et al., 2009*). In general, it is commonly assumed that early training falls under goal-directed control, while extensive training results in habitual control (*Adams, 1982*; *Dickinson et al., 1995*; *Tricomi et al., 2009*). However, whether or not overtraining may induce habitual behaviour in humans is currently debated (*de Wit et al., 2018*), and our data cannot speak to this issue as training may have been too limited to result in overtraining-related habits (which might require thousands of responses; *Tricomi et al., 2009*). Thus, training-dependent effects do not necessarily imply overtraining effects. However, findings of 'cognitive' and 'habitual' forms of navigational learning in rats demonstrated that stress hormones may accelerate a shift from 'cognitive' to 'habitual' learning that would otherwise only occur after extended training (*Packard, 1999*; *Siller-Pérez et al., 2017*). Thus, it is tempting to hypothesize that a similar mechanism might be at work during instrumental learning in stressed humans. This conclusion, however, remains speculative as we did not observe a training-dependent shift towards habitual control in the control group, and this group even showed reduced response and increased outcome representations over time, which rather suggests increased goal-directed processing across the task. Future studies are required to test this hypothesis by using the present neural decoding approach in combination with an extensive training protocol and groups of participants that are exposed to stress at distinct stages of the learning process.

In animal studies, devaluation sensitivity is usually assessed by means of a reinforcement test, in which the devalued outcome is delivered contingent on the relevant response. This procedure allows to control, for example, for a general lack of attention. In the present human study, the feedback provided on each trial varied between blocks. During NoDev blocks, participants saw the value of the card that was gained before as well as the corresponding response costs. In Dev blocks, however, this information was masked for all trials. This procedure is comparable to previous studies in humans in which the devalued outcome was not presented during critical test trials (e.g. *Schwabe and Wolf, 2009*; *Valentin et al., 2007*). Importantly, however, we also included consumption trials in all of the blocks which enabled us to rule out unspecific factors, such as a general lack of attention, altered contingency knowledge, or response perseveration. In these consumption trials, participants clearly preferred the high-valued cards over low-valued cards during NoDev blocks as well as the valued card over its devalued counterpart during Dev blocks, irrespective of stress. This finding demonstrates that participants in both groups were aware of the value of the card stimuli in a specific block but that responses of stressed participants were less guided by this knowledge about the value of the outcome engendered by the response. In addition, the specific response pattern in the consumption trials during devalued blocks also rules out general attentional or motivational deficits.

It is important to note that participants received an error feedback in devalued trials when they chose the response option that was not associated with the now devalued outcome. Given that acute stress may increase sensitivity to social cues (*Domes and Zimmer, 2019*), one might argue that stressed participants continued to respond towards devalued outcomes in order to avoid being presented with the error-feedback screen. However, we consider this alternative to be unlikely. First, with respect to the neural outcome and response representations, these were analysed during NoDev blocks in which groups did not differ in their behavioural performance accuracy and consequently not in the frequency of error feedback. Furthermore, participants' performance in devalued blocks was directly

associated with the observed changes in neural outcomes and response representations during the NoDev blocks, which again, could not be biased by differences in error feedback processing.

In addition, stressed participants showed an increase in insensitivity to outcome in Dev $O^{high}$ but not in Dev $O^{low}$ trials. Moreover, we found that the devaluation effect for $O^{high}$ stimuli was stronger compared to the effect for $O^{low}$ stimuli. This difference remained even when we analysed only the control group, excluding the possibility that the difference between $O^{low}$ and $O^{high}$ was merely due to the fact that stress increased specifically the insensitivity to the devaluation of $O^{high}$. However, why may the devaluation effect be lower for $O^{high}$ than for $O^{low}$ and why may stress have affected primarily the devaluation of $O^{high}$? These results suggest a stronger habit formation for stimuli that were paired with high rewards. A potential explanation for this pattern takes the links between rewards, stimulus saliency, and strength of S-R associations into account; the initial association with high valued outcomes may have increased the salience of the respective stimuli, which in turn may have promoted the formation of S-R associations. These stronger S-R associations may have resulted in more habitual responses for the devalued outcomes.

Interestingly, participants either primarily selected their action based on the outcome value (i.e. by avoiding the devalued outcome and thus were goal-directed) or responded to the devalued outcome in the vast majority of the trials and thus behaved habitually. Thus, participants showed an 'either-or' pattern, and there seemed to be interindividual differences in the tendency to perform in a goal-directed vs. habitual manner. Interestingly, also among stressed participants, there were substantial individual differences in the propensity to shift from goal-directed towards habitual behaviour, indicating that there is no overall effect of stress on the control of behaviour but that individuals differ in their sensitivity in these stress effects. This raises the important question of what makes individuals more or less vulnerable to the effects of stress on instrumental control (and cognition in general). Previous work suggested that genetic variants related to major stress response systems (i.e. noradrenergic and glucocorticoid activity) may play an important role in which multiple learning systems are engaged after stress (*Wirz et al., 2017*; *Wirz et al., 2018*). Understanding which further individual factors contribute to the interindividual propensity to shift from goal-directed to habitual behaviour needs to be addressed by future research. Furthermore, the fact that our neural data were not bimodal suggests that changes in neural representations may not translate directly into behavioural changes. The observed significant correlations between neural representation and performance in devalued trials show that there is a link between the behavioural and neural representation level. However, the correlations were obviously far below 1. Compared to the behavioural level which included only discrete yes-no responses, the neural data may have been much more sensitive and able to capture more fine-grained changes. The different sensitivity of behavioural and neural data is interesting in itself and points to another important question for future research: how do neural representation changes and behavioural changes relate to each other? Is there a particular threshold at which a change in representation triggers behavioural changes?

While we assume that the opposing effects of stress on neural outcome and response representations were due to the action of major stress response systems, there might have been other factors that have contributed to the present pattern of results, such as motivational factors or fatigue at the end of the task. Although we cannot completely rule out these alternatives, we consider them rather unlikely in light of our data. First, if these factors were a result of the mere amount of training, they should have also occurred in the control group which was not the case. Even if a specific interaction with the stress manipulation is assumed, it is important to note that reaction times remained fast in stressed participants until the end of training, and the response accuracy in valued trials or consumption trials remained very high. Furthermore, the observed specificity of the stress effects which occurred selectively in devalued trials cannot - in our view - be explained by unspecific influences, such as lack of motivation or fatigue.

In the present study, stress was induced before learning and outcome devaluation. Thus, stress could have affected the acquisition or the expression of instrumental behaviour, or both. While several previous studies demonstrated that acute stress (or the administration of stress hormones) before learning may shift instrumental behaviour from goal-directed to habitual control (*Braun and Hauber, 2013*; *Dias-Ferreira et al., 2009*; *Gourley et al., 2012*; *Hartogsveld et al., 2020*; *Schwabe et al., 2010*; *Schwabe et al., 2011b*; *Schwabe et al., 2012*; *Schwabe and Wolf, 2009*; *Soares et al., 2012*), there is evidence suggesting that stress before a test of behavioural expression may have a similar

impact, that is, stress may induce habitual responding even when stress left acquisition unaffected (*Schwabe et al., 2011b*; *Schwabe and Wolf, 2010*). The latter finding, however, does not rule out additional effects of stress on acquisition, and indeed the impact of stress appeared to be more pronounced when the stress exposure took place before learning (*Schwabe and Wolf, 2010*). The present study did not aim to distinguish between stress effects on acquisition, or expression of goal-directed vs. habitual behaviour, but focussed on the impact of stress of the control of instrumental behaviour. Thus, our findings do not allow us to distinguish between stress effects on acquisition vs. expression of instrumental behaviour.

Based on the associative-cybernetic model (*Dickinson and Balleine, 1993*), it could be predicted that the obtained pattern of increased outcome and decreased response representations even leads to reduced responding for devalued outcomes across training in controls. This may be because individuals need to encode a representation of the response being performed in order to attain the outcome for a response to be goal-directed. We did not observe such a decrease, which may be attributed to the overall relatively low rate of responses for devalued outcomes in control participants.

Goal-directed action and habits are commonly considered to be two sides of the same coin. If a behaviour is more goal-directed, it is automatically less habitual (and vice versa). Obviously, behaviour cannot be fully goal-directed and habitual at the same time according to canonical operational definitions (*Adams, 1982*; *Dickinson and Balleine, 1994*). However, behaviour may not necessarily always be either fully goal-directed or habitual, and there may be different degrees to which behaviour is under goal-directed or habitual control. Given that the two modes of behavioural control are subserved by distinct neural circuits (*Balleine and O'Doherty, 2010*), it should be possible to separate goal-directed and habitual contributions to learning at a specific point in time. Classical behavioural paradigms involving discrete responses, however, cannot disentangle goal-directed and habitual components in a specific pattern of responding (e.g. insensitivity to outcome devaluation). Furthermore, tests of related cognitive functions, such as inhibitory control, provide only indirect evidence if any on the balance of goal-directed and habitual processes. Recently, a free-operant model was proposed that allows a behavioural dissociation of goal-directed and habitual contributions to behaviour (*Perez and Dickinson, 2020*). Here, we used here an MVPA-based decoding approach that focussed on neural representations that are a hallmark feature of goal-directed and habitual control. Using this novel approach, we show that acute stress reduces outcome representations and, at the same time, increases response representations in healthy humans. These were both directly linked to the stress-induced increase in habitual responding, suggesting that stress might exert opposite effects on goal-directed and habitual processing that manifest in the dominance of habitual behaviour under stress.

## Materials and methods
### Participants and design

Sixty-two healthy volunteers participated in this experiment. This sample size was based on earlier studies on stress and mnemonic control in our lab (*Schwabe and Wolf, 2012*) and a priori power calculation using G*POWER 3 suggesting that this sample size would be sufficient to reveal a medium-sized effect in a mixed-design ANOVA with a power of 0.80. Exclusion criteria were checked in a standardized interview before participation and identified any current or chronic mental or physical disorders, medication intake, or drug abuse. Furthermore, smokers and women taking hormonal contraceptives were excluded from participation because previous studies revealed that smoking and hormonal contraceptive intake may alter the cortisol response to stress. In addition, participants were asked to refrain from food intake, caffeine, and physical activity for 2 hr before testing. Four participants had to be excluded from analysis due to medication intake shortly before participation - leaving a final sample of 58 participants (32 men, 26 women; age: M=29.53 years, SEM = 2.57 years). Participants received a monetary compensation of 30 € for participation, plus a performance-dependent compensation (2–5 €). All participants gave written informed consent before entering the study, which was approved by the local ethics committee. In a between-subjects design, participants were randomly assigned to the stress (15 men and 15 women) or control group (17 men and 11 women).

## Stress and control manipulation

Participants in the stress condition underwent the TSST (*Kirschbaum et al., 1993*), a standardized stress protocol known to reliably elicit both subjective and physiological stress responses (*Allen et al., 2014*; *Kirschbaum et al., 1993*). Briefly, the TSST consisted of a mock job interview during which participants were asked to give a 5 min free speech about why they are the ideal candidate for a job tailored to their interests and a 5 min mental arithmetic task (counting backwards in steps of 17 from 2043 as fast and accurate as possible; upon a mistake they had to stop and start again from 2023). Both the free speech and the mental arithmetic task were performed in front of a cold and non-reinforcing panel of two experimenters (1 man and 1 woman) who were dressed in white coats and introduced as experts in 'behavioural analysis'. Furthermore, participants were videotaped throughout the TSST and could see themselves on a large screen placed next to the panel. In the control condition, participants gave a 5 min speech about a topic of their choice (e.g. their last holiday) and performed a simple mental arithmetic task (counting in steps of two) for 5 min while being alone in the experimental room - no video recordings were taken. During the control condition, the experimenter waited in front of the door outside the room where he/she was able to hear whether the participants had complied with the instructions.

To assess the effectiveness of the stress manipulation, subjective and physiological measurements were taken at several time points across the experiment. More specifically, participants rated the stressfulness, difficulty, and unpleasantness of the previous experience immediately after the TSST or control manipulation on a scale from 0 ('not at all') to 100 ('very much'). In addition, blood pressure and pulse were measured using an OMRON M400 device (OMRON, Inc, MI) before, during, and immediately after the TSST/control manipulation as well as 20, 60, and 105 min after the TSST/control manipulation. To quantify cortisol concentrations and elevations during the experiment, saliva samples were collected from participants using Salivette collection devices (Sarstedt, Germany) before, during, as well as 20, 60, and 105 min after the TSST/control manipulation. Saliva samples were stored at –20°C until the end of testing. At the end of data collection, we determined the free fraction of cortisol from the saliva samples using a commercially available luminescence assay (IBL, Germany).

## Reinforcement learning task

In order to investigate goal-directed and habitual contributions to behaviour, we used a modification of a recently introduced reinforcement learning task (*Luque et al., 2017*). In this reinforcement learning task, participants played the role of space traders on a mission to trade fractal stimuli (either a pink or a green fractal) for playing cards (either scene or object playing cards) with aliens from two tribes ('red alien tribe' and a 'blue alien tribe'). Each tribe traded only one type of playing card (i.e. either scene or object playing cards). One type of card (the high-value outcome, $O^{high}$, worth 100 points) was more valuable than the other one (the low-value outcome, $O^{low}$, worth 20 points). In addition, one tribe of aliens exchanged a playing card only for a specific fractal (e.g. the red alien tribe only traded scene playing cards and only for a pink fractal but not for a green fractal). If a fractal was given to the alien tribe, the exchange was rejected by the alien (e.g. if the participant wanted to trade the green fractal with the red alien). In that case, participants did not receive a playing card and hence did not gain any points. Importantly, participants had to pay 5 points for every successful exchange (response costs). If a playing card was given to the incorrect alien and the exchange was denied, there were no response costs. Participants were encouraged to earn as many points as possible. At the end of the experiment, participants received a monetary reward dependent on the number of points earned throughout the reinforcement task.

Because the critical difference between goal-directed action and habitual responding is dependent on whether behaviour is sensitive to an outcome devaluation or not (*Adams and Dickinson, 1981*), specific outcomes were devalued in some of the task blocks. Participants were told that both types of cards needed to be kept in different boxes (scene cards in a 'scene box' and objects cards in an 'object box') but that sometimes one box was full and needed to be emptied. In that case, participants were not able to store cards of the said category anymore. Thus, in these blocks, the cards were worthless regardless of their usual value (i.e. the outcome was devalued).

The status of the boxes (available/not available) varied only between blocks. Participants were informed about the status of the box via a message on the screen at the beginning of each block. In blocks without outcome devaluation (NoDev), both types of cards could be kept in their respective

boxes and had their usual value. In blocks with devaluation of the high-valued outcome (Dev $O^{high}$), $O^{high}$ cards could not be stored and thus their new value was zero, whereas $O^{low}$ cards could be stored and thus had still the usual value. In blocks with devaluation of the low-valued outcome (Dev $O^{low}$), $O^{low}$ cards could not be saved and had no value anymore while $O^{high}$ cards still had high value (*Figure 1*). Importantly, due to the response costs (5 points for each card exchange), giving a card to an alien that traded the devalued card category was detrimental and should be avoided if behaviour is under goal-directed control.

Reinforcement learning trials were composed of S-R-O sequences. At the beginning of each trial, a fixation cross was presented on the centre of the computer screen for a random duration between 800 and 1200 ms. Then, one of the two highly distinct fractals (high-value stimulus, $S^{high}$, or low-value stimulus, $S^{low}$) was presented. After 2000 ms, pictures of two aliens were shown at either side of the screen. The two aliens represented two different response options (R1 and R2). Participants were instructed to choose via a button press as to which alien they wanted to give the fractal. After their response, the screen was cleared, and the outcome feedback appeared on the screen for 1000 ms. If participants chose the incorrect alien, the message 'The exchange was denied. You don't receive a playing card!' was presented. This error feedback was presented in both NoDev and Dev blocks. If they did not respond within 2 s, the message 'Time out, please respond faster!' was displayed. The specific stimuli used as $S^{high}$ and $S^{low}$, R1 and R2, and $O^{high}$ and $O^{low}$ were counterbalanced across participants. The position of the aliens (left/right) representing the response options was randomized across trials. The feedback provided on each trial varied between blocks. During NoDev blocks, participants saw the value of the card that was gained before, as well as the corresponding response costs ('+100/–5' for $O^{high}$ and '+20/–5' for $O^{low}$). In Dev blocks (Dev $O^{high}$ or Dev $O^{low}$ blocks), this information was masked for all trials. Before Dev blocks, participants were informed that the outcome feedback was unavailable during these blocks because of a solar interference (in devaluation blocks, participants saw on the last screen of each trial '???' instead of value), but that this solar interference would not influence the value of the cards that they could gain.

In addition to reinforcement learning trials, we included consumption trials in each block to test whether participants understood the task structure. At the beginning of the experiment, participants were instructed that sometimes the aliens become distracted and participants could take one of the two types of cards without trading it for fractals, and therefore without paying response costs. In these consumption trials, the message 'The aliens seem distracted …' was shown for 1000 ms, followed by a countdown (from 3 to 1) in the centre of the screen (over 3 s). Then, one scene and one object card appeared. One on the left and the other on the right (positions were selected randomly for each consumption trial). Participants choose one card via button press, and their choice revealed whether they were aware of the value of the stimuli in the respective block (NoDev, Dev $O^{high}$, and Dev $O^{low}$). Outcome screens were identical to those of the reinforcement learning trials.

Participants completed 27 NoDev blocks, 3 Dev $O^{high}$ blocks, and 3 Dev $O^{low}$ blocks, with 27 trials per block: 12 learning trials with $S^{high}$, 12 learning trials with $S^{low}$, and 3 consumption trials. Reinforcement learning trials within each block were presented randomly. Consumption trials were displayed at trial numbers 7, 14, and 21 within each block. The $O^{high}$ was always devaluated during the 2nd, 16th, and 29th block (Dev $O^{high}$), whereas $O^{low}$ was always devalued in the 3rd, 17th, and 30th block (Dev $O^{low}$). Thus, these Dev blocks were presented at the beginning, middle, and end of the training.

At the end of each block, participants saw how many cards they had gained in the previous block, the total point value of the cards earned in that block, and the total number of points they had gained until then. Then, the next block started with the information about which box(es) was/were available (i.e. whether any outcome was devalued) for the upcoming block.

Performance during the devalued trials revealed the degree to which instrumental behaviour was goal-directed or habitual. Goal-directed action is indicated by the formation of S-R-O associations. Thus, if participants used a goal-directed system, they should adapt their responses to the actual outcome value following outcome devaluation. During devalued trials, participants did not earn any points if they chose the response associated with the devalued outcome. Importantly, however, they had to pay response costs leading to a subtraction of points. Thus, goal-directed participants should choose the response option associated with a trade rejection. Under such circumstances, participants did not have to pay any response costs (*Figure 1*). In contrast, habitual behaviour is reflected in simpler S-R associations rendering instrumental behaviour insensitive to changes in outcome value.

Hence, choosing the action associated with the devalued outcome (where no points could be earned and response costs had to be paid) indicated less goal-directed and more habitual behaviour.

## DMS task

In order to analyse the neural representations of response options and action outcomes, we trained an EEG-based classifier (see below) on a DMS task. This task was presented before and after participants had completed the reinforcement learning task in order to avoid any time-dependent biases in the trained classifier. In each DMS task, participants completed 128 trials. Participants were presented four different target types: object cards, scene cards, blue, and red symbols. In addition to colour, symbols also differed in shape, line orientation, and line position (blue rectangles with left-oriented lines in the upper area vs. red ovals with right-oriented lines in the lower area). Pictures used as targets were selected randomly from a pool of 256 pictures (90×object cards, 90×scene cards, 48×blue symbols, and 48×red symbols) with the restriction that successive trials did not belong to the same category more than three consecutive times. The remaining pictures were used as targets during the second DMS task. On each trial, the target was shown for 2 s on the centre of a computer screen. Participants were asked to hold it in mind during a subsequent delay phase of 2 s during which they saw a blank screen. Then, a probe stimulus was presented (the target and a distractor belonging to the same category), and the participants had to indicate via button press which picture they saw before. The position of the target during the response choice (right vs. left) was randomized. Different stimuli were used in the second DMS task compared to the first one.

## Eye-tracking

We used a desktop mounted eye-tracker (EyeLink 1000; SR-Research Ltd., Mississauga, Ontario, Canada) to record monocular eye movements from the left eye at a sampling rate of 500 Hz. We used custom scripts implemented in MATLAB (The Mathworks, Natick, MA) to extract the mean saccades and blink information depending on the stimulus value, outcome devaluation, and time (0–2000 ms around the stimulus onset). Data for two participants was missing due to failed use of the eye-tracker.

## Control variables

In order to control for potential group differences in depressive mood, chronic stress, and anxiety, participants completed the Beck Depression Inventory (*Beck et al., 1996*), the Trier Inventory for the Assessment of Chronic Stress (*Schulz and Schlotz, 1999*), and State-Trait Anxiety Inventory (*Spielberger, 1983*) at the end of the experiment. In addition, participants completed a German mood questionnaire (MDBF; *Steyer et al., 1994*) that measures subjective feeling on three dimensions (elevated vs. depressed mood, wakefulness vs. sleepiness, and calmness vs. restlessness) at the beginning of the experiment.

## Procedure

All testing took place in the afternoon and early evening between 1 pm and 8 pm. Participants were instructed to refrain from excessive exercise and food or caffeine intake for the 2 hr before testing. After participants' arrival at the laboratory, EEG was prepared, blood pressure measurements were taken, and a first saliva sample was collected. Participants also completed the mood scale (*Steyer et al., 1994*). Then, participants performed the first DMS task. After completing this task, participants received written instructions about the reinforcement learning task. In order to further familiarize participants with the structure of this task, participants completed a 5 min short tutorial afterwards. Next, participants underwent either the TSST or the control manipulation. Immediately thereafter, subjective assessments of this manipulation and another saliva sample were collected, and blood pressure was measured once again. Next, participants were briefly reminded of the instructions for the reinforcement learning task they had received before. Twenty minutes after the TSST/control manipulation, when cortisol was expected to have reached peak levels (*Kirschbaum et al., 1993*), participants collected another saliva sample before the reinforcement learning task commenced. After the 15th block of the reinforcement task and after finishing the task, further saliva samples were collected, and blood pressure was measured again (~60 min and ~105 min after stress onset). Finally, participants performed the second DMS task.

## Statistical analysis

Subjective and physiological stress responses were analysed by mixed-design ANOVAs with the within-subject factor time point of measurement and the between-subjects factor group (stress vs. control). Participants' responses in the reinforcement learning task were subjected to mixed-design ANOVAs with the within-subject factor; stimulus type ($S^{high}$ and $S^{low}$), outcome devaluation (NoDev: 1st, 12th, and 28th block; Dev $O^{high}$: 2nd, 16th, and 29th block; Dev $O^{low}$: 3rd, 17th, and 30th block), time point (1st, 2nd, and 3rd), and the between-subject factor group. Significant interaction effects were followed by appropriate post hoc tests. All reported p values are two-tailed and were Bonferroni corrected ($p_{corr}$) when indicated. Statistical analyses were calculated using SPSS 25 (IBM SPSS Statistics) and JASP version 0.13.0.0 software (https://jasp-stats.org/).

For one participant, we obtained only invalid trials during the last Dev $O^{low}$ block. Thus, this participant could not be included in analyses of Dev $O^{low}$. Furthermore, two participants did not complete the Trier Inventory for the Assessment of Chronic Stress (*Schulz and Schlotz, 1999*). Data of 5 participants had to be excluded from the EEG analysis because of technical failure during the EEG, leaving a sample of 53 participants (control: n=25; stress: n=28) for EEG analyses.

## EEG recordings

During the DMS and reinforcement learning task, participants were seated approximately 80 cm from the computer screen in an electrically shielded and sound proof cabin. EEG was recorded using a 128-channel BioSemi ActiveTwo system (BioSemi, Amsterdam, The Netherlands) organized according to the 10–5 system digitized at 2024 Hz. Additional electrodes were placed at the left and right mastoids, approximately 1 cm above and below the orbital ridge of each eye and at the outer canthi of the eyes. The EEG data was online referenced to the BioSemi CMS-DRL (common mode sense-driven right leg) reference. Electrode impedances were kept below 30 kΩ. EEG was amplified with a low cut-off frequency of 0.53 Hz (=0.3 s time constant).

## EEG analysis
### Preprocessing

Preprocessing was performed offline using FieldTrip (*Oostenveld et al., 2011*) and EEGLAB (*Delorme and Makeig, 2004*) as well as custom scripts implemented and processed in MATLAB (The Mathworks, Natick, MA). The PREP pipeline procedure (*Bigdely-Shamlo et al., 2015*) was utilized to transform the channel EEG data using a robust average reference. In addition, bad channels were interpolated using the *spherical* option of EEGLAB *eeg_interp* function. Then, data was filtered with a high pass filter of 0.1 Hz and a low pass filter of 100 Hz and downsampled to 250 Hz. For MVPA, epochs from the DMS task (2000 ms relative to the delay onset) and from the reinforcement learning task (2000 ms relative to the onset of stimulus and response option presentation) were extracted. For ERP analysis, EEG data was segmented into epochs from –200 to 2000 ms around the stimulus onset and baseline-corrected by subtracting the average 200 ms prestimulus interval. In addition, blinks and eye movements were corrected by independent component analysis (infomax ICA, *Noh et al., 2014*). Using the automated procedure ADJUST (*Mognon et al., 2011*), ocular artefact-related components in EEG recordings were identified and subsequently removed. The ADJUST algorithm combines stereotypical artefact-specific spatial and temporal features to detect and differentiate artefact ICA components (*Chaumon et al., 2015*). For example, ADJUST computes the kurtosis of the event-related time course for frontal channels, since, for example, eye blinks are accompanied by abrupt amplitude jumps in frontal regions areas (stereotypical artefact-specific temporal feature). Additionally, ADJUST determines the spatial topography of the IC weights to compare the magnitude of the amplitudes between frontal and posterior areas (stereotypical artefact-specific spatial feature). Using the ADJUST procedure, on average 1.65 (SEM = 0.13) components per participant were detected and removed. Previous data shows that the automatic detection of artefact components using ADJUST leads to a comparable classification of artefact components that are afforded by manual classification by experiments (*Mognon et al., 2011*).

## ERP analysis

Based on previous studies (*Hickey et al., 2010*; *MacLean and Giesbrecht, 2015*), we expected to find an effect of stimulus value (i.e. the difference between activity elicited by $S^{high}$ and $S^{low}$) in the occipital P1 component peaking within the time window from 75 to 200 ms relative to stimulus onset. Because all stimuli were presented centrally, P1 activity was analysed in the midline occipital electrode Oz. The P1 peak was defined as the largest positive peak between 75 and 200 ms after the stimulus onset at Oz (averaging across all conditions). In line with previous research, a time window of 70 ms around that peak was then selected for analysis (*Luque et al., 2017*). Because the P1 maximum amplitude was at 115 ms from stimulus onset across participants, the P1 magnitude for each condition was defined as the mean EEG signal across the 80–150 ms time window.

Based on previous research on the neural underpinnings of goal-directed action (*Luque et al., 2017*), we further assessed the effects of stress on brain activity over centroparietal regions at a later time window. To this end, ERP data from 400 to 700 ms was subdivided into six consecutive, non-overlapping time bins - with a duration of 50 ms each. The later ERP data were analysed using mixed-design ANOVAs with the within-subject factors; outcome devaluation (NoDev, Dev $O^{high}$, and Dev $O^{low}$) and stimulus value ($S^{high}$ and $S^{low}$) and the between-subject factor group. Significant outcome devaluation × stimulus value × group interaction effects were appropriately corrected for multiple comparisons.

## MVPA training

The multivariate decoding analyses were implemented using the MVPA-Light toolbox (*Treder, 2020*). The classifier was trained within-subject, using a linear SVM on the preprocessed data of the DMS task (delay phase). All EEG channels were used as features. To improve the signal to noise ratio for MVPA, trials were averaged to pseudo trials (*Isik et al., 2014*). Each pseudo trial was an average of two trials. We performed two separate analyses corresponding to the following classes: object vs. scene ($SVM^{object/scene}$) and blue symbol vs. red symbol ($SVM^{blue/red}$). To identify the optimal time window for decoding per participant, we implemented a sliding window averaging 100 ms with a step size of 10 ms. The $SVM^{object/scene}$ and $SVM^{blue/red}$ with the highest performance were used to decode the neural outcome and response representation. Generalization of the classifier was evaluated using a leave-one-out procedure. If the classifier's performance remains significantly above chance, it indicates that the EEG patterns contain class-specific information and that the class can be reliably decoded from the EEG data (*Murphy et al., 2011*). The chance level in a simple two-class paradigm is not exactly 50% but 50% with a CI at a certain alpha level. Therefore, we calculated this interval utilizing the Wald interval with adjustments for a small sample size (*Agresti and Caffo, 2000*; *Müller-Putz et al., 2008*). The threshold for chance performance was 63.59% for the classification of blue vs. red symbols and 60.11% for object vs. scene images. Participants with classification accuracy below chance in the DMS task were not included in subsequent analyses. During the classification of blue vs. red symbols, the highest performance of two participants did not exceed the threshold for chance performance. During the classification of blue vs. red scenes, classification accuracy of 13 participants was not significant. Hence, the sample for analysis of the outcome representation (based on $SVM^{object/scene}$) and response representation (based on $SVM^{red/blue}$) reported in the main text was 41 (control: n=18; stress: n=23) and 51 (control: n=25; stress: n=26) participants. Importantly, we computed all analyses again including all participants regardless of significant classification scores. This additional analysis left our findings largely unchanged (*Supplementary file 1B*). Furthermore, the classification accuracy of the first and the second DMS task did not differ (blue/red symbol classification: $F[1, 51] = 0.66$, p=0.798, $\eta_p^2$=0.013, and 95% CI=0–0.127; DMS session × group interaction: $F[1, 51] = 0.03$, p=0.863, $\eta_p^2$=0.001, and 95% CI=0–0.024; object/scene classification: $F[1, 51] = 2.87$, p=0.096, $\eta_p^2$=0.053, and 95% CI=0–0.203; DMS session × group: $F[1, 51] = 0.03$, p=0.860, $\eta_p^2$=0.006, and 95% CI=0.003–0.011). Thus, we pooled the data from both DMS task sessions and trained an overall classifier to ensure that the classifier was not affected by any time-related biases, and that there is a sufficient number of trials to train a reliable classifier.

## MVPA decoding

The SVM$^{object/scene}$ and SVM$^{blue/red}$ trained on the independent DMS dataset were used to assess the respective outcome (object card vs. scene card) and response representation (blue alien vs. red alien) in the trial-by-trial reinforcement learning task during the NoDev blocks. Both classifiers were applied to the respective test data (with an epoch size of 2000 ms relative to stimulus onset) using an overlapping sliding window, with a time average of 100 ms and a step size of 10 ms. The maximum classification accuracy indicated the strength of outcome and response representation.

Goal-directed behaviour is characterized by an action-outcome (S-R-O) association, which should result in an anticipatory outcome representation during stimulus presentation and response choice. Hence, we applied the SVM$^{object/scene}$ to the stimulus presentation and response choice phase of the reinforcement learning task in order to decode the outcome representation. With increasing habitual control of behaviour, outcome representations should be reduced. Response representations, may in turn be relevant for both habitual S-R learning and goal-directed S-R-O learning. Therefore, we applied the SVM$^{blue/red}$ to the stimulus presentation phase to decode the response representation. In retrospect, a search-light RSA(representational similarity analysis) asking which features contribute the most to decoding neural outcome and response representations revealed that at the time of average maximal decoding accuracy (about 200 ms after stimulus onset for both classifications), a parieto-occipital cluster was contributing the most to decoding outcome representations. In contrast, a centroparietal cluster was contributing the most to decoding response representations (*Figure 2— figure supplement 4*).

Maximal accuracy values were then averaged over six blocks to get a reliable indicator of classification accuracy. Statistical analyses of the EEG decoding data were performed at the group level averaging across individual decoding accuracies. Decoding accuracies were analysed using mixed-design ANOVAs with the within-subject factor block (1–6, 7–12, 13–18, and 19–24 blocks) and the between-subject factor group. In addition, we computed Spearman correlations between Δ classification scores (averaged classification accuracy during the first six blocks minus the classification accuracy during the last six blocks) and Δ responses for devalued outcomes for Dev O$^{high}$ blocks (responses for devalued outcomes during the 1st devaluation block, i.e. the 2nd overall block, minus responses for devalued outcomes during the last devaluation block, i.e. the 29th overall block). Bayesian correlation analyses were conducted using JASP version 0.13 software (https://jasp-stats.org/) and the default Cauchy prior 0.707.

## Searchlight approach

To provide more insight into which electrophysiological information contributed most to differentiating between the respective categories (red vs. blue and objects vs. scenes, respectively), we also performed searchlight analyses. These allowed us to determine those topographic features that discriminated most between the two sets of categories. We first pooled the segmented ERP data (0–2 s relative to stimulus onset) from all participants. Then, we averaged the time range of ±100 ms relative to the respective maximum decodability. For each searchlight analysis, a cluster was constituted by a centre electrode and all neighbouring electrodes within a radius of 4 cm. The average cluster size was 10.61 electrodes (SEM = 0.31). We used an SVM as classifier and calculated the Wald interval with adjustments for a small sample size to evaluate the accuracy values (*Agresti and Caffo, 2000*; *Müller-Putz et al., 2008*).

## Acknowledgements

We thank Keyvan Khatiri, Charlotte Germer, Marian Wiskow, and Yichen Zhong for their assistance during data collection and Polina Perzich for critical proofreading of the manuscript. We further gratefully acknowledge the support of Carlo Hiller with programming of the software. This study was funded by the German Research Foundation (DFG, SCHW1357/23-1).

## Additional information

### Funding

| Funder | Grant reference number | Author |
|---|---|---|
| Deutsche Forschungsgemeinschaft | SCHW1357/23-1 | Lars Schwabe |

The funders had no role in study design, data collection and interpretation, or the decision to submit the work for publication.

### Author contributions

Jacqueline Katharina Meier, Data curation, Formal analysis, Investigation, Visualization, Methodology, Writing - original draft; Bernhard P Staresina, Conceptualization, Funding acquisition, Methodology, Writing - review and editing; Lars Schwabe, Conceptualization, Resources, Supervision, Funding acquisition, Methodology, Writing - original draft, Project administration, Writing - review and editing

### Author ORCIDs

Bernhard P Staresina (iD) http://orcid.org/0000-0002-0558-9745
Lars Schwabe (iD) http://orcid.org/0000-0003-4429-4373

### Ethics

All participants provided informed consent before participation in the experiment. The experiment was performed in line with the Declaration of Helsinki and approved by the ethics committee of the Faculty of Psychology and Human Movement Sciences at the Universität Hamburg (2018_197_Schwabe).

### Decision letter and Author response

Decision letter https://doi.org/10.7554/eLife.67517.sa1
Author response https://doi.org/10.7554/eLife.67517.sa2

## Additional files

### Supplementary files

• Supplementary file 1. Supplementary material. (a) Classification results (block × group interaction) block for alternatively grouped data. (b) Classification results on outcome representations for N=53. (c) Control variables: subjective mood, depressive mood, anxiety, and chronic stress. (d) Eye-tracking data: mean number of blinks and saccades across time, for $S^{high}$ and $S^{low}$ trials (N=51).

• Transparent reporting form

### Data availability

Data reported in this manuscript are available from the website: https://github.com/08122019/From-goal-directed-action-to-habit, copy archived at swh:1:rev:d52c6db7505b203310ec623db4c07fede10eb9d.

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
