## [Editor Report]

The authors combined a cleverly designed behavioral task with EEG-based multivariate pattern analysis and acute stress induction to assess the neural representations mediating an influence of stress on the balance between goal-directed and habitual responding. They found that stress induction reduced neural outcome representations, and that this representational change correlated with the degree of habitual performance. The topic and approach should be of interest to a wide audience, ranging from clinicians to economists and neuroscientists.

---

## [Decision Letter]

**Decision letter after peer review:**

Thank you for submitting your article "Stress diminishes outcome but enhances response representations during instrumental learning" for consideration by *eLife*. Your article has been reviewed by 3 peer reviewers, including Mimi Liljeholm as the Reviewing Editor and Reviewer #1, and the evaluation has been overseen by Floris de Lange as the Senior Editor.

Essential revisions:

(1) Several clarifications must be made to address the possibility that saccadic eye movements could be acting as a confound for some of the EEG decoding results, including an account of which features were most important for classification performance, and how eye-movements and blinks were dealt with in preprocessing.

(2) It appears that devaluation insensitivity, and reduced neural outcome representations, emerge at a point when stress measures no longer indicate a stress state. Please clarify how the time courses of these measures relate to the claims about an influence of stress on devaluation insensitivity.

(3) The proposed role of extended training is consistent with the fact that significant devaluation insensitivity and reduced outcome representations emerge in the last block in the stress group. However, the notion that habits develop with extended training seems inconsistent with the apparent decrease in response representation and increase in outcome representation across blocks in the control group, as well as the apparent increase in devaluation sensitivity in the stress group across blocks 1 and 2. These inconsistencies must be resolved and claims modified accordingly.

(4) An important aspect of assessing devaluation sensitivity in the animal literature is a reinforced test, in which the devalued outcome is delivered contingent on the relevant response – as a result, habitual animals rapidly reduce performance of the devalued action. Such tests rule out a general lack of attention, response perseveration etc. The absence of a reinforced test should be explicitly addressed and the implications discussed.

(5) It is unclear from Figure 1 whether the "Error" feedback screen was employed during devaluation blocks. If so, please address the potential influence of negative social feedback on avoidance of the non-devalued action, particularly in the group that had just been subjected to a social stress test.

(6) A range of statistical approaches are employed: please provide a section that details and justifies each.

*Reviewer #1 (Recommendations for the authors):*

(1) The devaluation, but not accuracy data, is clearly bimodal, in both the control and stress groups. I would like to see this discussed in more detail, in terms of consistency with physiological and neural measures, as well as in terms of broader theoretical implications.

(2) It looks like the response representation is actually decreasing across blocks in the control group. This seems inconsistent with the general argument that habits should develop as training progresses.

(3) "…while model-free learning can be present after only few trials, establishing habit behaviour requires extensive training." This statement is problematic for several reasons. First, it appears to be contradicted in the very next paragraph, where the authors acknowledge that the necessity and sufficiency of extensive training for habit expression in human subjects is controversial. Second, in terms of operational definitions, habits are instrumental responses that are insensitive to outcome devaluation, whether extensively trained or not. Likewise, theoretically, habits are usually attributed to the strength of the stimulus-response association, which may depend on repetition, but also on the magnitude of the reward, as well as, presumably, a range of other factors (e.g., stimulus salience).

(4) Given the schematic of the goal-directed system (top of Figure 1A), it's a bit strange that outcome and response representations would be uncorrelated.

(5) The violin plots are great for illustrating the bimodality in devaluation sensitivity, but it would also be useful to see all of these related variables plotted in the same way, preferably in a single figure. Please include line graphs showing the degree of responding for the devalued outcome, the decoding accuracy for responses and outcomes, and the physiological stress measures, across groups, in a way that allows for a visual, side-by-side, comparison.

(6) Please include graphics of devalued trials in Figure 1C, to illustrate the masking of outcomes and the nature of the error feedback on such trials.

(7) In the Results section, following the description of the task and procedures, but prior to reporting any results, please add a section detailing and justifying the range of statistical approaches. For example, some reported p-values for t-tests appear to be uncorrected (e.g., line 358), as are all p-values for correlation tests. Moreover, a Bayesian analysis is used to assess evidence against a correlation, but not evidence in favor of a correlation.

(8) The phrase "responses to devalued actions", employed throughout, is a bit confusing. I would change this to "responses for devalued outcomes".

(9) I'm guessing that Figure 3 is showing proportions, not percentages.

*Reviewer #2 (Recommendations for the authors):*

It was not clear to me what the main hypothesis of the study was. The authors seem to talk rather loosely about habits being developed after overtraining versus the mediating effect of stress on habits. The introduction should convey their main goal more clearly.

In my view, this task is not meant to test for overtraining effects as it includes repeated devaluation of the outcome within the same subject, which makes it very difficult to assess the extent to which overtraining -rather than simply lack of motivation at the end of the task or other factor- is explaining weaker devaluation effects late in training. This is a critique for any experiment looking at overtraining effects, bit it should be at least demonstrated that what is driving the effect found by the authors is indeed an effect of overtraining by looking at more than one extension of training. Did the authors test other extensions of training in pilot studies?

It does not seem to me that the task is relevant for the habit/goals distinction. The authors mention how the model-based/model-free distinction may not be interrogating the habit/goal distinction, but this task suffers from the same shortcomings. This is a problem in general with this type of experiments, in that a participant could behave in an S-R or R-O manner independently of the amount of training and switch back and forth between these two strategies throughout the task. The task is simply not well-suited for testing the habit/goal-directed distinction.

In addition, the task also employs multiple devaluation phases, which, if anything, should make participants more sensitive to the devaluation procedure. Was that the case?

It would be good to see the learning curves associated with each group of subjects. The authors' main finding is an effect that is only observed in the last block of training and only for one condition (when the outcome that was highly valued is devalued). Overtraining effects are only relevant after responding has plateaued. Inspecting if both groups have already asymptoted, or whether one did it before the other is an important point, I think.

The same authors have shown that stress manipulations after training render behavior insensitive to devaluation. This suggests that the effect of stress is on the performance of an action, not the learning process per se. How does this fit with their current results showing that stressing people before training changes devaluation sensitivity only in the last block of training? Why there is an extension of training effect in O representation for stressed participants if the effect is on performance?

The authors show that the representation of O for the controls increases with training (Figure 4) while the R representation decreases. Theoretically, these two observations should entail a weaker devaluation effect, as the subject needs to encode a representation of the response being performed in order to attain the outcome for a response to be goal-directed (see recent work from Bouton's lab and Dickinson and Balleine's associative-cybernetic model). Perhaps this should be discussed?

Related to this, the fact that a control group did not show evidence of a decrease in outcome representations at the end of training seems problematic, as their argument is based upon the notion that more extensive training makes responding habitual in the stress group but not in the control group. If stress impacts learning, it should there is no argument for the fact that overtraining in a control group does not change outcome representations.

Why is there a stronger effect of devaluation on the O-Low condition across the task? Why a less-valued outcome should be able to support this? (pp. 6, line 239)

The devaluation effect for the stress group in block 2 seems stronger than in block 1. If that is the case, then the overtraining argument becomes problematic, as it should supposedly lead to weaker effects of devaluation with training, that is, devaluation should be weaker in block 1 than block 2, and weaker in block 2 than block 3 (the latter contrast being the one they report as being statistically significant). Also, is the difference between stress and control in block 3 different to the difference between stress and control in block 2?

The theoretical aspect that the study aims go address is whether the devaluation effects are due to less goal-directed control or more habitual control. In their Discussion section, the authors argue that a response cannot be goal-directed and habitual at the same time. However, Perez and Dickinson (2020) have recently published a free-operant model where it is perfectly possible that a response is driven in equal proportions by each system. It is only in the case of discrete-trial procedures such as the present one and the 2-step task from the model-based/model-free tradition that the answer is difficult.

It seems puzzling that the control group shows decreased representations of the response with training. If anything, that should be maintained or augmented with training. This is another reason to question the overtraining argument of this paper.

Other points:

The analysis of response representations after extended training is based in comparisons of first half and last half of training, why that choice?

In the discussion, the authors mention that "habit behavior requires extensive training", but a recent preprint by Pool et al. (2021) shows that most participants can be habitual after limited training.

Pp 24. Line 522: "Furthermore, while model-free learnins is based on previously experienced outcomes, habit learning is defined as S-R learning without links to the outcome engendered by a response." I could not follow this idea, can you clarify?

During the presentation of the results, the authors show statistics for both the O-high and O-low conditions. However, the O-low condition did not add anything to the analysis, as it did not show any effects of devaluation. Therefore, for the sake of simplicity and the reader's joy, I'd simply leave that for the supplemental material and focus on the part of the experiment that is relevant for the point the authors are trying to make.

Table 1: The caption mentions that the scales ranged from 1-10, but the results show means higher than 10. Am I missing something here?

Pp. 6, line 134 "offer, trade" should apparently read "offer, or trade"

This approach was employed by McNamee and colleagues (2015) using fMRI. I think it should be cited.

pp. 15, line 308: "tended to be sensitive". I'd just say it was not significant.

pp. 18, line 370: P = 0.034, should be P_corr?

Pp 31, lines 683-684. Could you please clarify the trial structure and numbers? What do you mean by "27 trials each"?

*Reviewer #3 (Recommendations for the authors):*

1. I think that the manuscript would benefit from more of an effort to link the current results to other related work. For one, the authors briefly touch on how the results relate to the 'model-based and model-free' dichotomy described in other work, but mostly highlight how the operationalization of 'model-free behavior' differs from 'habits,' as described here. However, I think it is notable that the authors find that habitual behavior involves a reduction in prospective consideration of an outcome, similar to other work that has shown that model-based behavior increases prospective consideration of future states during decision-making (e.g. Doll et al., 2015). Do the authors believe that 'habit/goal' and 'model-free/model-based' decision-making strategies might share common mechanistic features?

2. I am somewhat concerned that saccadic eye movements could be acting as a confound for some of the EEG decoding results and would appreciate the authors clarifying a few points related to this and possibly doing some additional control analyses to rule out this possibility:

a. The authors mention that the symbols for the aliens differed by position – was this position fixed for each alien tribe, i.e. did the blue tribe always appear on the left for a particular participant? If so, this could drive anticipatory eye-movements in that direction.

b. It is not clear which electrodes are driving the change in outcome and response representation in the decoding analysis. An analysis examining which features are contributing most to successful decoding in the stimulus and response period would be particularly important to rule out the possibility that the main results of interest are not driven by anticipatory eye-movements, such as saccades in the response phase where two stimuli are placed side-by-side.

c. Relatedly, I would have appreciated more detail on the preprocessing pipeline used to clean-up the data, in particular how saccadic eye-movements and blinks were corrected, or removed, and if central fixation was enforced with eye-tracking or EOG.

3. Looking at the scatterplots of Figure 5A and 6, the values on the y-axis appear to be rather different. Where the control and stress groups are mostly separate on this dimension in Figure 5A, they are highly overlapping in Figure 6 such that there does not appear to be any group difference. Are these different data and if so what is the rationale for selecting one set of data for one analysis and a different set for the other?

[Editors' note: further revisions were suggested prior to acceptance, as described below.]

Thank you for resubmitting your work entitled "Stress diminishes outcome but enhances response representations during instrumental learning" for further consideration by *eLife*. Your article has now been evaluated by 2 peer reviewers, and by the Reviewing Editor, Mimi Liljeholm, under the supervision of Senior Editor Floris de Lange. The reviewers have opted to remain anonymous.

The manuscript has been improved but there are some remaining issues that need to be addressed, as outlined below:

(1) Reviewer 2 notes that enhanced response representations do not necessarily reflect an increased involvement of the habit system – this is particularly true if such representations are not negatively correlated with outcome representations, since the latter retrieves the former according to several associative accounts of goal-directed behavior. Thus, the stated assumption, that "habitual responding is reflected in enhanced response representations" is not valid. The reviewer suggests, and I agree, that claims based on this assumption should be removed or significantly toned down.

(2) Reviewer 3 notes that the new FRN analyses are not sufficiently contextualized, and may even be erroneous, and also that the previous request for more information regarding decoding features went unanswered.

(3) For my part, I am still confused about the distinction between model-free RL and habits. The authors state:

"It might be argued that an alternative way of disentangling goal-directed and habitual contributions to instrumental responding under stress would be through an analysis of model-based and model-free learning …"

Who would argue that? Why would that analysis be better? What exactly would evaluation of these models add here?

And then:

"For instance, while model-free learning can be present after only few trials, establishing habit behaviour relies on the strength of the stimulus-response association, which among other factors such as reward magnitude or stimulus saliency, has been suggested to depend on repetition Furthermore, while model-free learning is based on previously experienced outcomes, habit learning is defined as S-R learning without links to the outcome engendered by a response … In other words, whereas model-free learning is dependent on the expected reward and hence the outcome that follows a response, habit learning is operationally defined as being independent of the motivational value of the outcome

This is incorrect. The model-free Q-value *is* the S-R association: both depend on the reward prediction error and the learning rate, both reflect previous reinforcement, and in that sense an expectation of reward, both are void of sensory-specific outcome features, and thus insensitive to devaluation, and both can develop rapidly or not, depending on the learning rate. I suggest that the entire section on model-based and model-free learning is remove.

(4) Finally, the writing needs some attention. As an example, it is repeatedly stated that canonical behavioral assays can "hardly" disentangle habitual and goal-directed behavior. The term "hardly" means barely, as in "I can barely see that distant object", or alternatively, it can mean definitely not, as in "You can hardly expect me to …" I don't think either is the intended meaning here, and there are other problems throughout the manuscript. I strongly suggest proofing by a native English speaker.

*Reviewer #2 (Recommendations for the authors):*

The authors have made a good effort to respond to all of my concerns. Given that my initial concerns with the task were not well received, I will not insist at this point on the merits of this task to produce or investigate habitual control, so I will focus on the author's responses only and some theoretical points that I think should be re-written to improve the quality of the manuscript.

The authors insist that their aim *was not* to investigate overtraining, but the very definition of overtraining that one could infer from the habit literature is that it refers to the amount of training that produces habitual behaviour. They used devaluations across different extensions of training to investigate whether " stress leads to an increase in habitual processing, a decrease in goal-directed processing or both ", so I'm still not convinced with their assertion.

Related to this, they are strongly assuming that the decoding analysis will give insights as to the interaction and development of habitual and goal-directed (g-d), but you need a theory of habits and actions in order to test for that. In my view, what they are doing in this paper is more empirical than theoretical ("do outcome representations decrease with training?; do response representations increase with training?), and I'd suggest they delete any reference to what they believe is the interaction between the systems in this task – right now they are motivating the paper as providing insights into the question of whether habits are a consequence of increased habits or decreased g-d control. I personally see this as a more empirical than theoretical paper, and the current motivation is poorly justified from a theoretical perspective, I think. For example, they assume that R representations should be stronger the more active a habit is, but Bouton and his colleagues have demonstrated -at least behaviorally- that this not necessarily the case.

The authors state that "goal-directed but not habitual control relies on the motivational value of the outcome" (pp. 4). Without wanting to be pedantic, habits do depend on the motivational value of the outcome to develop (unless you take a Guthrian view of S-R contiguity being sufficient to produce them), so I'd rather rephrase this as "only the g-d system is sensitive to changes in the motivational value of the outcome in absence of new experience with it", or something along those lines. This should make it clear that it is the devaluation test what distinguishes between the two types of control.

McNamee and colleagues did a very similar paper using fMRI decoding, but the authors cite this paper briefly, without any reference to what the paper is about and what it found. I think that they should be more detailed about what these other authors did in that paper, discussing it in the introduction and motivating their experiment as an extension of the same idea using a different method (EEG).

Pp 10. "Degraded actions" suggests degraded action-outcome contingency. This is not the manipulation employed in the paper.

Again, without wanting to be pedantic, the authors state that: ""In other words, whereas model-free learning is dependent on the expected reward and hence the outcome that follows a response (Dayan and Berridge, 2014; O'Doherty et al., 2017), habit learning is operationally defined as being independent of the motivational value of the outcome (Adams, 1982; Adams and Dickinson, 1981; Balleine and Dickinson, 1998).", but algorithmic representations of habit learning (and pavlovian conditioning) since the time of Bush and Mosteller and before are based on the notion of prediction-error, that is, they are represented as model-free algorithms. The "expected reward" in MF algorithms is not a psychological, but statistical concept (it turns out that prediction-error learning can be taken as estimating the value of stimuli or states.) What is critical is that these algorithms are not sensitive to changes in outcome value unless the new value is re-experienced and updated, whereas model-based or g-d algorithms are.

*Reviewer #3 (Recommendations for the authors):*

The authors have addressed my main concerns about potential eye-movement artifacts acting as confounds in the decoding analysis, and mostly answered my questions about the preprocessing procedures for the EEG data. In the process of addressing many of the points raised during review, they had had to substantially tamp down some of their claims and provide more context, caveats and speculation regarding explanations for their observations. I don't think this is necessarily a bad thing, as the pattern of data rarely perfectly match our predictions and such deviations from expectation and theory are usually the grist of new research and investigation. Moreover, the main results and their interpretation hold up despite some points that remain a bit muddy (e.g. why the effects of stress occur later in the experiment rather than closer to the time of the stressful event). However, not all of my points were addressed in the initial round of reviews and I have some remaining questions – mostly arising from new analyses in the revision, and would also like additional clarity on some methodological points.

1. In responding to the other reviewers, the authors have added some new analyses of the feedback-related negativity to test if participants in the stress and control groups differed in how the processed error feedback. Interpretation of the null result here seems rather indirect, as it assumes that this ERP would change in response to differences in the sensitivity of participants to error feedback. I suspect that previous work has found such individual differences in groups with greater error sensitivity before, but that work is not cited here. It would be helpful to provide more background on what this ERP is thought to reflect to provide additional context to this result.

2. The FRN in Figure 2 —figure supplements 4-6 looks very different from what I expected, and what I believe, is conventional in the literature. The response on error trials is generally more positive than errors in the 200-300 ms post-feedback window that the authors focus on in this analysis. Generally, I believe this ERP is visible as a more negative deflection on error trials riding on a larger positive response to the feedback stimulus in this time window (e.g. Yeung et al. 2005, Cerebral Cortex). The baseline for error and correct trials also differ substantially in the Fz electrode – and differ quite a bit from zero. The unusual appearance of these ERPs make me somewhat concerned that there might be an error in the analysis.

3. This may have been understandably lost in the long list of comments in the last round of review, but the authors did not respond to my request that they provide more information about the features that contribute to the decoding of the outcome and response – i.e. the particular channels and time points that are contributing to the decoding accuracy measures. While they have convincingly argued that EOG artifacts are unlikely to drive their results, I think it would still be valuable to also see which features are contributing to most to decoding.

4. This is a more minor point, but it would be helpful to have more information about how ICA was used to remove motor and eye-movement activity E.g. how many components were removed, how were they identified and how did the authors verify success of this preprocessing. The current one sentence mention of ICA is not very illuminating about the specific approach used for this study.

[Editors' note: further revisions were suggested prior to acceptance, as described below.]

Thank you for resubmitting your work entitled "Stress diminishes outcome but enhances response representations during instrumental learning" for further consideration by *eLife*. Your revised article has been evaluated by Floris de Lange (Senior Editor) and a Reviewing Editor.

The manuscript has been improved but there are some remaining issues that need to be addressed, as outlined below:

Reviewer 2 is still confused about the intended, claimed, and actual role of overtraining. I think the easiest way to deal with this is to not talk about "extended" training (what does that mean after all – "extended" could mean an hour, days, or months) but instead characterize effects as training-dependent based on devaluation at the beginning, middle or end of a task. You can be clear that you are assessing training-dependent dynamics, but that the extent of training here is quite limited compared to that in the overtraining literature. In other words, training-dependent effects do not necessarily imply overtraining effects – please make that distinction and you should be good.

Of even greater importance is the request by Reviewer 1, that you provide more details about the searchlight analysis.

As always, make sure to detail revisions or rebuttals for each reviewer comment listed below.

*Reviewer #2 (Recommendations for the authors):*

The authors have improved the manuscript. I think it's a much better version than the previous one. They have deleted all those paragraphs that made their arguments and motivation for the experiment confusing.

I'm still super confused about their argument that they are not testing for overtraining effects. I thought overtraining was by definition the amount of training that produced habits. Are they saying in their reply that the effect of stress speeds up habit formation? What is their view on this? If their aim was not to test for "training extension" effects, why are they doing two devaluation manipulations?

I don't think is enough to add this paragraph in light of my comment:

"Although we did not aim to test training-induced changes in the balance of outcome and response processing per se, we included transient outcome devaluations after initial, moderate, and extended training in the instrumental learning task."

And then, a few paragraphs later, saying the following:

"To assess whether the balance of goal-directed and habitual behaviour and its modulation by stress depends on the extent of training, we presented devaluation blocks early during training, after moderate training, and again after extended training at the end of the task."

And, having a section called "Stress boosts response representations after extended training."

As a reader, I'm extremely confused about these apparent contradictions. I'd love to see in the introduction a more precise paragraph where their expectations are clearly mentioned.

This comment has been carrying over since the first review when I made the following comment:

"It was not clear to me what the main hypothesis of the study was. The authors seem to talk rather loosely about habits being developed after overtraining versus the mediating effect of stress on habits. The introduction should convey their main goal more clearly."

Sorry if I'm being too stupid, but it's not clear to me why they are using training extension and stress to test devaluation sensitivity and outcome/response representations if their aim was not to overtrain participants.

*Reviewer #3 (Recommendations for the authors):*

The authors have satisfactorily responded to my previous major concerns. I only have one minor outstanding request: Please provide more information on how the searchlight analysis was carried out (e.g. how many electrodes were included in each 'searchlight', how electrodes were grouped together in this analysis).

---

## [Author Response]

Essential revisions:(1) Several clarifications must be made to address the possibility that saccadic eye movements could be acting as a confound for some of the EEG decoding results, including an account of which features were most important for classification performance, and how eye-movements and blinks were dealt with in preprocessing.

We appreciate the opportunity to clarify this issue. First of all, the classifier applies those electrophysiological properties to the reinforcement learning task (RT) that had been identified in the delayed-matching-to-sample (DMS) task for the separation of object vs scene and blue vs red symbols. Artefacts that occur only during the reinforcement task will simply be ignored as they did not constitute discriminative features in the training data set and are therefore not part of the classifier (Grootswagers et al., 2017). Thus, artefacts would have affected the classification only when the same artefact occurred in both the DMS and RT, which is rather unlikely. Furthermore, the training DMS data were collected in two sessions, of which one was completed before participants underwent the stress induction and another after the completion of the RT. Importantly, the classifier performance did not differ between these DMS sessions and was not modulated by the experimental treatment. Consequently, we think that (i) group-specific artefacts and (ii) emotional and motivational confounds that could have arisen from the RT completion, such as fatigue, cannot account for our decoding results. Moreover, the positions of the symbols of the aliens on the computer screen were fully randomized. Thus, anticipatory eye movements, such as saccades during the response phase, could not drive the EEG findings in the stress group at the end of the task. Apart from the methodological principles of the EEG decoding and our specific task design, there is also evidence that classifiers have the capacity to ignore bad channels or suppress noise during training, making artefact correction, in general, less critical in decoding analyses (Grootswagers et al., 2017). Together, we think it is highly unlikely that artefacts related, for example to saccades or blinks, biased our decoding results.

For the ERP analysis, we removed blinks and eye movements via independent component analysis. We added this information now in the manuscript, please see page 45, lines 1024 to 1025:

“In addition, we removed muscular artefacts and corrected for blinks and eye movements via independent component analysis.”

Finally, we used an eyetracker during the EEG measurement to control specifically for eye movements and these data showed that there were no significant changes in saccades or blinks across the task or depending on the trial type, and no modulation thereof by the stress manipulation (outcome devaluation × stimulus value × time × group: *F*(4,196) = 0.78, *P* = 0.54, *ƞ_p_²* = 0.02, 95% CI = 0.002 to 0.008; outcome devaluation × stimulus value × group: *F*(2,98) = 1.03, *P* = 0.36, *ƞ_p_²* = 0.02, 95% CI = 0.005 to 0.020). We have added this information to the text. Please see page 14, lines 532 to 537:

“Furthermore, we recorded eye tacking data to control for potential group differences in saccades or eye blinks. These control analyses showed that there were no significant group differences in saccades or eye blinks across the task or depending on the trial type (outcome devaluation × stimulus value × time × group: *F*(4,196) = 0.78, *P* = 0.54, *ƞ_p_²* = 0.02, 95% CI = 0.002 to 0.008; outcome devaluation × stimulus value × group: *F*(2,98) = 1.03, *P* = 0.36, *ƞ_p_²* = 0.02, 95% CI = 0.005 to 0.020; see supplementary file 1D).”

Furthermore, we added a brief section on the eyetracking to the methods section. Please see page 42, lines 959 to 964:

“Eyetracking. We used a desktop mounted eye-tracker (EyeLink 1000; SR-Research Ltd., Mississauga, Ontario, Canada) to record monocular eye movements from the left eye at a sampling rate of 500Hz. We used custom scripts implemented in MATLAB (The Mathworks, Natick, MA) to extract the mean saccades and blink information depending on the stimulus value, outcome devaluation and time (zero to 2000 ms around the stimulus onset). Data for two participants were missing due to a failure of the used eye-tracker.”

(2) It appears that devaluation insensitivity, and reduced neural outcome representations, emerge at a point when stress measures no longer indicate a stress state. Please clarify how the time courses of these measures relate to the claims about an influence of stress on devaluation insensitivity.

It is important to note that stress effects are not necessarily over when acute levels of stress mediators returned to baseline. Specifically, several major stress mediators are known to have ‘after-effects’ that outlast acute elevations of these measures (Joëls and Baram, 2009). For example, glucocorticoids are assumed to act as a “two-stage rocket”, with rapid, non-genomic actions and delayed, genomic actions (Joëls et al., 2012). The latter genomic actions typically set in well after acute glucocorticoid elevations vanished (Kloet et al., 2008). Thus, the fact that autonomic measures and cortisol levels returned to baseline during the learning task, does not imply that the stress system activation had been over at this point. Moreover, acutely elevated stress mediators may have affected early learning processes in a way that became apparent only as training progressed.

We discuss now explicitly the fact that devaluation insensitivity, reduced neural outcome representations and increased neural response representations emerge at a point when stress parameters were not acutely elevated anymore and provide potential explanations for this pattern on page 27, lines 596 to 607:

“Importantly, at the time when stress effects on devaluation sensitivity and outcome representation were observed, stress mediators were no longer elevated anymore. However, it is important to note that stress effects are not necessarily over when acute levels of stress mediators returned to baseline. Specifically, several major stress mediators are known to have ‘after-effects’ that outlast acute elevations of these measures (Joëls and Baram, 2009). For example, glucocorticoids are assumed to act as a “two-stage rocket”, with rapid, non-genomic actions and delayed, genomic actions (Joëls et al., 2012). The latter genomic actions typically set in well after acute glucocorticoid elevations vanished (Kloet et al., 2008). Thus, the fact that autonomic measures and cortisol levels returned to baseline during the learning task, does not imply that the stress system activation had been over at this point. Moreover, acutely elevated stress mediators may have affected early learning processes in a way that became apparent only as training progressed.”

(3) The proposed role of extended training is consistent with the fact that significant devaluation insensitivity and reduced outcome representations emerge in the last block in the stress group. However, the notion that habits develop with extended training seems inconsistent with the apparent decrease in response representation and increase in outcome representation across blocks in the control group, as well as the apparent increase in devaluation sensitivity in the stress group across blocks 1 and 2. These inconsistencies must be resolved and claims modified accordingly.

In the stress group, we did not find reliable evidence for an increase of devaluation sensitivity from block 1 to block 2 (block × treatment interaction: *F*(1,56) = 1.575, *P* = 0.215, *ƞ_p_²* = 0.027, 95% CI = -0.013 to 0.052). In addition, groups did not differ during this second block (*t*_56_ = 0.165, *P* = 0.870, Cohen’s *d* = 0.043, -0.472 to 0.558). We have added these statistics to the text, please see page 15, lines 327 to 334:

“Furthermore, while there was no evidence for an interaction of devaluation block number (1 vs. 2) and experimental treatment (*F*(1,56) = 1.575, *P* = 0.215, *ƞ_p_²* = 0.027, 95% CI = -0.013 to 0.052) when analysing the changes from the first to the second devaluation block, we obtained a significant interaction between block (2 vs. 3) and treatment when analysing the changes from block 2 to block 3 (*F*(1,56) = 13.589, *P* < 0.001, *ƞ_p_²* = 0.195, 95% CI = 0.105 to 0.319). Moreover, follow-up tests showed that after moderate training intensity (i.e. in block 2), groups did not differ in responses for the devalued outcome (*t*_56_ = 0.165, *P* = 0.870, Cohen’s *d* = 0.043, -0.472 to 0.558).”

However, we completely agree that the observed decrease in neural response representations and the increase in neural outcome representations in the control group speak against a training-induced bias towards habits (previously referred to as ‘overtraining effect’) in control participants. Importantly, however, we did not aim to assess an overtraining effect per se in the present study and this study was not designed to test for such a general overtraining effect (e.g., we might have needed an even longer or repeated training). Instead, we aimed to test how stress affects specific neural signatures (derived from EEG-based decoding) of goal directed and habit behaviour, respectively, and to what extent such stress effects may depend on the extent of training. The fact that there was no overtraining effect in the control group of the present study, does not question the training-dependency of the observed stress effect. Our idea that stress might accelerate a shift from goal-directed to habitual behaviour is based on previous rodent studies that did show that stress (hormones) accelerate a shift from ‘cognitive’ to ‘habitual’ responses that would otherwise only occur after extended training (Packard, 1999; Siller-Pérez et al., 2017). We agree that based on our own data, this idea remains rather speculative as we do not see evidence of a training-dependent bias towards habitual behavior/processing in control participants across the task.

We have now clarified the objectives of our study, emphasizing that this study did not aim to test the impact of training intensity on the control of instrumental behaviour per se. Please see page 5, lines 116 and 117:

“Although we did not aim to test training-induced changes in the balance of goal-directed and habitual processing per se […]”

Moreover, we state now explicitly that the idea that stress might accelerate a shift from goal directed to habitual control is based on respective animal data but that this idea remains speculative here as we did not find evidence for a training-related bias from goal-directed to habitual control in control participants (but even initial evidence against this notion). Please see page 30, lines 660 to 667:

“Findings on ‘cognitive’ and ‘habitual’ forms of navigational learning in rats, however, demonstrated that stress hormones may accelerate a shift from ‘cognitive’ to ‘habit’ learning that would otherwise only occur after extended training (Packard, 1999; Siller-Pérez et al., 2017). Thus, it is tempting to hypothesize that a similar mechanism might be at work during instrumental learning in stressed humans. This conclusion, however, remains rather speculative as we did not observe a training-dependent shift towards habits in the control group and this group even showed reduced response and increased outcome representations over time, which rather suggests increased goaldirected processing across the task.”

(4) An important aspect of assessing devaluation sensitivity in the animal literature is a reinforced test, in which the devalued outcome is delivered contingent on the relevant response – as a result, habitual animals rapidly reduce performance of the devalued action. Such tests rule out a general lack of attention, response perseveration etc. The absence of a reinforced test should be explicitly addressed and the implications discussed.

We agree that a reinforced test, in which the devalued outcome is delivered contingent on the relevant responses, is of special importance for the assessment of devaluation sensitivity and the exclusion of unspecific effects in animal studies. In the present human study, the feedback provided on each trial varied between blocks. During NoDev blocks, participants saw the value of the card that was gained before as well as the corresponding response costs. In Dev blocks, however, this information was masked for all trials. This procedure is comparable to previous studies in humans in which the devalued outcome was not presented during the critical test trials (e.g. Schwabe and Wolf, 2009; Valentin et al., 2007). Nevertheless, we agree that this kind of experience with the devalued outcome differs from procedures utilised in the animal literature. Importantly, however, we additionally included consumption trials in all of the blocks, which enabled us to rule out unspecific factors, such as a general lack of attention, altered contingency knowledge or response perseveration. In these consumption trials, participants clearly preferred the high-valued cards over low-valued cards during NoDev blocks as well as the valued card over its devalued counterpart during Dev blocks, irrespective of stress. This finding demonstrates that participants of both groups were aware of the value of the card stimuli in a specific block but that responses of stressed participants were less guided by this knowledge about the value of the outcome engendered by the response. The specific response pattern in the consumption trials during devalued blocks also rules out general attentional or motivational deficits.

We discuss this deviation of the present paradigm from the typical devaluation procedure in animal studies and its implications now on page 30, lines 671 to 688:

“In animal studies, devaluation sensitivity is usually assessed by means of a reinforcement test, in which the devalued outcome is delivered contingent on the relevant response. This procedure allows to control, for example, for a general lack of attention. In the present human study, the feedback provided on each trial varied between blocks. During NoDev blocks, participants saw the value of the card that was gained before as well as the corresponding response costs. In Dev blocks, however, this information was masked for all trials. This procedure is comparable to previous studies in humans in which the devalued outcome was not presented during the critical test trials (e.g. Schwabe and Wolf, 2009; Valentin et al., 2007). Importantly, however, we additionally included consumption trials in all of the blocks, which enabled us to rule out unspecific factors, such as a general lack of attention, altered contingency knowledge or response perseveration. In these consumption trials, participants clearly preferred the high-valued cards over low-valued cards during NoDev blocks as well as the valued card over its devalued counterpart during Dev blocks, irrespective of stress. This finding demonstrates that participants of both groups were aware of the value of the card stimuli in a specific block but that responses of stressed participants were less guided by this knowledge about the value of the outcome engendered by the response. In addition, the specific response pattern in the consumption trials during devalued blocks also rules out general attentional or motivational deficits.”

(5) It is unclear from Figure 1 whether the "Error" feedback screen was employed during devaluation blocks. If so, please address the potential influence of negative social feedback on avoidance of the non-devalued action, particularly in the group that had just been subjected to a social stress test.

The error-feedback screen was presented throughout the entire task (i.e. an error message appeared also during the devaluation blocks). We completely agree that this is an important aspect of our paradigm and we make this feature now more explicit, both in the text (please see page 40, lines 894 and 895) and in the revised Figure 1. While we cannot completely rule out that this form of feedback somehow interacted with our social stress manipulation, we consider it rather unlikely that the “error” feedback screen had a major impact on our results. With respect to our neural decoding data it is important to note that we decoded neural response and outcome representations only during NoDev blocks, in which “error” feedback was extremely rare (only 5.4 % of all NoDev trials) and its frequency did not differ between groups (t_56_ = 0.898, P = 0.373, d = 0.235, 95% CI = -0.282 to 0.752). With respect to our behavioural data, it is important to note that the responses for devalued outcomes were directly correlated with the neural response and outcome representations, respectively, which again could not be biased by differences in error feedback processing.

Moreover, in response to the reviewers’ comment, we analysed now participants’ error related negativity as a neural measure of (negative) feedback processing. If the social stress exerted its effect primarily via an avoidance of negative feedback, then stress should have also modulated negative feedback processing. Our analysis of the error-related potentials, however, showed no such effect. We have added this finding to our Results section. Please see also page 23, lines 508 to 518:

“No group differences in feedback processing

The error-feedback screen was presented throughout the entire task. Thus, an error message appeared also during the devaluation blocks. To test whether this form of feedback interacted with our social stress manipulation (e.g. that stressed participants continued in responding towards devalued outcomes in order to avoid the presentation of the error-feedback screen), we analysed participants’ error-related potentials as a neural measure of feedback processing. If the social stress exerted its effect primarily via an avoidance of negative feedback, then stress should have also modulated feedback processing. Our analysis of the error related potentials, however, showed no such effect (outcome devaluation × correctness × group for Cz, FCz and Fz: all F(2,54) < 1.176, all P > 0.316, all ƞ_p_² < 0.042, all 95% CI = 0.010 to 0.039, Figure 2 —figure supplements 4-6).” Please see page 46, lines 1048 to 1054 for methodological details:

“Feedback related potentials. Artefact-free epochs were averaged participant- and conditionwise with respect of outcome devaluation block and correctness (noDev correct, noDev error, Dev O^high^ correct, Dev O^high^ error, Dev O^low^ correct and Dev O^low^ error). Thereafter, mean amplitudes were segmented from 200 to 300 ms relative to feedback onset at midline electrodes Cz, FCz and Fz (Hajcak et al., 2006; Pfabigan et al., 2011; Pfabigan et al., 2015). For statistical analysis, we applied ANOVAs with the within-subject factors outcome devaluation and correctness (correct and error) for Cz, FCz and Fz.”

Moreover, we have added a paragraph to the discussion in which we address the potential role of negative social feedback on the performance in devalued blocks. Please see page 31, lines 689 to 704:

“It is important to note that participants received an error feedback in devalued trials when they chose the response option that was not associated with the now devalued outcome. Given that acute stress may increase sensitivity to social cues (Domes and Zimmer, 2019), one might argue that stressed participants continued in responding towards devalued outcomes in order to avoid the presentation of the error-feedback screen. We consider this alternative, however, to be rather unlikely. First, with respect to the neural outcome and response representations, these were analysed during NoDev blocks, in which groups did not differ in their behavioural performance accuracy and consequently not in the frequency of error feedback. Furthermore, participants’ performance in devalued blocks was directly associated with the observed changes in neural representations during the NoDev blocks. Finally, if stress exerted its effect through an avoidance of negative feedback, then stress should have also altered neural feedback processing. To directly test this alternative, we analysed the error-related activity, a well-known neural correlate of (negative) feedback processing (Falkenstein et al., 2000; Holroyd and Coles, 2002). Our results, however, showed no group differences in the error-related potentials, which appears to speak against altered negative feedback processing as the major driving force of the obtained results.”

(6) A range of statistical approaches are employed: please provide a section that details and justifies each.

We appreciate the opportunity to describe and justify in more detail the range of statistical approaches that we employed in order to analyse the impact of acute stress on goal-directed and habitual processing during instrumental learning. As suggested, we have added a new paragraph in the Results section in which we justify the range of the different statistical approaches. Please see page 10, lines 198 to 227:

“In order to adequately assess whether increased responding to devalued or degraded actions is due to reduced goal-directed or enhanced habitual control (or both), we employed a variety of measures and statistical approaches. At the behavioural level, we applied a devaluation paradigm to examine whether stressed participants respond differently to transient outcome devaluations compared to control participants and to assess whether the predicted neural changes are linked to behavioural outcomes. Behavioural analyses focussed mainly on participants’ choice in valued and devalued trials. Based on previous animal studies (Dias-Ferreira et al., 2009), we additionally investigated whether the effects of stress on the control mode would depend on the amount of training and therefore implemented outcome devaluation after initial, moderate and extended training. At the physiological level, we analysed pulse, systolic and diastolic blood pressure as well as salivary cortisol concentrations at different time points across the experiment to assess the effectiveness of the stress manipulation. At the neural level, we mainly focused on the decoding of outcome and response representations, which provide insights into the mechanisms through which stress affects the sensitivity to outcome devaluation, the primary objective of the present study. Outcome representations are indicative for goal-directed processing and response representations rather point to habitual processing. To further address the important question whether response and outcome representations represent signatures of distinct control systems, we additionally analysed the correlations between both neural representations. For this purpose, we used Bayesian correlational analyses. Depending on the magnitude of the Bayes factor (reflecting the likelihood ratio of the data under the alternative hypothesis and the data under the null hypothesis), the Bayesian approach can provide evidence in favor of the alternative hypothesis or evidence in favor of the null (Hoijtink, 2012; Kass and Raftery, 1995; Nuzzo, 2017). Thus, we utilized Bayesian analyses to provide clear evidence for or against the null hypothesis. To further assess the association between the behavioural data (i.e. classification accuracy) and the strength of the neural representation (i.e. classification accuracies), we also computed Spearman correlations. To additionally analyse previously proposed “attentional habits” (Luque et al., 2017) on the one hand, and to address the alternative explanation of altered negative feedback processing on the other hand we analysed event-related potentials.”

Please see also page 48, lines 1105 to X1115:

“Statistical analyses of the EEG decoding data were performed at the group level averaging across individual decoding accuracies. Decoding accuracies were analysed by mixed-design ANOVAs with the within-subject factor block (1-6, 7-12, 13-18 and 19-24 block) and the between-subject factor group. In addition, we computed Spearman correlations between ∆ classification scores (averaged classification accuracy during the first six blocks minus the classification accuracy during the last six blocks) and ∆ responses for devalued outcomes for Dev O^high^ blocks responses for devalued outcomes during the first devaluation block, i.e. the 2^nd^ overall block, minus responses for devalued outcomes during the last devaluation block, i.e. the 29^th^ overall block. Bayesian correlation analyses were conducted using JASP version 0.13 software (https:/jasp-stats.org/) and the default Cauchy prior 0.707.”

Reviewer #1 (Recommendations for the authors):(1) The devaluation, but not accuracy data, is clearly bimodal, in both the control and stress groups. I would like to see this discussed in more detail, in terms of consistency with physiological and neural measures, as well as in terms of broader theoretical implications.

We thank the reviewer for highlighting this important issue. Indeed, behavioural accuracy data are bimodally distributed. This suggests that participants either mainly selected their action based on the outcome value, i.e., did avoid the devalued outcome and thus were goal-directed, or responded to the devalued outcome in the vast majority of the trials and thus behaved habitually. Thus, participants seemed to show an “either-or” pattern and there seemed to be interindividual differences in the tendency to perform in a goal-directed vs. habitual manner. Interestingly, also among stressed participants, there were substantial individual differences in the propensity to shift from goal-directed towards habitual behaviour, indicating that there is no overall effect of stress on the control of behaviour but that individuals differ in their sensitivity to these stress effects. This raises the important question of what makes individuals more or less vulnerable to stress effects on instrumental control (and cognition in general). Previous work suggested that genetic variants related to major stress response systems (i.e. noradrenergic and glucocorticoid activity) may play an important role in how stress affects the engagement of multiple learning systems (Wirz et al., 2017; Wirz et al., 2018). Understanding which further individual factors contribute to the interindividual propensity to shift from goal-directed to habitual behaviour needs to be addressed by future research.

The fact that our neural data were not bimodal suggests that changes in the neural representation translate not directly into behavioural changes. The observed significant correlations between neural representation and performance in devalued trials show that there is a link. However, the correlations were obviously clearly below 1. Compared to the behavioural level which included only discrete yes-no responses, the neural data may have been much more sensitive and able to capture more fine-grained changes. The different sensitivity of behavioural and neural data is interesting in itself and points to another important question for future research: how do neural representation changes and behavioural changes relate to each other? Is there a particular threshold at which a change in representation at which behaviour changes?

As suggested by the reviewer, we make the bimodal distribution of the behavioural responses in the devaluation blocks more explicit now and discuss the theoretical implications of this bimodal distribution and its link to the neural data in a separate paragraph of the discussion now. Please see page 32, lines 708 to 742:

“Interestingly, participants either mainly selected their action based on the outcome value, i.e. did avoid the devalued outcome and thus were goal-directed value, or responded to the devalued outcome in the vast majority of the trials and thus behaved habitually. Thus, participants showed an “either-or” pattern and there seemed to be interindividual differences in the tendency to perform in a goal-directed vs. habitual manner. Interestingly, also among stressed participants, there are substantial individual differences in the propensity to shift from goal-directed towards habitual behaviour, indicating that there is no overall effect of stress on the control of behaviour but that individuals differ in their sensitivity in these stress effects. This raises the important question of what makes individuals more or less vulnerable to stress effects on instrumental control (and cognition in general). Previous work suggested that genetic variants related to major stress response systems (i.e. noradrenergic and glucocorticoid activity) may an important role in which of multiple learning systems is engaged after stress (Wirz et al., 2017; Wirz et al., 2018). Understanding which further individual factors contribute to the interindividual propensity to shift from goal-directed to habitual behaviour needs to be addressed by future research. Furthermore, the fact that our neural data were not bimodal suggests that changes in the neural representation may not translate directly into behavioural changes. The observed significant correlations between neural representation and performance in devalued trials show that there is a link between the behavioural and neural representation level. However, the correlations were obviously far below 1. Compared to the behavioural level which included only discrete yes-no responses, the neural data may have been much more sensitive and able to capture more fine-grained changes. The different sensitivity of behavioural and neural data is interesting in itself and points to another important question for future research: how do neural representation changes and behavioural changes relate to each other? Is there a particular threshold at which a change in representation triggers behavioural changes?”

(2) It looks like the response representation is actually decreasing across blocks in the control group. This seems inconsistent with the general argument that habits should develop as training progresses.

Participants in the control group showed indeed a reduced response representation in the second half of the task (t_24_ = 3.50, P = 0.002, d = 0.701, 95% CI = 0.256 to 1.134). We completely agree that this finding – as well as the fact that outcome representations did not decrease in controls across the task – speaks against a training-dependent shift towards habits. It is, however, important to note that we did not aim to assess an overtraining effect per se in the present study and that the study was not designed to test for such a general overtraining effect, but to test the impact of stress on the neural signature (identified vie EEG-based decoding) of goal-directed and habit processes, respectively, and to what extent this impact depended on the amount of training.

Our idea that stress might accelerate the shift from goal-directed to habit learning that would otherwise only occur after extended training is based on rodent studies showing exactly this pattern for stress hormone effects on the balance of ‘cognitive’ and ‘habitual’ learning (Packard, 1999; Siller-Pérez et al., 2017). We make now clearer that our argument is mainly based on these rodent data but remains speculative based on the present data as we did not see any training-dependent bias towards habit behaviour or processing in the control group across the task. In addition, we state now explicitly, that the pattern observed in the control group speaks against a training-dependent shift towards habits.

Please see page 30, lines 660 to 667:

“Findings on ‘cognitive’ and ‘habitual’ forms of navigational learning in rats, however, demonstrated that stress hormones may accelerate a shift from ‘cognitive’ to ‘habit’ learning that would otherwise only occur after extended training (Packard, 1999; Siller-Pérez et al., 2017). Thus, it is tempting to hypothesize that a similar mechanism might be at work during instrumental learning in stressed humans. This conclusion, however, remains rather speculative as we did not observe a training-dependent shift towards habits in the control group and this group even showed reduced response and increased outcome representations over time, which rather suggests increased goaldirected processing across the task.”

(3) "…while model-free learning can be present after only few trials, establishing habit behaviour requires extensive training." This statement is problematic for several reasons. First, it appears to be contradicted in the very next paragraph, where the authors acknowledge that the necessity and sufficiency of extensive training for habit expression in human subjects is controversial. Second, in terms of operational definitions, habits are instrumental responses that are insensitive to outcome devaluation, whether extensively trained or not. Likewise, theoretically, habits are usually attributed to the strength of the stimulus-response association, which may depend on repetition, but also on the magnitude of the reward, as well as, presumably, a range of other factors (e.g., stimulus salience).

We completely that this statement has been an oversimplification. “Extensive training” was too far-reaching and we further agree that repetition may be one factor among others that is important for the strength of the stimulus-response association. Therefore, we have rephrased this sentence accordingly (please see page 28, lines 633 to 637):

“For instance, while model-free learning can be present after only few trials, establishing habit behaviour relies on the strength of the stimulus-response association, which among other factors such as reward magnitude or stimulus saliency, has been suggested to depend on repetition (Adams, 1982; Dickinson et al., 1995; Tricomi et al., 2009; but see Wit et al., 2018).”

(4) Given the schematic of the goal-directed system (top of Figure 1A), it's a bit strange that outcome and response representations would be uncorrelated.

The goal-directed system might also involve a response representation at the time of stimulus presentation as it is supposed to learn S-R-O associations. However, outcome representations should prevail as action-outcome learning is at the heart of the goal-directed system and the stimulus association may play a rather minor role. Further the stimulus representation may serve a different purpose in the goal-directed system, where it is mainly used as an indicator of the correct action required to achieve a certain outcome, which is distinct from the prompting of a specific “reflexive” response in habitual S-R learning. The latter should be particularly distinct from the processes indicated by the outcome representation. Moreover, it is important to note that we decoded the response representation at the time point of the stimulus presentation and at this time point, the response representation is certainly more relevant for the habit system that relies on S-R associations. We aimed to indicate the differential relevance of the response representation for the goal-directed and habits systems by the different size of the respective bubbles. We have added now to the legend of Figure 1 that the larger bubbles are supposed to reflect the relevance of the respective representations for the two systems. Moreover, we mention now in the text that a stimulus representation might be involved in ‘goal-directed’ S-R-O learning and in ‘habitual’ S-R learning, however, presumably with a different function and a different relevance for the actual learning process. Please see page 8, lines 170 to 177:

“Note that the goal-directed system might also involve a response representation at the time of stimulus presentation as it is supposed to learn S-R-O associations. However, outcome representations should prevail as action-outcome learning is at the heart of the goal-directed system and the stimulus association may play a minor role. Further, the stimulus representation may serve a different purpose in the goal-directed system, where it is mainly used as an indicator of the correct action required to achieve a certain outcome, which is distinct from the prompting of a specific “reflexive” response in habitual S-R learning.”

(5) The violin plots are great for illustrating the bimodality in devaluation sensitivity, but it would also be useful to see all of these related variables plotted in the same way, preferably in a single figure. Please include line graphs showing the degree of responding for the devalued outcome, the decoding accuracy for responses and outcomes, and the physiological stress measures, across groups, in a way that allows for a visual, side-by-side, comparison.

We thank the reviewer for this constructive suggestion. We modified the former Figure 4 as suggested and illustrate the physiological stress measurements, the proportion of responses for devalued outcomes, and the decoding accuracy for the outcome representation during stimulus presentation and response choice as well as response representation during stimulus presentation in a single figure (now Figure 2); in addition to the violin plots (shown in Figure 3).

(6) Please include graphics of devalued trials in Figure 1C, to illustrate the masking of outcomes and the nature of the error feedback on such trials.

We have modified Figure 1C (and the respective legend) now to illustrate the masking of the outcomes and the nature of the error feedback in the devalued trials.

Please see page 9.

(7) In the Results section, following the description of the task and procedures, but prior to reporting any results, please add a section detailing and justifying the range of statistical approaches. For example, some reported p-values for t-tests appear to be uncorrected (e.g., line 358), as are all p-values for correlation tests. Moreover, a Bayesian analysis is used to assess evidence against a correlation, but not evidence in favor of a correlation.

First of all, we unintentionally reported one uncorrected p-value. We have corrected this mistake and present now the Bonferroni-corrected value, which does however not affect our conclusions (i.e. the significance remains). Please see page 20, lines 435 to 439:

“When we analysed the outcome representations at the time of the choice between the two aliens, a very similar pattern emerged: at the end of the task, participants in the stress group showed a decreased outcome representation at the time point of the choice (t_22_ = 2.94, P_corr_ = 0.016, d = 0.613, 95% CI = 0.161 to 1.054), whereas there was no such effect in the control group (t_17_ = 0.59, P_corr_ = 1, d = 0.138, 95% CI = -0.328 to 0.600).”

We conducted Bayesian correlation analyses between the classification accuracies to test whether outcome and response representations reflected independent or linked neural representations. Depending on the magnitude of the Bayes factor (reflecting the likelihood ratio of the data under the alternative hypothesis and the data under the null hypothesis) this approach can provide evidence in favor of the alternative hypothesis as well as evidence in favor of the null hypothesis (Hoijtink, 2012; Kass and Raftery, 1995; Nuzzo, 2017). In other words, this analysis does not selectively test for evidence against a correlation but may provide evidence for or against a correlation (our data provide evidence against a correlation).

In response to the reviewer’s suggestion, we further provide additional information detailing and justifying the range of statistical approaches. Please see page 10, lines 198 to 227:

“In order to adequately assess whether increased responding to devalued or degraded actions is due to reduced goal-directed or enhanced habitual control (or both), we employed a variety of measures and statistical approaches. At the behavioural level, we applied a devaluation paradigm to examine whether stressed participants respond differently to transient outcome devaluations compared to control participants and to assess whether the predicted neural changes are linked to behavioural outcomes. Behavioural analyses focussed mainly on participants’ choice in valued and devalued trials. Based on previous animal studies (Dias-Ferreira et al., 2009), we additionally investigated whether the effects of stress on the control mode would depend on the amount of training and therefore implemented outcome devaluation after initial, moderate and extended training. At the physiological level, we analysed pulse, systolic and diastolic blood pressure as well as salivary cortisol concentrations at different time points across the experiment to assess the effectiveness of the stress manipulation. At the neural level, we mainly focused on the decoding of outcome and response representations, which provide insights into the mechanisms through which stress affects the sensitivity to outcome devaluation, the primary objective of the present study. Outcome representations are indicative for goal-directed processing and response representations rather point to habitual processing. To further address the important question whether response and outcome representations represent signatures of distinct control systems, we additionally analysed the correlations between both neural representations. For this purpose, we used Bayesian correlational analyses. Depending on the magnitude of the Bayes factor (reflecting the likelihood ratio of the data under the alternative hypothesis and the data under the null hypothesis), the Bayesian approach can provide evidence in favor of the alternative hypothesis or evidence in favor of the null (Hoijtink, 2012; Kass and Raftery, 1995; Nuzzo, 2017). Thus, we utilized Bayesian analyses to provide clear evidence for or against the null hypothesis. To further assess the association between the behavioural data (i.e. classification accuracy) and the strength of the neural representation (i.e. classification accuracies), we also computed Spearman correlations. To additionally analyse previously proposed “attentional habits” (Luque et al., 2017) on the one hand, and to address the alternative explanation of altered negative feedback processing on the other hand we analysed event-related potentials.”

Please see also page 48, lines 1105 to 1115:

“Statistical analyses of the EEG decoding data were performed at the group level averaging across individual decoding accuracies. Decoding accuracies were analysed by mixed-design ANOVAs with the within-subject factor block (1-6, 7-12, 13-18 and 19-24 block) and the between-subject factor group. In addition, we computed Spearman correlations between ∆ classification scores (averaged classification accuracy during the first six blocks minus the classification accuracy during the last six blocks) and ∆ responses for devalued outcomes for Dev O^high^ blocks responses for devalued outcomes during the first devaluation block, i.e. the 2^nd^ overall block, minus responses for devalued outcomes during the last devaluation block, i.e. the 29^th^ overall block. Bayesian correlation analyses were conducted using JASP version 0.13 software (https:/jasp-stats.org/) and the default Cauchy prior 0.707.”

(8) The phrase "responses to devalued actions", employed throughout, is a bit confusing. I would change this to "responses for devalued outcomes".

We agree and have rephrased the referring phrase accordingly, throughout the manuscript. Please see, for instance, page 13, lines 265 and 266:

“Proportion of responses for devalued outcomes across the reinforcement learning task during Dev O^high^ blocks.”

Or page 21, line 450:

“Correlations of outcome and response representation during stimulus presentation with responses for devalued outcomes during Dev O^high^ blocks.”

(9) I'm guessing that Figure 3 is showing proportions, not percentages.

Thank you for noticing this mistake, which has now been corrected. Please see the new y-axis label of Figure 3 (page 16):

We changed the labelling of the Figure 3—figure supplement 1 and Figure 3—figure supplement 2 accordingly (please see Figure 3—figure supplements 1 and 2).

Reviewer #2 (Recommendations for the authors):It was not clear to me what the main hypothesis of the study was. The authors seem to talk rather loosely about habits being developed after overtraining versus the mediating effect of stress on habits. The introduction should convey their main goal more clearly.

Thanks for the opportunity to clarify our main objective and hypothesis. Previous research showed that stress may induce a shift from goal-directed to habit behaviour, yet it remained unclear whether this is due to increased habit behaviour, decreased goal-directed behaviour or both. Existing tasks did not allow a distinction between these alternatives, nor a distinction between goal-directed and habitual contributions to behaviour in general. To overcome these methodological limitations and examine whether stress leads to an upregulation of habitual processing or a downregulation of goal-directed processing (or both), we used EEG-based decoding to identify neural signatures of the goal-directed and habit system, respectively, and their modulation by stress. Thus, our main goal was to investigate whether stress leads to an increase in habitual processing, a decrease in goal-directed processing or both – and to link these neural changes to participants’ actual responses for devalued outcomes. Based on previous rodent studies (Dias-Ferreira et al., 2009; Packard, 1999; Siller-Pérez et al., 2017), we further asked whether the stress effect on goal-directed and habitual processing, respectively, would depend on the extent of training. We did not aim, however, to assess overtraining effects per se (i.e. independent of stress). We have clarified the rationale of our study now. Please see page 5, lines 97 to 101:

“Here, we aimed to overcome these shortcomings of classical paradigms for the assessment of the mode of behavioural control and to examine whether stress leads to an upregulation of habitual processing or a downregulation of goal-directed processing (or both). To these ends, we leveraged electroencephalography (EEG) in combination with multivariate pattern analysis (MVPA)-based decoding of neural representations.”

Please see also page 5, lines 116 to 121:

“Because previous rodent studies suggested that stress or stress hormones modulate the balance of goal-directed and habitual forms of learning after more extended training (Dias-Ferreira et al., 2009; Packard, 1999; Siller-Pérez et al., 2017), we further asked whether the stress effect on goal-directed and habitual processing, respectively, would depend on the extent of training. Although we did not aim to test training-induced changes in the balance of goal-directed and habitual processing per se, we therefore included transient outcome devaluations after initial, moderate, and extended training in the instrumental learning task to assess if and how the predicted changes in neural representations are linked to behavioural manifestations of stress-induced changes in behavioural control.”

Furthermore, we state the main goal of this study also explicitly on page 6, lines 123 to 126:

“The goal of this study was to elucidate the mechanisms underlying the impact of stress on the control of instrumental behaviour. Specifically, we aimed to leverage an EEG-based decoding approach to determine stress-induced changes in outcome and response representations that constitute key distinguishing features of goal-directed action and habitual responding.”

In my view, this task is not meant to test for overtraining effects as it includes repeated devaluation of the outcome within the same subject, which makes it very difficult to assess the extent to which overtraining -rather than simply lack of motivation at the end of the task or other factor- is explaining weaker devaluation effects late in training. This is a critique for any experiment looking at overtraining effects, bit it should be at least demonstrated that what is driving the effect found by the authors is indeed an effect of overtraining by looking at more than one extension of training. Did the authors test other extensions of training in pilot studies?

We completely agree that this study was not designed to test overtraining effects and we did not intend to do so. We state this now more explicitly in the manuscript. Please see, for example, page 5, lines 116 and 117:

“Although we did not aim to test training-induced changes in the balance of goal-directed and habitual processing per se […]”

As we did not intend to test for overtraining effects per se, we did not test other extensions of training.

In terms of alternative explanations for the devaluation effects at the end of training (e.g. lack of motivation), we think that we can rule out many of these. First, participants’ reaction times became faster throughout the task (without a reduction in response accuracy in NoDev trials, thus arguing against an altered speed-accuracy tradeoff), which speaks against lack of motivation or general fatigue. Second, participants continued to perform the valued responses with high accuracy even after more extended training and without relevant changes across blocks. Moreover, if there were general motivation or fatigue effects, it would be difficult to explain why these should occur specifically during devaluation trials (note that even devaluation blocks of one outcome (high vs. low value) included valued trials of the respective other outcome). Third, we included consumption trials in all of the blocks and participants consistently choose the high valued response option throughout the reinforcement learning task, thus speaking again against unspecific effects related to the extent of training. Moreover, it might be difficult to explain the specific links that we see between response and outcome representations on the one hand and participants’ behaviour in the last devaluation block on the other hand with the suggested unspecific effects. We discuss this aspect now on page 33 line 743 to 754:

“While we assume that the opposite effects of stress on neural outcome and response representations were due to the action of major stress response systems, there may have been other factors that have contributed to the present pattern of results, such as motivational factors or fatigue at the end of the task. Although we cannot completely rule out these alternatives, in light of our data we consider them rather unlikely. First, if these factors were a result of the mere amount of training, they should have also occurred in the control group, which was not the case. Even if a specific interaction with the stress manipulation is assumed, it is important to note that reaction times remained fast in stressed participants until the end of training and the response accuracy in valued trials or consumption trials remained very high. Furthermore, the observed specificity of the stress effects, which occurred selectively in devalued trials, can in our view be hardly explained by unspecific influences, such as lack of motivation or fatigue.”

It does not seem to me that the task is relevant for the habit/goals distinction. The authors mention how the model-based/model-free distinction may not be interrogating the habit/goal distinction, but this task suffers from the same shortcomings. This is a problem in general with this type of experiments, in that a participant could behave in an S-R or R-O manner independently of the amount of training and switch back and forth between these two strategies throughout the task. The task is simply not well-suited for testing the habit/goal-directed distinction.

We completely agree that the present behavioural task – as well the existing previous tasks – cannot distinguish between the specific contributions of the goal-directed and habitual system, respectively. This is exactly why we used the specific EEG decoding approach, which helps us to identify specific signatures of the goal-directed and habitual system, respectively. Using EEG decoding, we can identify neural representations of the outcome during (stimulus presentation and) response selection, indicative of action-outcome learning, which is at the heart of the goal-directed system. Further, we can identify neural representations of the response during stimulus presentation, indicative of stimulus-response learning, which is at the heart of habit learning. Interestingly, our Bayesian correlation analysis provides evidence that these outcome and response representations are uncorrelated, suggesting that these may indeed reflect neural signatures of distinct processes. Furthermore, as theoretically predicted, response representations were positively correlated with participants’ responses for devalued outcomes, whereas outcome representations were negatively correlated with participants’ responses for devalued outcomes. Thus, using EEG-based decoding, we aimed to overcome shortcomings of existing behavioural paradigms (including the present behavioural task) in the distinction between specific contributions of the goal-directed and habit system, respectively. We have made this rationale for the present study explicit in the introduction of the manuscript. Please see page 5, lines 97 to 101:

“Here, we aimed to overcome these shortcomings of classical paradigms for the assessment of the mode of behavioural control and to examine whether stress leads to an upregulation of habitual processing or a downregulation of goal-directed processing (or both). To these ends, we leveraged electroencephalography (EEG) in combination with multivariate pattern analysis (MVPA)-based decoding of neural representations.”

Furthermore, we state repeatedly that the present behavioural task (same as other existing tasks) cannot distinguish between goal-directed and habitual contributions to behaviour. Please see, for example, page 5, lines 93 to 96:

“Although these elegant paradigms provided valuable insights into mechanisms involved in behavioural control, they cannot show whether increased responding to devalued or degraded actions is due to reduced goal-directed or enhanced habitual control (or both).”

Or, in the discussion on page 26, lines 562 to 565, directly related to the present task:

“However, similar to the behavioural insensitivity to an outcome devaluation after stress, these ERPs cannot separate reduced goal-directed from increased habitual responding. To disentangle goal-directed and habitual processing, we used an MVPA-based decoding approach.”

In addition, the task also employs multiple devaluation phases, which, if anything, should make participants more sensitive to the devaluation procedure. Was that the case?

We included multiple devaluation phases because previous rodent studies suggested that the stress-induced bias towards ‘habit’ behaviour occurs primarily after more extended training (Packard, 1999; Siller-Pérez et al., 2017). In response to this comment, we have tested whether the multiple devaluations per se made participants more sensitive to the devaluation procedure. We tested this idea by analysing potential changes in the responses across the repeated devaluation blocks in the control group. Importantly, we did not observe any changes (neither increases nor decreases in the proportion of responses for devalued outcomes) that would suggest an increased sensitivity to the devaluation in the responses in the repeated devaluation blocks in the control group (p = .240). This result speaks against the idea that the repeated devaluation per se would affect participants’ sensitivity to the devaluation procedure. For stressed participants, we observed even a reduced sensitivity to devaluation across training (in particular in the last block). Thus, if one would assume that the repeated devaluation would have led to an increased sensitivity to the devaluation, the observed stress effects would be even more remarkable.

We report now explicitly that behaviour did not differ across the repeated devaluation blocks in the control group, which speaks against any unspecific sensitization effects due to the repeated devaluation. Please see page 14, line 294 to line 297.

“In the control group, instrumental behaviour did not differ across the different devaluation blocks (F(2,54) = 1.466, P = 0.240, ƞ_p_² = 0.052, 95% CI = 0.013 to 0.049), which indicates that the repeated devaluation phases per se did not result in an increased sensitivity to the devaluation procedure.”

It would be good to see the learning curves associated with each group of subjects. The authors' main finding is an effect that is only observed in the last block of training and only for one condition (when the outcome that was highly valued is devalued). Overtraining effects are only relevant after responding has plateaued. Inspecting if both groups have already asymptoted, or whether one did it before the other is an important point, I think.

We agree and present now additional line graphs in Supplementary file 2, as suggested. As can be seen in this graph, both groups reached a performance plateau relatively quickly and at about the same time. Performance in NoDev blocks did further not differ between the control and stress groups and the time course of learning was comparable in the two groups (time × group interaction: *F*(2,112) = 2.44, *P* = 0.092, *ƞ_p_²* = 0.042, 95% CI = 0 to 0.123; main effect group: *F*(1,56) = 0.30, *P* = 0.585, *ƞ_p_²* = 0.005, 95% CI = 0 to 0.096). We make this latter point clearer in the Results section and refer the reader to the line graphs in supplemental Figure 2. Please see page 14, lines 286 to 291:

“Both groups reached a performance plateau relatively quickly and at about the same time (Figure 3—figure supplement 1). Performance in NoDev blocks did further not differ between the control and stress groups and the time course of learning was comparable in the two groups (time × group interaction: *F*(2,112) = 2.44, *P* = 0.092, *ƞ_p_²* = 0.042, 95% CI = 0 to 0.123; main effect group: *F*(1,56) = 0.30, *P* = 0.585, *ƞ_p_²* = 0.005, 95% CI = 0 to 0.096), suggesting that stress did not affect instrumental learning per se.”

Please see also Figure 3—figure supplement 1:

The same authors have shown that stress manipulations after training render behavior insensitive to devaluation. This suggests that the effect of stress is on the performance of an action, not the learning process per se. How does this fit with their current results showing that stressing people before training changes devaluation sensitivity only in the last block of training? Why there is an extension of training effect in O representation for stressed participants if the effect is on performance?

This is a relevant point which might not have been sufficiently addressed in the previous version of the manuscript. There are by now several studies, from our lab as well as from others, showing that stress or stress hormone administration before learning may induce a bias from goal-directed to habit behaviour (Braun and Hauber, 2013; Dias-Ferreira et al., 2009; Gourley et al., 2012; Hartogsveld et al., 2020; Schwabe et al., 2010; Schwabe et al., 2011; Schwabe et al., 2012; Schwabe and Wolf, 2009; Soares et al., 2012). In one study to which the reviewer is referring to, we showed that stress leads to a comparable shift when stress is induced after learning and before an extinction test (Schwabe and Wolf, 2010). Although this finding shows that stress may interfere with behavioural expression if learning remained unaffected, this finding does not at all rule out that learning would not be affected. Moreover, it seemed that the shift towards habits was somewhat more pronounced when stress was induced before learning than when induced before the test (as discussed in Schwabe and Wolf, 2010). Thus, we do not see how our previous findings would be in conflict with the present findings. Nevertheless, we address this aspect now in the discussion, stating also explicitly that we were not able (and did not aim) to distinguish between stress effects on acquisition and expression of goal-directed vs. habitual processes. Please see page 34, lines 755 to 770:

“In the present study, stress was induced before learning and outcome devaluation. Thus, stress could have affected the acquisition or the expression of instrumental behaviour (or both). While several previous studies demonstrated that acute stress or the administration of stress hormones before learning may shift instrumental behaviour from goal-directed to habitual control (Braun and Hauber, 2013; Dias-Ferreira et al., 2009; Gourley et al., 2012; Hartogsveld et al., 2020; Schwabe et al., 2010; Schwabe et al., 2011; Schwabe et al., 2012; Schwabe and Wolf, 2009; Soares et al., 2012), there is evidence suggesting that stress before a test of behavioural expression may have a similar impact, i.e. stress may induce habitual responding even when stress left acquisition unaffected (Schwabe et al., 2011; Schwabe and Wolf, 2010). The latter finding, however, does not rule out additional effects of stress on acquisition and indeed the impact of stress appeared to be more pronounced when the stress exposure took place before learning (Schwabe and Wolf, 2010). The present study did not aim to distinguish between stress effects on acquisition or expression of goal-directed vs. habitual behaviour but focussed on the impact of stress of the control of instrumental behaviour per se. Our findings thus do not allow a distinction between stress effects on acquisition vs. expression of instrumental behaviour.”

The authors show that the representation of O for the controls increases with training (Figure 4) while the R representation decreases. Theoretically, these two observations should entail a weaker devaluation effect, as the subject needs to encode a representation of the response being performed in order to attain the outcome for a response to be goal-directed (see recent work from Bouton's lab and Dickinson and Balleine's associative-cybernetic model). Perhaps this should be discussed?

We completely agree that the observed decreases of response representations and increases in outcome representations that we observed in control participants should reduce a potential devaluation effect. However, we did not observe a decrease in devalued actions in controls which may be due to the fact that the proportion of responses for devalued outcomes was generally rather low in controls. We briefly discuss this point on page 34, lines 771 to 777:

“Based on the associative-cybernetic model (Dickinson and Balleine, 1993), it could be predicted that the obtained pattern with increased outcome and decreased response representations leads to even reduced responding for devalued outcomes across training in controls, as individuals need to encode a representation of the response being performed in order to attain the outcome for a response to be goal-directed. We did not observe such a decrease, which may be related to the overall relatively low rate of responses for devalued outcomes in control participants.”

Related to this, the fact that a control group did not show evidence of a decrease in outcome representations at the end of training seems problematic, as their argument is based upon the notion that more extensive training makes responding habitual in the stress group but not in the control group. If stress impacts learning, it should there is no argument for the fact that overtraining in a control group does not change outcome representations.

We completely agree that the lack of a decrease in outcome representations (paralleled by a decrease in response representations) across the training in the control group speaks against a practice-related shift towards habits, in the sense of an ‘overtraining effect’, in the present study. Again, our study was not designed to test an overtraining effect per se and we did not aim to do so. Instead, we aimed to test how stress affects specific neural signatures (derived from EEG-based decoding) of goal-directed and habit behaviour, respectively, and to what extent such stress effects may depend on the extent of training.

However, the fact that there was no overtraining effect in the control group of the present study does, in our view, not at all question the training-dependency of the observed stress effect. Our idea that stress might accelerate a shift from goal-directed to habitual behaviour is based on previous rodent studies that showed that stress or stress hormones accelerate a shift from ‘cognitive’ to ‘habitual’ responses that would otherwise only occur with extended training (Packard, 1999; Siller-Pérez et al., 2017). We agree that based on our own data, this idea remains speculative as we do not see evidence of a training-dependent bias towards habitual behavior/processing. We make this latter point clearer now. Please see page 30, lines 660 to 667:

“Findings on ‘cognitive’ and ‘habitual’ forms of navigational learning in rats, however, demonstrated that stress hormones may accelerate a shift from ‘cognitive’ to ‘habit’ learning that would otherwise only occur after extended training (Packard, 1999; Siller-Pérez et al., 2017). Thus, it is tempting to hypothesize that a similar mechanism might be at work during instrumental learning in stressed humans. This conclusion, however, remains rather speculative as we did not observe a training-dependent shift towards habits in the control group and this group even showed reduced response and increased outcome representations over time, which rather suggests increased goal-directed processing across the task.”

Why is there a stronger effect of devaluation on the O-Low condition across the task? Why a less-valued outcome should be able to support this? (pp. 6, line 239)

Overall, there was a highly significant devaluation effect for both low- and high-valued outcomes. The more pronounced valued vs. devalued difference for O^low^ per se (i.e. outcome devaluation × stimulus value interaction) remained even when we analysed only the control group (outcome devaluation × stimulus value interaction: *F*(2,52) = 70.601, *P* < 0.001, *ƞ_p_²* = 0.731, 95% CI = 0.391 to 0.720; valued vs. devalued during Dev O^high^: *t*_27_ = 8.482, *P*_corr_ < 0.001, *d* = 1.603, 95% CI = 1.032 to 2.16; valued vs. devalued during Dev O^low^: *t*_26_ = 8.654, *P*_corr_ < 0.001, *d* = 1.665, 95% CI = 1.071 to 2.246), suggesting that it is not just due to the increased responding for devalued high-outcomes in the stress group. We added this information in the Results section, please see page 14, lines 304 to 310:

“The more pronounced valued vs. devalued difference for O^low^ per se (i.e. outcome devaluation × stimulus value interaction) remained even when we analysed only the control group (outcome devaluation × stimulus value interaction: *F*(2,52) = 70.601, *P* < 0.001, *ƞ_p_²* = 0.731, 95% CI = 0.391 to 0.720; valued vs. devalued during Dev O^high^: *t*_27_ = 8.482, *P*_corr_ < 0.001, *d* = 1.603, 95% CI = 1.032 to 2.16; valued vs. devalued during Dev O^low^: *t*_26_ = 8.654, *P*_corr_ < 0.001, *d* = 1.665, 95% CI = 1.071 to 2.246).”

This observed pattern may be due to a higher habitualization for O^high^ stimuli. The association with a higher reward/outcome value may have led to a higher saliency of the stimuli, which may have resulted in steeper learning curves (as shown in our study) and promoted the formation of stimulus-response (S-R) associations. These stronger S-R associations may have resulted in more responses for the respective outcomes even when these were devalued. We address this potential explanation in the discussion now. Please see page 32, lines 705 to 717:

“In addition, stressed participants showed an increase of insensitivity to outcome in Dev O^high^ but not in dev O^low^. Moreover, we found that the devaluation effect for O^high^ stimuli was stronger compared to the effect for O^low^ stimuli. This difference remained even when we analyzed only the control group, i.e. excluding the possibility that the difference between O^low^ and O^high^ was merely due to the fact that stress increased specifically the insensitivity to the devaluation of O^high^. However, why may the devaluation effect be lower for O^high^ than for O^low^ and why may stress have affected primarily the devaluation of O^high^? These results suggest a stronger habit formation for stimuli that were paired with high rewards. A potential explanation for this pattern takes the links between rewards, stimulus saliency and strength of stimulus-response associations into account: the initial association with high valued outcomes may have increased the salience of the respective stimuli, which in turn may have promoted the formation of stimulus-response associations. These stronger S-R associations, in turn, may have resulted in more habitual responses for the devalued outcomes.”

The devaluation effect for the stress group in block 2 seems stronger than in block 1. If that is the case, then the overtraining argument becomes problematic, as it should supposedly lead to weaker effects of devaluation with training, that is, devaluation should be weaker in block 1 than block 2, and weaker in block 2 than block 3 (the latter contrast being the one they report as being statistically significant). Also, is the difference between stress and control in block 3 different to the difference between stress and control in block 2?

We performed the suggested analyses and obtained no significant differences. There was no evidence for an interaction of block number (1 vs. 2) and experimental treatment (*F*(1,56) = 1.575, *P* = 0.215, *ƞ_p_²* = 0.027, 95% CI = 0.013 to 0.052), which would have been indicative of the decrease from block 1 to 2 the reviewer was referring to. There was, however, a significant interaction between block (2 vs 3) and treatment when analysing the changes from block 2 to block 3 (*F*(1,56) = 13.589, *P* < 0.001, *ƞ_p_²* = 0.195, 95% CI = 0.105 to 0.319). Follow-up tests showed that after moderate training intensity (i.e. in block 2), groups did not differ in responses for the devalued outcome (*t*_56_ = 0.165, *P* = 0.870, Cohen’s *d* = 0.043, -0.472 to 0.558). As reported in the Results section, after a higher number of repetitions (i.e. in devaluation block 3), stressed participants responded significantly more often for the devalued outcome than control participants. Thus, while there were no significant changes from block 1 to block 2, there was a significant increase in devalued responses in the stress group, but not in the control group, from block 2 to block 3. We focus now more explicitly on the (potential) changes from block 1 to 2 and block 2 to 3 on page 15, line 327 to 334:

“Furthermore, while there was no evidence for an interaction of devaluation block number (1 vs. 2) and experimental treatment (*F*(1,56) = 1.575, *P* = 0.215, *ƞ_p_²* = 0.027, 95% CI = 0.013 to 0.052) when analysing the changes from the first to the second devaluation block, we obtained a significant interaction between block (2 vs. 3) and treatment when analysing the changes from block 2 to block 3 (*F*(1,56) = 13.589, *P* < 0.001, *ƞ_p_²* = 0.195, 95% CI = 0.105 to 0.319). Moreover, follow-up tests showed that after moderate training intensity (i.e. in block 2), groups did not differ in responses for the devalued outcome (*t*_56_ = 0.165, *P* = 0.870, Cohen’s *d* = 0.043, -0.472 to 0.558).”

The theoretical aspect that the study aims go address is whether the devaluation effects are due to less goal-directed control or more habitual control. In their Discussion section, the authors argue that a response cannot be goal-directed and habitual at the same time. However, Perez and Dickinson (2020) have recently published a free-operant model where it is perfectly possible that a response is driven in equal proportions by each system. It is only in the case of discrete-trial procedures such as the present one and the 2-step task from the model-based/model-free tradition that the answer is difficult.

According to canonical operational definitions, instrumental behaviour can at least not be fully goal-directed and fully habitual at the same time (Adams, 1982; Dickinson and Balleine, 1994). However, we completely agree that behaviour may not always necessarily be either fully goal-directed or habitual and there may be different degrees to which behaviour is under goal-directed or habitual control (this assumption formed actually the basis of our study idea). We further agree that both the goal-directed and habit system may contribute to the probability of responding, as proposed by Perez and Dickinson (2020). Here, we used an MVPA-based decoding approach to overcome the methodological shortcomings of behavioural paradigms with discrete behavioural responses and to unravel to contributions of the goal-directed and habitual systems to behaviour. Our findings point to distinct (neural) contributions of these systems.

We refer now also to the recent work by Perez and Dickinson, 2020 when we discuss the dissociable contributions of goal-directed and habitual processes to behaviour. Please see page 35, line 786 to 788:

“Classical behavioural paradigms involving discrete responses, however, can hardly disentangle goal-directed and habitual components in a specific pattern of responding (e.g. insensitivity to outcome devaluation). […] Recently, a free-operant model was proposed that allows a behavioural dissociation of goal-directed and habitual contributions to behaviour (Perez and Dickinson, 2020).”

It seems puzzling that the control group shows decreased representations of the response with training. If anything, that should be maintained or augmented with training. This is another reason to question the overtraining argument of this paper.

We completely agree that the control group did not show signs of increased habit behaviour across the reinforcement learning task. Again, our study was not designed to test an overtraining effect per se and we did not aim to do so. Instead, we aimed to test how stress affects specific neural signatures (derived from EEG-based decoding) of goal-directed and habit behaviour, respectively, and to what extent such stress effects may depend on the extent of training. This rational of the study has now been made clearer in the introduction (please see page 5, lines 97 to 101).

“Here, we aimed to overcome these shortcomings of classical paradigms for the assessment of the mode of behavioural control and to examine whether stress leads to an upregulation of habitual processing or a downregulation of goal-directed processing (or both). To these ends, we leveraged electroencephalography (EEG) in combination with multivariate pattern analysis (MVPA)-based decoding of neural representations.”

Please see also page 5, lines 112 to 121:

“Because previous rodent studies suggested that stress or stress hormones modulate the balance of goal-directed and habitual forms of learning after more extended training (Dias-Ferreira et al., 2009; Packard, 1999; Siller-Pérez et al., 2017), we further asked whether the stress effect on goal-directed and habitual processing, respectively, would depend on the extent of training. Although we did not aim to test training-induced changes in the balance of goal-directed and habitual processing per se, we therefore included transient outcome devaluations after initial, moderate, and extended training in the instrumental learning task to assess if and how the predicted changes in neural representations are linked to behavioural manifestations of stress-induced changes in behavioural control.”

We referred to the decrease in the neural response representation across the task in controls on page 21, lines 467 to 472:

“As shown in Figure 2H, participants of the stress group showed a stronger response representation, reflected in a higher classification accuracy for the response categories, with increasing training intensity (first half vs. last half: *t*_25_ = 2.51, *P*_corr_ = 0.038, *d* = 0.491, 95% CI = 0.079 to 0.894), whereas there was even a decrease in the control group (first half vs. last half: *t*_24_ = 3.50, *P*_corr_ = 0.004, *d* = 0.701, 95% CI = 0.256 to 1.134).”

Please also see page 30, line 660 to 667 in the Discussion section:

“Findings on ‘cognitive’ and ‘habitual’ forms of navigational learning in rats, however, demonstrated that stress hormones may accelerate a shift from ‘cognitive’ to ‘habit’ learning that would otherwise only occur after extended training (Packard, 1999; Siller-Pérez et al., 2017). Thus, it is tempting to hypothesize that a similar mechanism might be at work during instrumental learning in stressed humans. This conclusion, however, remains rather speculative as we did not observe a training-dependent shift towards habits in the control group and this group even showed reduced response and increased outcome representations over time, which rather suggests increased goal-directed processing across the task.”

Other points:The analysis of response representations after extended training is based in comparisons of first half and last half of training, why that choice?

We contrasted the first against the last half of the training to ensure a sufficient number of trials within the decoding approach. Importantly, however, when we grouped the accuracy neural representation accuracy data differently, the pattern of results was largely comparable.

The results of alternatively grouped data are presented in the Supplementary File 1A.

In the discussion, the authors mention that "habit behavior requires extensive training", but a recent preprint by Pool et al. (2021) shows that most participants can be habitual after limited training.

We agree that our initial wording was a bit too far-reaching and have rephrased this sentence, also based on a respective comment of reviewer #1 as follows (page 28, lines 633 to 637):

“For instance, while model-free learning can be present after only few trials, establishing habit behaviour relies on the strength of the stimulus-response association, which among other factors such as reward magnitude or stimulus saliency, has been suggested to depend on repetition (Adams, 1982; Dickinson et al., 1995; Tricomi et al., 2009; but see Wit et al., 2018).”

The recent manuscript by Pool et al. (2021) showed indeed that some participants perform habitually after ‘limited’ training. It is, however, important to note that this ‘limited’ training comprised already several hundred responses (12 blocks of 40 seconds and 3-4 responses per second). Thus, these findings are not in conflict with the idea that habit behaviour requires (among other factors) repetition, as we state now in the rephrased sentence.

Pp 24. Line 522: "Furthermore, while model-free learnins is based on previously experienced outcomes, habit learning is defined as S-R learning without links to the outcome engendered by a response." I could not follow this idea, can you clarify?

We thank the reviewer for the opportunity to clarify this aspect. Model-free learning is based on the value of rewards, i.e. it explicitly takes the value of an outcome into account. This is in contrast to habitual S-R-processes which are assumed to depend on the association between responses and antecedent stimuli regardless of the outcome that is engendered by the action. We added this information in the manuscript, please see page 29 lines 640 to 644:

“In other words, whereas model-free learning is dependent on the expected reward and hence the outcome that follows a response (Dayan and Berridge, 2014; O'Doherty et al., 2017), habit learning is operationally defined as being independent of the motivational value of the outcome (Adams, 1982; Adams and Dickinson, 1981; Balleine and Dickinson, 1998).”

During the presentation of the results, the authors show statistics for both the O-high and O-low conditions. However, the O-low condition did not add anything to the analysis, as it did not show any effects of devaluation. Therefore, for the sake of simplicity and the reader's joy, I'd simply leave that for the supplemental material and focus on the part of the experiment that is relevant for the point the authors are trying to make.

Please note that *eLife* does not allow supplementary text (only Supplementary files including figures). As we do think that the dependency of the effects on the value of the outcome (high vs. low) is relevant and for the sake of completeness, we would strongly prefer to leave the respective information in the (main) text – if both the reviewers and editors agree.

Table 1: The caption mentions that the scales ranged from 1-10, but the results show means higher than 10. Am I missing something here?

Thank you for noticing this mistake, which has now been corrected. The correct range is from 1-100. Please see page 12:

Pp. 6, line 134 "offer, trade" should apparently read "offer, or trade"

Thank you for noticing this mistake, is should actually read “and trade”. Please see page 6, line 138 to 140:

“Finally, participants received feedback about whether the alien accepted the offer and traded one of the desired cards, and how many points were earned (O).”

This approach was employed by McNamee and colleagues (2015) using fMRI. I think it should be cited.

We agree and cite this reference now on page 8, line 170:

“Outcome representations at the time of S presentation and R are indicative for goal-directed control. Habitual control, on the other hand, should be indicated by increased R representations at the time of stimulus (S) presentation (McNamee et al., 2015).”

pp. 15, line 308: "tended to be sensitive". I'd just say it was not significant.

We agree and refer to this result now as a non-significant trend. Please see page 18, lines 390 to 398:

“Moreover, we identified a late component that showed a non-significant trend towards sensitivity to the outcome devaluation during Dev O^high^ blocks in control participants (devalued vs. valued: *t*_24_ = 1.91, *P* = 0.068, *d* = 0.382, 95% CI = -0.028 to 0.785) but not in stressed participants (devalued vs. valued: *t*_27_ = 1.57, *P* = 0.127, *d* = 0.297, 95% CI = -0.084 to 0.673; outcome devaluation × stimulus value × group interaction: *F*(2,102) = 5.20, *P*_corr_ = 0.042, *ƞ_p_²* = 0.093, 95% CI = 0.008 to 0.199; stimulus value × group interaction: *F*(1,51) = 6.05, *P* = 0.017, *ƞ_p_²* = 0.106, 95% CI = 0.003 to 0.273; no such effect in NoDev and Dev O^low^ blocks: stimulus value × group interaction: both *F*(1,51) < 1.44, both *P* > 0.236, both *ƞ_p_²* < 0.027, both 95% CI = 0 to 0.159).”

pp. 18, line 370: P = 0.034, should be P_corr?

The mentioned statistic refers to an overall interaction effect, for which only one test was performed. Therefore, a correction for multiple testing is not required here.

Pp 31, lines 683-684. Could you please clarify the trial structure and numbers? What do you mean by "27 trials each"?

Participants completed 27 NoDev blocks, 3 Dev O^high^ blocks, and 3 Dev O^low^ blocks. Each block consisted of 27 trials (12 learning trials with S^high^, 12 learning trials with S^low^, and three consumption trials). We clarified this in the text; please see page 71, lines 317 to 319:

“Participants completed 27 NoDev blocks, 3 Dev O^high^ blocks, and 3 Dev O^low^ blocks, with 27 trials per block: 12 learning trials with S^high^, 12 learning trials with S^low^, and three consumption trials.”

Reviewer #3 (Recommendations for the authors):1. I think that the manuscript would benefit from more of an effort to link the current results to other related work. For one, the authors briefly touch on how the results relate to the 'model-based and model-free' dichotomy described in other work, but mostly highlight how the operationalization of 'model-free behavior' differs from 'habits,' as described here. However, I think it is notable that the authors find that habitual behavior involves a reduction in prospective consideration of an outcome, similar to other work that has shown that model-based behavior increases prospective consideration of future states during decision-making (e.g. Doll et al., 2015). Do the authors believe that 'habit/goal' and 'model-free/model-based' decision-making strategies might share common mechanistic features?

Although we focussed in the manuscript mainly on differences between the two frameworks to point out that an analysis of model-free vs. model-based processes would be less helpful to address the key question of our study, we fully agree with the reviewer and we do think that habitual/goal-directed and model-free/model-based processes may share some mechanistic features. These commonalities relate to the consideration of an outcome but also to the underlying neural structures. We have added now that there are also several commonalities between the habit/goal-directed and model-free/model-based frameworks and we link our findings more to other work in this area.

Please see page 28, lines 614 to 628:

“There are indeed several commonalities between goal-directed vs. habitual processes on the one hand and model-based vs. model-free processes on the other hand and these processes may share some mechanistic features. For instance, both goal-directed and model-based processes involve a complex representation of the environment and allow for flexible, deliberate action compared to rather rigid model-free or habitual processes (Dolan and Dayan, 2013). Moreover, at a neural level both goal-directed and model-based behaviour are associated with overlapping prefrontal and medial temporal areas (Balleine and O'Doherty, 2010; Gläscher et al., 2010; Valentin et al., 2007). Further, there is evidence that also the balance of model-based and model-free processes can by modulated by stress (Cremer et al., 2021; Otto et al., 2013; Radenbach et al., 2015) and our finding that more habitual responses after stress correlated negatively with neural outcome representations appears to be generally in line with earlier findings showing that model-based behaviour increases prospective consideration of future states during decision-making (Doll et al., 2015).”

2. I am somewhat concerned that saccadic eye movements could be acting as a confound for some of the EEG decoding results and would appreciate the authors clarifying a few points related to this and possibly doing some additional control analyses to rule out this possibility:a. The authors mention that the symbols for the aliens differed by position – was this position fixed for each alien tribe, i.e. did the blue tribe always appear on the left for a particular participant? If so, this could drive anticipatory eye-movements in that direction.

The positions of the symbols for the aliens on the computer screen were fully randomized. Thus, it is highly unlikely that anticipatory eye-movements could bias our decoding results. We have added this important information in the method section, please see page 40, lines 897 to 898:

“The position of the aliens (left/right) representing the response options was randomized across trials.”

b. It is not clear which electrodes are driving the change in outcome and response representation in the decoding analysis. An analysis examining which features are contributing most to successful decoding in the stimulus and response period would be particularly important to rule out the possibility that the main results of interest are not driven by anticipatory eye-movements, such as saccades in the response phase where two stimuli are placed side-by-side.

We appreciate the opportunity to clarify this issue. First of all, the classifier applies those electrophysiological properties for testing in the reinforcement learning task (RT) that had been identified in the delayed-matching-to-sample (DMS) task for the separation of object vs scene and blue vs red symbols. Artefacts that occur only during the reinforcement task will simply be ignored as they did not constitute discriminative features in the training data set (Grootswagers et al., 2017). Thus, artefacts could have affected the classifier only if the same artefacts were present during both the DMS task and the RT, which is rather unlikely. Moreover, while the classifier was trained on the epoched EEG data of the DMS delay phase, during which participants were constantly presented with a blanked screen, the trained classifier was applied to the EEG data during stimulus presentation and response choice in the RT. Thus, the visual input during the train and test data differed substantially, thus making, for example, similar saccades even more unlikely. Furthermore, the training DMS data were collected in two sessions, of which one was completed before participants underwent the stress induction and another after completion of the reinforcement learning task. Importantly, the classifier performance did not differ between both DMS sessions and was not modulated by the experimental treatment. Consequently, we think that (i) group-specific artefacts and (ii) emotional and motivational confounds that could have arisen from the RT completion such as fatigue cannot account for our decoding results. Moreover, the positions of the symbols for the aliens on the computer screen were fully randomized. Thus, we do not think that anticipatory eye movements, such as saccades during the response phase, could have had driven the EEG in the stress group at the end of the task. Apart from the methodological principles of the EEG decoding and our specific task design, there is also evidence that classifiers have the capacity to ignore bad channels or suppress noise during training, making artefact correction, in general, less critical in decoding analyses (Grootswagers et al., 2017). Together, we think it is highly unlikely that could have biased the decoding results.

In addition, we used eye-tracking during the EEG measurement to rule out biases due to eye movements. We have added now data showing that stress did not interact with outcome value, trial type or training intensity to alter blinks or saccades. Please see page 24, lines 532 to 537:

“Furthermore, we recorded eye-tracking data to control for potential group differences in saccades or eye blinks. These control analyses showed that there were no significant group differences in saccades or eye blinks across the task or depending on the trial type (outcome devaluation × stimulus value × time × group: *F*(4,196) = 0.78, *P* = 0.54, *ƞ_p_²* = 0.02, 95% CI = 0.002 to 0.008; outcome devaluation × stimulus value × group: *F*(2,98) = 1.03, *P* = 0.36, *ƞ_p_²* = 0.02, 95% CI = 0.005 to 0.020; see supplementary file 1D).”

Furthermore, we added a brief section on the eye-tracking to the methods section.

Please see page 42, lines 959 to 964:

“Eye-tracking. We used a desktop mounted eye-tracker (EyeLink 1000; SR-Research Ltd., Mississauga, Ontario, Canada) to record monocular eye movements from the left eyes at a sampling rate of 500Hz. We used custom scripts implemented in MATLAB (The Mathworks, Natick, MA) to extract the mean saccades and blink information depending on the stimulus value, outcome devaluation and time (zero to 2000 ms around the stimulus onset). Data for two participants were missing due to a failure of the used eye-tracker.”

c. Relatedly, I would have appreciated more detail on the preprocessing pipeline used to clean-up the data, in particular how saccadic eye-movements and blinks were corrected, or removed, and if central fixation was enforced with eye-tracking or EOG.

For both, the ERP and MVPA analysis, we first applied the PREP pipeline to transform the channel EEG data using a robust average reference. Furthermore, bad channels were interpolated and a high pass filter of 0.1 Hz and a low pass filter of 100 applied.

Similar to previous studies (e.g. Mai et al., 2019), we did not perform any artefact correction on the input data for the MVPA approach. It has been shown that classifiers are highly robust against movement and eye-blink artefacts (Grootswagers et al., 2017). As the classifier is able to ignore bad channels and noise in the training procedure, artefact correction is not an essential part of the decoding pre-processing pipeline (Grootswagers et al., 2017). Furthermore, participants completed two DSM sessions: the first took place after participants underwent the stress or control manipulation and the second DMS task was completed after the reinforcement task. Importantly, the classifier performance did not differ between both DMS sessions and was not modulated by the experimental treatment. Therefore, our experimental design in combination with methodological properties inherent in the classifier approach makes it rather unlikely that our decoding results could have been biased by artefacts.

For the ERP analysis, we removed muscular artefacts and corrected for blinks and eye movements via independent component analysis. We added this information now in the manuscript, please see page 45, lines 1028 to 1029:

“In addition, we removed muscular artefacts and corrected for blinks and eye movements via independent component analysis.”

3. Looking at the scatterplots of Figure 5A and 6, the values on the y-axis appear to be rather different. Where the control and stress groups are mostly separate on this dimension in Figure 5A, they are highly overlapping in Figure 6 such that there does not appear to be any group difference. Are these different data and if so what is the rationale for selecting one set of data for one analysis and a different set for the other?

For some data points in the previous Figures 5A and 6A, the colour-based group assignment was incorrect (i.e., some data points were coloured in blue instead of red). We apologize and have corrected these mistakes and present now the corrected colouring. Please see page 21.

[Editors' note: further revisions were suggested prior to acceptance, as described below.]

The manuscript has been improved but there are some remaining issues that need to be addressed, as outlined below:(1) Reviewer 2 notes that enhanced response representations do not necessarily reflect an increased involvement of the habit system – this is particularly true if such representations are not negatively correlated with outcome representations, since the latter retrieves the former according to several associative accounts of goal-directed behavior. Thus, the stated assumption, that "habitual responding is reflected in enhanced response representations" is not valid. The reviewer suggests, and I agree, that claims based on this assumption should be removed or significantly toned down.

We completely agree that, according to associative learning accounts, response representations may be relevant for both ‘goal-directed’ S-R-O learning and ‘habitual’ S-R learning. Therefore, the assumption that response representations are indicate for the habit system is not valid. On the other hand, our empirical data do, in our view, lend some support for the idea that outcome representations are indicative for the goal-directed system, whereas response representations are linked to the habit system: while outcome representations were negatively correlated with participants’ responding for devalued outcomes, response representations were positively correlated with responses for devalued outcomes. Nevertheless, we are now more careful in our wording and rephrased the relevant sentences throughout the manuscript (including the abstract) to (i) avoid the impression that we consider response representations as being directly indicative for the habit system and (ii) make clear that response representations may be relevant for both goal-directed and habit learning. Moreover, we refer now simply to outcome and response representations or processing, instead of goal-directed and habitual processing, to stick closer to what we actually measured, as suggested by the reviewer. Please see, for example, page 8, lines 175 to 177:

“Outcome representations at the time of S presentation and R are indicative of goal-directed control. In contrast, response representations at the time of stimulus-representations, may be relevant for both goal-directed S-R-O learning and habitual S-R learning.”

Or on page 21, lines 458 to 461:

“While it is assumed that the outcome representation that is crucial for goal-directed S-R-O learning is reduced with increasing habitual behaviour control, response (R) representations at the time of stimulus (S) presentation may be involved in both goal-directed S-R-O and habitual S-R processing.”

Or on page 24, lines 538 to 539:

“…to provide evidence that acute stress results in a decrease of outcome-related processing that is critical for goal-directed control, and paralleled by an increase in response processing.”

(2) Reviewer 3 notes that the new FRN analyses are not sufficiently contextualized, and may even be erroneous, and also that the previous request for more information regarding decoding features went unanswered.

We have checked the FRN analyses but decided – based on the comments of reviewer 3 – to remove the FRN from this revised manuscript because this analysis appeared to be (a) only indirectly relevant to the present study and (b) potentially unreliable given the low number of errors.

We provide now information on the features that contributed the most to decoding the responses and outcomes. A parieto-occipital cluster was contributing the most to decoding outcome representations and a centro-parietal cluster was contributing the most to decoding response representations. For more details, please see our responses to the respective comments below.

(3) For my part, I am still confused about the distinction between model-free RL and habits. The authors state:"It might be argued that an alternative way of disentangling goal-directed and habitual contributions to instrumental responding under stress would be through an analysis of model-based and model-free learning …"Who would argue that? Why would that analysis be better? What exactly would evaluation of these models add here?And then:"For instance, while model-free learning can be present after only few trials, establishing habit behaviour relies on the strength of the stimulus-response association, which among other factors such as reward magnitude or stimulus saliency, has been suggested to depend on repetition Furthermore, while model-free learning is based on previously experienced outcomes, habit learning is defined as S-R learning without links to the outcome engendered by a response … In other words, whereas model-free learning is dependent on the expected reward and hence the outcome that follows a response, habit learning is operationally defined as being independent of the motivational value of the outcomeThis is incorrect. The model-free Q-value *is* the S-R association: both depend on the reward prediction error and the learning rate, both reflect previous reinforcement, and in that sense an expectation of reward, both are void of sensory-specific outcome features, and thus insensitive to devaluation, and both can develop rapidly or not, depending on the learning rate. I suggest that the entire section on model-based and model-free learning is remove.

As the section on model-based and model-free learning is only indirectly relevant for the current study and seemed to be less helpful, we decided to remove this entire section, as suggested by the Reviewing Editor.

(4) Finally, the writing needs some attention. As an example, it is repeatedly stated that canonical behavioral assays can "hardly" disentangle habitual and goal-directed behavior. The term "hardly" means barely, as in "I can barely see that distant object", or alternatively, it can mean definitely not, as in "You can hardly expect me to …" I don't think either is the intended meaning here, and there are other problems throughout the manuscript. I strongly suggest proofing by a native English speaker.

We went over the manuscript again, focussing in particular on language aspects.

Furthermore, our revised manuscript has now been proofread by a native English speaker. We hope that the problems the reviewer was referring to have now been corrected.

Reviewer #2 (Recommendations for the authors):The authors have made a good effort to respond to all of my concerns. Given that my initial concerns with the task were not well received, I will not insist at this point on the merits of this task to produce or investigate habitual control, so I will focus on the author's responses only and some theoretical points that I think should be re-written to improve the quality of the manuscript.The authors insist that their aim *was not* to investigate overtraining, but the very definition of overtraining that one could infer from the habit literature is that it refers to the amount of training that produces habitual behaviour. They used devaluations across different extensions of training to investigate whether " stress leads to an increase in habitual processing, a decrease in goal-directed processing or both ", so I'm still not convinced with their assertion.

We used a first devaluation block very early in training (i.e. after just one block of training) and another one at the end of the training session. However, the training as a whole took place in a single session that included about 300 responses per stimulus. Although this trial number allows an analysis of the goal-directed vs. habitual control of behaviour (see also Valentin et al., 2007 or Wit et al., 2009 for paradigms including even less trials), compared to studies that were specifically interested in overtraining effects, this trial number is rather low. For instance, Tricomi et al. (2009) used a three-day training protocol that involved in total more than 10.000 responses (12 sessions of 8 minutes each, with on average 2 responses per second). Similar protocols, including significantly more responses than in the present task, were employed in recent studies that specifically aimed to test for overtraining effects (Pool et al., 2022; Wit et al., 2018). Thus, we do not think that the number of trials realized in the present study was sufficient to test for potential overtraining effects, which, again, were not a focus of the present study.

We elaborate on this aspect on page 27, lines 596 to 600:

“However, whether or not overtraining may induce habitual behaviour in humans is currently debated (Wit et al., 2018) and our data cannot speak to this issue as training may have been too limited to result in overtraining-related habits (which might require thousands of responses; Tricomi et al., 2009).”

Related to this, they are strongly assuming that the decoding analysis will give insights as to the interaction and development of habitual and goal-directed (g-d), but you need a theory of habits and actions in order to test for that. In my view, what they are doing in this paper is more empirical than theoretical ("do outcome representations decrease with training?; do response representations increase with training?), and I'd suggest they delete any reference to what they believe is the interaction between the systems in this task – right now they are motivating the paper as providing insights into the question of whether habits are a consequence of increased habits or decreased g-d control. I personally see this as a more empirical than theoretical paper, and the current motivation is poorly justified from a theoretical perspective, I think. For example, they assume that R representations should be stronger the more active a habit is, but Bouton and his colleagues have demonstrated -at least behaviorally- that this not necessarily the case.

Although our data do show that outcome representations correlate negatively with responding for devalued actions, whereas response representations correlate positively with this behavioural readout of habitual behaviour, we completely agree that, from a theoretical point, it is not valid to take response representations as a direct indication of the habit system (as response representations are relevant for both S-R-O and S-R learning). We are now more careful in our wording and removed passages which indicated that response representations would be a direct read-out of the habit system. Instead, we emphasize now repeatedly that response representations are relevant both for goal-directed S-R-O and habitual S-R learning. As suggested by the reviewer, we stick now primarily to what we have measured and simply refer to outcome and response representations (or processing), respectively.

Please see, for example, page 8, lines 175 to 177:

“Outcome representations at the time of S presentation and R are indicative of goal-directed control. In contrast, response representations at the time of stimulus-representations, may be relevant for both goal-directed S-R-O learning and habitual S-R learning.”

Or on page 21, lines 458 to 461:

“While it is assumed that the outcome representation that is crucial for goal-directed S-R-O learning is reduced with increasing habitual behaviour control, response (R) representations at the time of stimulus (S) presentation may be involved in both goal-directed S-R-O and habitual S-R processing.”

Or on page 24, lines 538 to 539:

“…provide evidence that acute stress results in a decrease of outcome-related processing that is critical for goal-directed control, and paralleled by an increase in response processing.”

The authors state that "goal-directed but not habitual control relies on the motivational value of the outcome" (pp. 4). Without wanting to be pedantic, habits do depend on the motivational value of the outcome to develop (unless you take a Guthrian view of S-R contiguity being sufficient to produce them), so I'd rather rephrase this as "only the g-d system is sensitive to changes in the motivational value of the outcome in absence of new experience with it", or something along those lines. This should make it clear that it is the devaluation test what distinguishes between the two types of control.

We agree and have rephrase this sentence as suggested.

Please see page 4, lines 89 to 92:

“These paradigms are based on the key distinctive feature of goal-directed and habitual control, i.e. that only the goal-directed system is sensitive to changes in the motivational value of the outcome in absence of new experience with it (Adams, 1982; Dickinson and Balleine, 1994).”

McNamee and colleagues did a very similar paper using fMRI decoding, but the authors cite this paper briefly, without any reference to what the paper is about and what it found. I think that they should be more detailed about what these other authors did in that paper, discussing it in the introduction and motivating their experiment as an extension of the same idea using a different method (EEG).

We agree and refer to the study by McNamee et al. (2015) and its findings in more detail. Please see pages 5 and 6, lines 115 to 119:

“Using a similar decoding approach on fMRI data, a previous study showed that brain regions implicated in goal-directed control contained information about outcomes and responses, whereas regions associated with habitual responding contained only information about responses (but not outcomes) at the time of stimulus presentation (McNamee et al., 2015).”

Pp 10. "Degraded actions" suggests degraded action-outcome contingency. This is not the manipulation employed in the paper.

We completely agree and have corrected this mistake. The sentence reads now “In order to adequately assess whether increased responding to devalued actions is due to reduced goal-directed or enhanced habitual control (or both),…” (please see page 8, lines 178 to 179).

Again, without wanting to be pedantic, the authors state that: ""In other words, whereas model-free learning is dependent on the expected reward and hence the outcome that follows a response (Dayan and Berridge, 2014; O'Doherty et al., 2017), habit learning is operationally defined as being independent of the motivational value of the outcome (Adams, 1982; Adams and Dickinson, 1981; Balleine and Dickinson, 1998).", but algorithmic representations of habit learning (and pavlovian conditioning) since the time of Bush and Mosteller and before are based on the notion of prediction-error, that is, they are represented as model-free algorithms. The "expected reward" in MF algorithms is not a psychological, but statistical concept (it turns out that prediction-error learning can be taken as estimating the value of stimuli or states.) What is critical is that these algorithms are not sensitive to changes in outcome value unless the new value is re-experienced and updated, whereas model-based or g-d algorithms are.

We agree. However, as suggested by the Reviewing Editor, we have now decided to remove the entire section on model-based and model-free learning, as this section was only indirectly relevant to the present study and seemed to be rather confusing.

Reviewer #3 (Recommendations for the authors):The authors have addressed my main concerns about potential eye-movement artifacts acting as confounds in the decoding analysis, and mostly answered my questions about the preprocessing procedures for the EEG data. In the process of addressing many of the points raised during review, they had had to substantially tamp down some of their claims and provide more context, caveats and speculation regarding explanations for their observations. I don't think this is necessarily a bad thing, as the pattern of data rarely perfectly match our predictions and such deviations from expectation and theory are usually the grist of new research and investigation. Moreover, the main results and their interpretation hold up despite some points that remain a bit muddy (e.g. why the effects of stress occur later in the experiment rather than closer to the time of the stressful event). However, not all of my points were addressed in the initial round of reviews and I have some remaining questions – mostly arising from new analyses in the revision, and would also like additional clarity on some methodological points.1. In responding to the other reviewers, the authors have added some new analyses of the feedback-related negativity to test if participants in the stress and control groups differed in how the processed error feedback. Interpretation of the null result here seems rather indirect, as it assumes that this ERP would change in response to differences in the sensitivity of participants to error feedback. I suspect that previous work has found such individual differences in groups with greater error sensitivity before, but that work is not cited here. It would be helpful to provide more background on what this ERP is thought to reflect to provide additional context to this result.

We agree that the interpretation of the null result is rather indirect here. Even more importantly, however, participants made substantially fewer incorrect than correct trials, which may render the FRN analysis less reliable and may have actually resulted in the rather unusual pattern of the FRN to which the reviewer is referring in his/her next comment. We therefore decided to remove this analysis from the manuscript.

Please note that we added the FRN analysis in the previous round of revision in response to a reviewer comment asking us to address the potential influence of negative social feedback on avoidance of the non-devalued action. We address this issue explicitly and argue that it is in our view rather unlikely that the negative feedback had a major impact on our results. These arguments remain and, as mentioned by the reviewer, the FRN would have been only of indirect relevance here. Please see page 28, line 629 to page 29, line 639:

“It is important to note that participants received an error feedback in devalued trials when they chose the response option that was not associated with the now devalued outcome. Given that acute stress may increase sensitivity to social cues (Domes and Zimmer, 2019), one might argue that stressed participants continued to respond towards devalued outcomes in order to avoid being presented with the error-feedback screen. However, we consider this alternative to be unlikely. First, with respect to the neural outcome and response representations, these were analysed during NoDev blocks in which groups did not differ in their behavioural performance accuracy, and consequently not in the frequency of error feedback. Furthermore, participants’ performance in devalued blocks was directly associated with the observed changes in neural outcomes and response representations during the NoDev blocks, which again, could not be biased by differences in error feedback processing.”

2. The FRN in Figure 2 —figure supplements 4-6 looks very different from what I expected, and what I believe, is conventional in the literature. The response on error trials is generally more positive than errors in the 200-300 ms post-feedback window that the authors focus on in this analysis. Generally, I believe this ERP is visible as a more negative deflection on error trials riding on a larger positive response to the feedback stimulus in this time window (e.g. Yeung et al. 2005, Cerebral Cortex). The baseline for error and correct trials also differ substantially in the Fz electrode – and differ quite a bit from zero. The unusual appearance of these ERPs make me somewhat concerned that there might be an error in the analysis.

We agree that the pattern of results differs from conventional findings. Therefore, we have re-examined our analyses. We obtained a minor mistake in the baseline correction. After correcting this mistake, we obtained plausible baseline values for all three electrodes. However, the general pattern of the FRN remained unchanged and critically distinct from the typically observed FRN (e.g. Yeung et al., 2005). In retrospect, this finding may be owing to the generally very low number of incorrect compared to correct trials (e.g., 99% correct trials in the NoDev blocks), which may render the FRN in the present study less reliable. In light of concerns of reliability of the FRN, we decided to remove the FRN from the present manuscript, which may have been only of indirect value for our interpretation anyways (please see our response to the previous comment).

Please note that we took the minor mistake in the analysis of the FRN as an occasion to critically check all of our analysis, which showed however no further mistakes.

3. This may have been understandably lost in the long list of comments in the last round of review, but the authors did not respond to my request that they provide more information about the features that contribute to the decoding of the outcome and response – i.e. the particular channels and time points that are contributing to the decoding accuracy measures. While they have convincingly argued that EOG artifacts are unlikely to drive their results, I think it would still be valuable to also see which features are contributing to most to decoding.

We apologize for missing this aspect in our previous revision. We have now analysed which features are contributing the most to outcome and response decoding, respectively. More specifically, we have run a search-light RSA asking which features are most informative. These analyses revealed that a frontal-left cluster and a left parieto-occipital cluster were contributing the most to decoding outcome representations, whereas a centro-parietal cluster regions was contributing the most to decoding response representations. We have added this information to the manuscript on page 46, lines 1044 to 1049:

“In retrospect, a search-light RSA asking which features contribute the most to decoding neural outcome and response representations, revealed that at the time of average maximal decoding accuracy (about 200 ms after stimulus onset for both classifications), a parieto-occipital cluster was contributing the most to decoding outcome representations. In contrast, a centro-parietal cluster was contributing the most to decoding response representations (Figure 2 —figure supplement 4).”

We also added more information on the searchlight procedure in the methods section. Please see page 47, lines 1062 to 1068.

“Searchlight approach. To provide more insight into which electrophysiological information contributed most to differentiating between the respective categories (red vs. blue and objects vs. scenes, respectively), we also performed a searchlight analysis. This allowed us to determine the topographic features that discriminated most between the two categories. In order to implement the searchlight approach, a SVM was used. Again, we calculated the Wald interval with adjustments for a small sample size (Agresti and Caffo, 2000; Müller-Putz et al., 2008).”

4. This is a more minor point, but it would be helpful to have more information about how ICA was used to remove motor and eye-movement activity E.g. how many components were removed, how were they identified and how did the authors verify success of this preprocessing. The current one sentence mention of ICA is not very illuminating about the specific approach used for this study.

We agree and elaborate now on the ICA. Please see page 42, line 963 to page 43, line 977:

“In addition, blinks and eye movements were corrected by independent component analysis (infomax ICA, Noh et al., 2014). Using the automated procedure ADJUST (Mognon et al., 2011), ocular artifact-related components in EEG recordings we identified and subsequently removed. The ADJUST algorithm combines stereotypical artifact-specific spatial and temporal features to detect and differentiate artifact ICA components (Chaumon et al., 2015). For example, ADJUST computes the kurtosis of the event-related time course for frontal channels, since, for example, eye blinks are accompanied by abrupt amplitude jumps in frontal regions areas (stereotypical artifact-specific temporal feature). Additionally, ADJUST determines the spatial topography of the IC weights to compare the magnitude of the amplitudes between frontal and posterior areas (stereotypical artifact-specific spatial feature). Using the ADJUST procedure, on average 1.65 (SEM = 0.13) components per participant were detected and removed. Previous data shows that the automatic detection of artifact components using ADJUST, leads to a comparable classification of artifact components that is afforded by manual classification by experiments (Mognon et al., 2011).”

References

Adams CD. 1982. Variations in the sensitivity of instrumental responding to reinforcer devaluation. The Quarterly Journal of Experimental Psychology Section B 34:77–98.

doi: 10.1080/14640748208400878.

Agresti A, Caffo B. 2000. Simple and effective confidence intervals for proportions and differences of proportions result from adding two successes and two failures. American Statistician 54:280. doi: 10.2307/2685779.

Chaumon M, Bishop DVM, Busch NA. 2015. A practical guide to the selection of independent components of the electroencephalogram for artifact correction. Journal of Neuroscience Methods 250:47–63. doi: 10.1016/j.jneumeth.2015.02.025.

Dickinson A, Balleine B. 1994. Motivational control of goal-directed action. Animal Learning & Behavior 22:1–18. doi: 10.3758/BF03199951.

McNamee D, Liljeholm M, Zika O, O'Doherty JP. 2015. Characterizing the associative content of brain structures involved in habitual and goal-directed actions in humans: A multivariate FMRI study. Journal of Neuroscience 35:3764–3771. doi: 10.1523/JNEUROSCI.4677-14.2015.

Mognon A, Jovicich J, Bruzzone L, Buiatti M. 2011. ADJUST: An automatic EEG artifact detector based on the joint use of spatial and temporal features. Psychophysiology 48:229–240. doi: 10.1111/j.1469-8986.2010.01061.x.

Müller-Putz G, Scherer R, Brunner C, Leeb R, Pfurtscheller G. 2008. Better than random? A closer look on BCI results. International Journal of Bioelectromagnetism 10:52–55.

Noh E, Herzmann G, Curran T, Sa VR de. 2014. Using single-trial EEG to predict and analyze subsequent memory. NeuroImage 84:712–723. doi: 10.1016/j.neuroimage.2013.09.028.

Pool ER, Gera R, Fransen A, Perez OD, Cremer A, Aleksic M, Tanwisuth S, Quail S, Ceceli AO, Manfredi DA, Nave G, Tricomi E, Balleine B, Schonberg T, Schwabe L, O'Doherty JP. 2022. Determining the effects of training duration on the behavioral expression of habitual control in humans: a multilaboratory investigation. Learning & Memory 29:16–28. doi: 10.1101/lm.053413.121.

Tricomi E, Balleine BW, O'Doherty JP. 2009. A specific role for posterior dorsolateral striatum in human habit learning. European Journal of Neuroscience 29:2225–2232. doi: 10.1111/j.14609568.2009.06796.x.

Valentin VV, Dickinson A, O'Doherty JP. 2007. Determining the neural substrates of goal-directed learning in the human brain. Journal of Neuroscience 27:4019–4026.

doi: 10.1523/JNEUROSCI.0564-07.2007.

Wit S de, Corlett PR, Aitken MR, Dickinson A, Fletcher PC. 2009. Differential engagement of the ventromedial prefrontal cortex by goal-directed and habitual behavior toward food pictures in humans. Journal of Neuroscience 29:11330–11338. doi: 10.1523/JNEUROSCI.1639-09.2009.

Wit S de, Kindt M, Knot SL, Verhoeven AAC, Robbins TW, Gasull-Camos J, Evans M, Mirza H, Gillan CM. 2018. Shifting the balance between goals and habits: five failures in experimental habit induction. Journal of Experimental Psychology: General 147:1043–1065. doi: 10.1037/xge0000402.

Yeung N, Holroyd CB, Cohen JD. 2005. ERP correlates of feedback and reward processing in the presence and absence of response choice. Cerebral cortex (New York, N.Y.: 1991) 15:535–544. doi: 10.1093/cercor/bhh153.

[Editors' note: further revisions were suggested prior to acceptance, as described below.]

Reviewer 2 is still confused about the intended, claimed, and actual role of overtraining. I think the easiest way to deal with this is to not talk about "extended" training (what does that mean after all – "extended" could mean an hour, days, or months) but instead characterize effects as training-dependent based on devaluation at the beginning, middle or end of a task. You can be clear that you are assessing training-dependent dynamics, but that the extent of training here is quite limited compared to that in the overtraining literature. In other words, training-dependent effects do not necessarily imply overtraining effects – please make that distinction and you should be good.

We thank the Editor for the very helpful advice. We now make the distinction between training-dependent effects and overtraining effects clearer and also omit the term “extended” training, as suggested. Instead, we talk now about effects at the beginning, middle or end of the task. Moreover, we make the distinction between training-dependent effects and overtraining effects explicit, as suggested.

Of even greater importance is the request by Reviewer 1, that you provide more details about the searchlight analysis.

We now provide more details on the searchlight analysis, as requested by the Reviewer. For details, please see our response to the Reviewer below.

Reviewer #2 (Recommendations for the authors):The authors have improved the manuscript. I think it's a much better version than the previous one. They have deleted all those paragraphs that made their arguments and motivation for the experiment confusing.I'm still super confused about their argument that they are not testing for overtraining effects. I thought overtraining was by definition the amount of training that produced habits. Are they saying in their reply that the effect of stress speeds up habit formation? What is their view on this? If their aim was not to test for "training extension" effects, why are they doing two devaluation manipulations?I don't think is enough to add this paragraph in light of my comment:"Although we did not aim to test training-induced changes in the balance of outcome and response processing per se, we included transient outcome devaluations after initial, moderate, and extended training in the instrumental learning task."And then, a few paragraphs later, saying the following:"To assess whether the balance of goal-directed and habitual behaviour and its modulation by stress depends on the extent of training, we presented devaluation blocks early during training, after moderate training, and again after extended training at the end of the task."And, having a section called "Stress boosts response representations after extended training."As a reader, I'm extremely confused about these apparent contradictions. I'd love to see in the introduction a more precise paragraph where their expectations are clearly mentioned.This comment has been carrying over since the first review when I made the following comment:"It was not clear to me what the main hypothesis of the study was. The authors seem to talk rather loosely about habits being developed after overtraining versus the mediating effect of stress on habits. The introduction should convey their main goal more clearly."Sorry if I'm being too stupid, but it's not clear to me why they are using training extension and stress to test devaluation sensitivity and outcome/response representations if their aim was not to overtrain participants.

We included devaluations tests at the beginning, middle and end of the task to assess whether stress effects on the control of instrumental behaviour are training-dependent, as suggested by earlier rodent studies. It is, however, important to distinguish between training-dependent stress effects and overtraining effects per se. As suggested by the editors “training-dependent effects do not necessarily imply overtraining effects”. In order to make this aspect clearer and avoid confusion, we do not speak of “moderate” or “extended” training any more. Moreover, we state that we are interested in training-dependent dynamics of stress effects and explicitly distinguish training-dependent effects of stress from overtraining effects.

Please see, for example, page 6, lines 120 to 130:

Because previous rodent studies suggested that stress or stress hormone effects on the balance of goal-directed and habitual forms of learning are training-dependent (Dias-Ferreira et al., 2019; Packard, 1999; Siller-Pérez et al., 2017), we also assessed training-dependent dynamics in the stress effect on outcome and response processing. Although we did not aim to test overtraining-induced changes in the balance of outcome and response processing, for which the number of trials may have been too limited in the present study, we included transient outcome devaluations at the beginning, middle and end of the instrumental learning task to assess whether stress effects on instrumental behaviour are training-dependent.

Please see also the changes of the subheadings on pages 19 and 21:

“Stress reduces outcome representations at the end of training”

“Stress boosts response representations at the end of training”

And on page 27, line 203:

“Thus, training-dependent effects do not necessarily imply overtraining effects.”